# Quantifying uncertainties of climate signals in Chemistry Climate Models related to the 11–year solar cycle. Part I: Annual mean response in heating rates, temperature and ozone

Markus Kunze[1], Tim Kruschke[5], Ulrike Langematz[1], Miriam Sinnhuber[3], Thomas Reddmann[3], and Katja Matthes[2,4]

[1]Institut für Meteorologie, Freie Universität Berlin, 12165 Berlin, Germany
[2]Research Division Ocean Circulation and Climate, GEOMAR Helmholtz Centre for Ocean Research, 24105 Kiel, Germany
[3]Karlsruhe Institute of Technology, 76021 Karlsruhe, Germany
[4]Christian-Albrechts Universität zu Kiel, 24105 Kiel, Germany
[5]Swedish Meteorological and Hydrological Institute - Rossby Centre, Norrköping, Sweden

*Correspondence to:* Markus Kunze (markus.kunze@met.fu-berlin.de)

**Abstract.** Variations of the solar spectral irradiance (SSI) with the 11-year sunspot cycle have been shown to have a significant impact on temperatures and the mixing ratios of atmospheric constituents in the stratosphere and mesosphere. Uncertainties in modelling the effects of SSI variations arise from uncertainties in the empirical models reconstructing the prescribed SSI data set as well as from uncertainties in the chemistry-climate model (CCM) formulation. In this study CCM simulations with the ECHAM MESSy Atmospheric Chemistry (EMAC) model and the Community Earth System Model 1 (CESM1)–Whole Atmosphere Chemistry Climate Model (WACCM) have been performed to quantify the uncertainties of the solar responses in chemistry and dynamics that are due to the usage of five different SSI data sets or the two CCMs. We apply a two-way analysis of variance (ANOVA) to separate the influence of the SSI data sets and the CCMs on the variability of the solar response in shortwave heating rates, temperature and ozone. The solar response is derived from climatological differences of time slice simulations prescribing SSI for the solar maximum in 1989 and near the solar minimum in 1994. The SSI values for the solar maximum of each SSI data set are created by adding the SSI differences between November 1994 and November 1989 to a common SSI reference spectrum for near solar minimum conditions based on ATLAS-3 (Atmospheric Laboratory of Applications and Science-3). The ANOVA identifies the SSI data set with the strongest influence on the variability of the solar response in shortwave heating rates in the upper mesosphere and in the upper stratosphere/lower mesosphere. The strongest influence on the variability of the solar response in ozone and temperature is identified in the upper stratosphere/lower mesosphere. However, in the region of the largest ozone mixing ratio, in the stratosphere from 50 to 10 hPa, the SSI data sets do not contribute much to the variability of the solar response when the Spectral And Total Irradiance REconstructions-T (SATIRE-T) SSI data set is omitted. The largest influence of the CCMs on variability of the solar responses can be identified in the upper mesosphere. The solar response in the lower stratosphere also depends on the CCM used, especially in the tropics and northern hemispheric subtropics and mid latitudes, where the model dynamics modulate the solar responses. Apart from the upper mesosphere, there are also regions where the largest fraction of the variability of the solar response is explained by randomness, especially for the solar response in temperature.

# 1 Introduction

Solar ultraviolet (UV) radiation is largely absorbed in the stratosphere and mesosphere, thereby heating these regions and forming the ozone layer, filtering the most harmful part out of the solar spectrum and protecting life on Earth. Especially the UV wavelengths from 120 to 380 nm are subject to variations with the 11–year solar cycle ranging between 50 and 0.5%, whereas the respective total solar irradiance (TSI) variation is only about 0.07%. The response of the middle atmosphere to the 11–year variations of solar activity has been addressed in numerous studies over the recent decades. Early studies were confined to the lower stratosphere, using stratospheric analyses based on radiosondes (Labitzke, 1987; Labitzke and van Loon, 1988). Enhanced evidence for an effect of solar variability on middle atmospheric temperatures and constituents has been achieved since satellite observations are available (e.g., McCormack and Hood, 1996; Soukharev and Hood, 2006; Randel and Wu, 2007; Maycock et al., 2016; Ball et al., 2019). Modelling studies range from early investigations with 2-dimensional atmospheric and chemistry models (Garcia et al., 1984; Haigh, 1994) and 3-dimensional general circulation models (GCMs) (e.g., Matthes et al., 2004) to studies with advanced chemistry-climate models (CCMs) (SPARC CCMVal, 2010) and CCMs coupled to an ocean model, as partly used within the 5th Coupled Model Intercomparison Project (CMIP5) (Mitchell et al., 2015; Misios et al., 2015; Hood et al., 2015).

While a good understanding of the chemical and dynamical implications of the 11–year solar cycle on the middle atmosphere has been achieved (e.g., Gray et al., 2010, 2013; Ermolli et al., 2013), there are still discrepancies between observed responses to the solar cycle and modelling studies, as well as between different models. The variability induced by the 11–year solar cycle SSI and TSI variations is part of the natural variability of the climate system. Besides the ability of GCMs and CCMs to model the right climatological state of the atmosphere and its chemical species, it is also an important aspect of climate models to realistically reproduce this natural variability. As explained in the following, there are two major sources for the uncertainty in the modelled solar responses: differences in the model formulation such as for example the implemented UV radiation parameterisations, photolysis schemes and dynamical characteristics of the models, and the prescribed solar irradiance data set.

The impact of the 11–year solar cycle on the atmosphere can be separated into two parts: a) an influence via the absorption of UV-radiation by ozone and oxygen in the middle atmosphere, with the direct solar heating response inducing a dynamical signal that propagates downward through the so-called "top-down" mechanism (Kodera and Kuroda, 2002), and b) an influence based on the absorption of the visible and infrared parts of the solar spectrum at the surface (often expressed by variations in TSI), which is amplified by the so-called "bottom-up" mechanism (van Loon et al., 2007; Meehl et al., 2009). A realistic modelling of the "top-down" effect therefore relies on prescribing spectrally resolved solar irradiances (SSI) for the radiation and photolysis parameterisations of the applied CCMs. The SSI variation over the 11–year solar cycle leads to a modulation of stratospheric ozone through photochemistry (e.g., Haigh, 1994; Ball et al., 2014), as well as variations of the heating rates in the middle atmosphere through absorption of UV-radiation (e.g., Garcia et al., 1984).

It has been shown that the spectral resolution of the radiation parameterisation has a large impact on the 11-year solar response of the short-wave (SW) heating rates and is one source of uncertainty in model simulations (Nissen et al., 2007;

Forster et al., 2011). The SSI dependent changes in both ozone and SW heating rates over the 11–year solar cycle determine the resulting solar induced temperature signal. SPARC CCMVal (2010) identified a large model spread in solar responses for ozone and temperature of 18 CCMVal-2 CCMs, mainly caused by differences in the spectral resolution of the SW radiation parameterisations or the treatment of photolysis in the CCMs. The simulated solar response in annual mean tropical (25°S–

25°N) temperature (1960–2004) near the stratopause ranges from 0.45 to 1.4 K, whereas the SSU satellite data (1979–2005) show 0.85 K for a comparable height region, and ERA-40 reanalyses (1979–2001) show 1.4 K. The annual mean solar response in ozone mixing ratio for the same region and time frame shows less model spread with an ozone increase of 2% in the upper stratosphere, which is in good agreement with observations. Towards lower altitudes the model spread increases and discrepancies to the observations get larger (SPARC CCMVal, 2010).

Model intercomparisons as CCMVal-2, CCMI, and CMIP5 focus on the solar response in the troposphere and stratosphere. Higher up in the mesosphere, where shorter wavelengths are not absorbed yet, the irradiance variations over the 11–year solar cycle are even larger and have a strong effect on atmospheric trace gases, like $H_2O$ and $CO_2$, where they produce large solar responses in $HO_x$, CO and also affect $O_3$ by subsequent catalytic cycles (e.g. Marsh et al., 2007; Merkel et al., 2011; Beig et al., 2012).

The SSI data prescribed in the models are the second source of uncertainty when modelling the solar response. Shapiro et al. (2011) investigated the influence of the 27-day variations of four different SSI observations on the chemistry of the upper mesosphere in a 1-dimesional radiative-convective-chemistry model. The deviant solar cycle behaviour of the SORCE (Solar Radiation and Climate Experiment) measurements has motivated a number of CCM studies (e.g. Haigh et al., 2010; Merkel et al., 2011; Ball et al., 2011, 2016; Swartz et al., 2012) comparing simulations using prescribed SORCE SSI data with

reconstructed SSI of the Naval Research Laboratory (NRLSSI) or the Spectral And Total Irradiance REconstructions (SATIRE) model.

Besides its role for the solar cycle response of the thermal structure and dynamics in the middle atmosphere, a different spectral distribution of a SSI data set can also have an impact on the averaged middle atmospheric temperature, as was found in studies comparing different SSI data sets. It was shown that differences in the energy distribution during the solar minimum

phases of individual SSI data sets may cause significant differences in the simulated temperatures in the middle atmosphere (e.g., Zhong et al., 2008; Oberländer et al., 2012). Even when scaled to the same TSI, the variable spectral distribution of energy within the SSI data sets can cause significant changes of the simulated climatological temperatures in the middle atmosphere. As shown in Matthes et al. (2017), climatological annual mean middle atmospheric temperatures in the tropics can be up to 1.6 K lower when using the CMIP6 recommended SSI data set instead of NRLSSI1. Therefore, when isolating the pure effect of

the amplitude of SSI changes over the solar cycle, it is necessary to use a common SSI data set for solar minimum conditions that serves as a baseline for adding the solar amplitude of the different TSI/SSI data sets. By adding the solar amplitude, i.e. the differences of the TSI/SSI between the solar maximum and solar minimum state of the different SSI data sets, to the reference state, the effects of the 11–year solar cycle differences in spectral distribution and amplitude of the individual SSI data sets can be quantified in a more appropriate way than in previous work.

The aim of this study is to estimate the uncertainty of the solar cycle signal resulting from the two above described sources of uncertainty: the specification of the 11-year solar cycle SSI amplitude and the models' SW radiation and photolysis schemes and their dynamical characteristics. In this Part I of our study we concentrate on the annual mean solar response in heating rates, temperature and ozone, while Part II (in preparation) focuses on the dynamical solar and auroral responses in northern winter.

We apply different SSI data sets in two CCMs, EMAC and CESM(WACCM), as described in Section 3 to identify regions where significant differences in the solar responses can be attributed to differences in either the SSI data sets or the CCMs. Both CCMs have participated and their solar responses have been evaluated in the CCMVal-2 activity (SPARC CCMVal, 2010). Here, we use five different SSI data sets that are all based on empirical or semi-empirical models of SSI and TSI (see Section 2 for more details). In contrast to previous studies, we use a common observation based reference SSI spectrum for
solar minimum conditions where the five SSI amplitudes are added to create the solar maximum forcings (s. Section 3.1). After analysing the solar response of the ensemble mean and its variance in Section 4, the individual solar responses are discussed in Section 5. To separate the influence of the SSI data sets and the CCMs on the solar responses in SW heating rates, temperature and ozone, a two-way analysis of variance (ANOVA) method (e.g Fisher, 1925; von Storch and Zwiers, 1999) has been applied. While the ANOVA is a well established method in many scientific fields, it is used rarely in the field of
climate research (e.g Geinitz et al., 2015; Evin et al., 2019). Here we use ANOVA for the first time to quantify the uncertainty of the atmospheric response to decadal solar variability. The ANOVA-approach enables us to analyse if the usage of different CCMs or different SSI forcing data sets yields significantly different solar responses and to quantify which share of the total variance of the ensemble's solar response is related to either of the two factors (called treatment in the ANOVA-context). The climatological differences between both CCMs, that are partially responsible for differences in the solar responses, are
discussed in Section 5.2. In Section 6 the solar response in total column ozone and its variability are analysed, followed by a summary and conclusions in Section 7.

## 2  Spectral solar irradiance data sets

The record of observed TSI covers a relatively short period of time, since the first satellite mission to monitor TSI was launched in 1978. The record of SSI observations is even shorter and does not cover the solar spectrum continuously, as required for
climate modelling studies. The construction of a continuous SSI data set exploiting all available space-borne measurements was only recently addressed by the "First European Comprehensive Solar Irradiance Data Exploitation project" (SOLID) (Haberreiter et al., 2017). In order to perform multi-decadal simulations with GCMs and CCMs covering the recent past and the near future, as done within CMIP5 and the Chemistry-Climate Model Initiative (CCMI), SSI and TSI data sets are needed that are based on reconstructions with empirical models. Such models rely on SSI and TSI proxy data, which are available for
longer time periods. However, there is an ongoing debate about the reliability of TSI and SSI reconstructions. Coddington et al. (2019) compare solar amplitudes of 11–year solar cycles in the satellite period produced with the NRLSSI2 and SATIRE-S data sets for a number of broad wavelength bands to SSI amplitudes derived from the SOLID composite. In the FUV spectral region, they report the highest SSI amplitude for the SATIRE-S data set and a negative secular interminima trend over the

satellite period in the SATIRE-S TSI and SSI from the FUV to NIR spectral regions which is not present in any observational record or other TSI/SSI reconstructions. Yeo et al. (2015) compare the SSI variability of NRLSSI1 and SATIRE-S with SSI observations over the satellite period and report a low UV variability of NRLSSI1 compared to the SATIRE-S data set, whereas the latter is in better agreement to the satellite SSI observations. The SSI/TSI reconstructions of SATIRE, NRLSSI/TSI, and the combination of both in the CMIP6 SSI/TSI data set are the most common SSI/TSI data sets used in GCMs and CCMs and, therefore, subject to our investigation.

The standard data set for TSI and SSI in recent model intercomparison studies like CMIP5, CCMVal-2, and CCMI was the NRLSSI1 data set (Lean, 2000; Wang et al., 2005). This data set is known to have the lowest solar cycle variability in the spectral range from 200–400 nm, compared to other SSI reconstructions and measurements (Ermolli et al., 2013). This is of particular importance for the formation of ozone and the related heating rates in the middle atmosphere (Ermolli et al., 2013). The low variability of the NRLSSI1 data set might lead to an underestimation of simulated 11–year solar cycle effects in global models (Ermolli et al., 2013) and motivated the compilation of a new TSI/SSI data set to be used for CMIP6 (Matthes et al., 2017) which is a combination of two data sets: NRLSSI2 (Coddington et al., 2016) and SATIRE (Krivova et al., 2009; Yeo et al., 2014). In the following subsection we give a brief introduction of the SSI/TSI data sets applied in this study.

**ATLAS3 based reference spectrum**

The SSI data set used in this study for the solar minimum reference state between 0.1–2,395 nm is the Atmospheric Laboratory of Applications and Science-3 (ATLAS-3) SSI reference spectrum (Thuillier et al., 2004), obtained during the third ATLAS mission in November 1994 near the minimum of solar cycle 22. It is a composite spectrum that comprises SSI measurements from instruments on three space platforms, including measurements with the SOLar SPECtrum instrument (SOLSPEC) and the Solar Ultraviolet Spectral Irradiance Monitor (SUSIM) experiment on board of the space shuttle (see Thuillier et al., 2004, for more details). The ATLAS-3 SSI data set covers wavelengths up to 2,395 nm only. We use the NRLSSI1 data set for wavelengths between 2,395–99,975 nm and the SATIRE-S data set from 99,975–165,000 nm to extend the spectrum to the infrared and to derive the TSI for the ATLAS-3 spectrum. To assure smooth transitions at 2,395 and 99,975 nm, the NRLSSI1 and SATIRE-S data sets are scaled accordingly. The extended ATLAS-3 spectrum was then scaled with a constant factor to obtain the integrated TSI of 1361.05 W m$^{-2}$ for November 1994, derived from Total Irradiance Monitor (TIM) TSI measurements on NASA's Solar Radiation and Climate Experiment (SORCE) (Kopp and Lean, 2011). The resulting compiled and scaled SSI data set serves as a reference state for solar minimum conditions to which the solar amplitudes of all other SSI data sets have been added to get the respective SSI data sets for solar maximum condition.

**NRLSSI**

The Naval Research Laboratory (NRL) SSI models (Lean et al., 1997; Lean, 2000; Coddington et al., 2016) are based on the empirical, wavelength-dependent relationship between sunspot darkening and facular brightening on the solar disk with SSI changes. Indices which are derived from observations and proxies for sunspot darkening and faculae are used in regression models to determine the coefficients required to estimate the time-varying SSI changes. The SSI changes of the empirical

model are added to a quiet sun reference state, based on the WHI (whole heliosphere interval) SSI reference spectrum and the ATLAS-1 measurements (Thuillier et al., 1998). The TSI changes are added to a quiet Sun reference state of 1365.5 W m$^{-2}$ (NRLSSI1) and 1360.45 W m$^{-2}$ (NRLSSI2), based on SORCE/TIM measurements (Kopp and Lean, 2011). The required model coefficients are determined from a multiple linear regression of the proxy time series on the observed TSI from SORCE/TIM and observed SSI from SORCE/SOLSTICE and SORCE/SIM for NRLSSI2 and UARS/SOLSTICE for NRLSSI1. For facular brightening the composite MG II index of the University of Bremen (Snow et al., 2014) and as index for sunspot darkening the sunspot area as recorded by ground-based observatories are used (Lean et al., 1998).

**SATIRE**

The SATIRE (Spectral And Total Irradiance REconstructions) model (Krivova et al., 2009; Yeo et al., 2014) for the reconstruction of SSI and TSI is a semi-empirical model that is based on variations of the solar surface magnetic field. The intensity spectra of the quiet Sun reference state, faculae, network, sunspot umbrae, and sunspot penumbrae are derived by applying a radiative transfer code (Unruh et al., 1999; Yeo et al., 2014). The resultant SSI is given as a weighted sum of these five contributions, where the weights (filling factors) are retrieved from magnetograms and continuum images that allow to estimate the fractional solar surface that is covered by the brightening (faculae and network) and darkening (sunspot umbrae and penumbrae) features. Two different data sets from the SATIRE model that both span a wavelength range from 115 – 160000 nm, SATIRE-T (telescope era) and SATIRE-S (satellite era), are used in this study. Differences between SATIRE-T and SATIRE-S arise from the estimation of the filling factor that describes the fractional surface coverage of the quiet Sun and the brightening and darkening features. SATIRE-S relies on full-disc magnetograms and intensity images, which allow to reconstruct TSI and SSI back to 1974. Satire-T (Krivova et al., 2010) is intended to reconstruct the SSI/TSI in the pre-satellite era when only lower quality data for the estimation of the state of the photosphere are available. Whereas for SATIRE-S detailed information of the photospheric structure can be used, it is assumed to be homogeneous for SATIRE-T. The SATIRE-T filling factors for sunspot umbrae and penumbrae are calculated from the observed sunspot areas. The filling factors for faculae and network are derived from the evolution of the solar photospheric magnetic flux estimated by a coarse physical model (Solanki et al., 2000). As our study is based on the SSI/TSI data of November 1994, the most reliable reconstruction of the SATIRE model is given by the SATIRE-S data set, however SATIRE-T is also included for comparison. The direct comparison of SATIRE-T and SATIRE-S for the same time frame can be beneficial for modelling studies using SATIRE-T in the pre-satellite era (e.g., the Maunder Minimum) and comparing to simulations for present-day conditions, which also use SATIRE-T SSI.

**CMIP6 data set**

The SSI and TSI data sets of the NRLSSI2 and SATIRE-S/T models, introduced in the previous sections, both cover the required time span (1850–2300) for CMIP6 simulations, and have been widely tested in modelling studies. The new recommended SSI/TSI data set for CMIP6 has been derived by averaging the NRLSSI2 reconstructions with the SSI and TSI of SATIRE-S/T as described in detail in Matthes et al. (2017). As we only use data for November 1989 and November 1994, the

CMIP6 SSI and TSI data consists of an average of output from NRLSSI2 and SATIRE-S.

[Table 1 about here.]

[Figure 1 about here.]

[Figure 2 about here.]

The quantification of the uncertainties of the solar responses in the CCMs is based on an exemplary solar amplitude with decreasing solar irradiances from the maximum of solar cycle 22 in November 1989 to a state near the solar minimum in November 1994, which is motivated by the timing of the ATLAS3 measurements. Table 1 gives an overview of the applied SSI data sets with details of their percentage solar cycle amplitude in the Lyman-$\alpha$, far-UV (FUV, 121–200 nm), UV in the Herzberg continuum (partly overlapping with the Hartley-bands) (201–242 nm), UV in the Hartley-/Huggins-bands (243–380 nm), and the visible (381–780 nm) spectral regions. The first value ($\Delta$SSI) represents the percentage SSI change from solar minimum to maximum relative to the ATLAS3 solar minimum SSI in November 1994, and is also shown in Figure 1b. The second value represents $\Delta$SSI relative to the TSI change from solar minimum to maximum ($\frac{\Delta SSI}{\Delta TSI}$) in %. Whereas $\Delta$SSI emphasises the large variability of the solar irradiance at Lyman-$\alpha$ and in the Schumann-Runge-continuum/-bands over the 11-year solar cycle, the $\Delta$SSI weighted by $\Delta$TSI emphasises the large solar cycle variation of absolute energy in the UV and visible wavelengths. High $\Delta$SSI variability in the FUV is leading to large increases in the photolysis of oxygen and water vapour in the upper mesosphere during solar maximum. More important for the solar response of stratospheric ozone mixing ratios are the two UV spectral regions. While the irradiance increases in the 201–242 nm spectral region lead to more oxygen photolysis and subsequent ozone production in the stratosphere during solar maximum, the irradiance increases between about 243–380 nm lead to more ozone destruction through photolysis during solar maximum, further discussed in section 5.1. The NRLSSI1 and NRLSSI2 data sets have the lowest $\frac{\Delta SSI}{\Delta TSI}$ ratio in the Hartley-/Huggins-UV-band among the SSI models used here, whereas the SATIRE-S data set shows the highest and has also the highest $\frac{\Delta SSI}{\Delta TSI}$ ratio in the Lyman-$\alpha$, FUV, and Herzberg continuum/Hartley bands spectral regions.

The magnitude of the applied solar amplitude in this study can be regarded as representative for the second half of the 21$^{st}$ century when the solar cycles 19 to 23 showed relatively large 11–year solar cycle amplitudes. However, the individual solar cycles show different, spectrally resolved characteristics in their amplitudes, which also differ among the individual SSI data sets. Compared to other solar cycle amplitudes in the satellite era (see Table S1 in the supplement), the one used in this study is neither especially weak nor especially strong. The averaged $\Delta SSI$ is shown in Figure 2a, with the error bars indicating the 95% confidence interval of the $\Delta SSI$ within each spectral region. The main characteristics of the solar amplitude chosen here are also present in the averaged solar cycle amplitude, such as the small solar amplitude of SATIRE-T in the FUV and most of the ranking of the SSI data sets within the spectral regions. All deviations of the chosen solar cycle amplitude from the averaged solar cycle amplitude are within the range of the 95% confidence intervals (Figure 2b). Therefore, the selected solar cycle amplitude can be regarded as representative for most of the solar cycle amplitudes of the satellite era.

Besides the ATLAS3 reference SSI data set more recent SSI reference data sets are available such as SOLAR-ISS (Meftah et al., 2020, 2018), which is representative of the 2008 solar minimum. The usage of an alternative SSI reference dataset may have an influence on the climatological state of the CCMs, as higher or lower SSI values in certain spectral bands lead to higher or lower SW heating rates, thus affecting the temperature and potentially also the zonal wind of the CCMs. As the reference SSI data set serves as a common base state for the solar minimum of all other SSI data sets, we do not expect significant differences in the uncertainties of the solar responses when using a different SSI reference spectrum.

## 3 Chemistry-climate models and simulations

Two state-of-the-art CCMs have been used in this study to quantify the uncertainty of the modelled solar response related to the 11-year solar cycle. These are EMAC (ECHAM/MESSy Atmospheric Chemistry) (Jöckel et al., 2016) and CESM1(WACCM) (Community Earth System Model 1–Whole Atmosphere Chemistry Climate Model) (Marsh et al., 2013). Both CCMs have a good spectral resolution of their SW radiation and photolysis parameterisation and therefore are well suited for this study. The main difference between EMAC and WACCM, as applied here, is WACCM's model top in the lower thermosphere which allows for a better representation of the chemical processes in the upper mesosphere in WACCM.

**EMAC**

EMAC is a CCM that includes sub-models describing tropospheric and middle atmospheric processes and their interaction with oceans, land and human influences (Jöckel et al., 2010). It uses the second version of the Modular Earth Sub-model System (MESSy2) to link multi-institutional computer codes. The core atmospheric model is the 5th generation European Centre Hamburg general circulation model (ECHAM5, Roeckner et al., 2006). For the present study we applied EMAC (ECHAM5 version 5.3.02, MESSy version 2.52; Jöckel et al., 2016) in T42L47MA-resolution, i.e. with a spherical truncation of T42 (corresponding to a quadratic Gaussian grid of approx. 2.8°latitude by 2.8°longitude) with 47 hybrid pressure levels up to 0.01 hPa ($\sim$80 km). EMAC includes the MECCA (Module Efficiently Calculating the Chemistry of the Atmosphere) (Sander et al., 2011a) chemical mechanism, which contains 155 species with 224 gas phase, 12 heterogeneous, and 74 photolytic reactions. The photolysis rate coefficients are calculated with JVAL (Sander et al., 2014) using updated rate coefficients recommended by JPL (Sander et al., 2011b) and resolves the solar Lyman-$\alpha$ line and 8 spectral bands in the UV and VIS range (178–683 nm). RAD/RAD-FUBRAD (Dietmüller et al., 2016) provides the parameterisation of radiative transfer based in the SW on Fouquart and Bonnel (1980) and Roeckner et al. (2003) (RAD) with 4 bands from 250 to 4,000 nm. For a better resolution of the UV-VIS spectral band RAD-FUBRAD is used for pressures lower than 70 hPa, increasing the spectral resolution in the UV-VIS from one band to 55, 81, or 106 bands (Nissen et al., 2007; Kunze et al., 2014). Here we use the updated version of RAD-FUBRAD with 81 bands which substitutes the single band parameterisation for the heating rates in the Schumann-Runge bands (Strobel, 1978) by the parameterisation based on 19 bands, as given in Strobel (1978), and 14 bands in the Chappuis bands (407.5–690 nm). The stratospheric equatorial zonal winds are relaxed towards an observed Quasi-Biennial Oscillation (QBO) with the submodel QBO (Giorgetta and Bengtsson, 1999). With an upper boundary in the upper mesosphere,

EMAC is not able to capture the thermospheric influx of $NO_y$. Therefore, the simulations presented in this study employ the UBCNOX parameterisation (Sinnhuber et al., 2018; Funke et al., 2016) in the upper mesosphere to include $NO_y$ produced in the thermosphere by auroral and medium-energy electrons.

**WACCM**

The Whole Atmosphere Community Climate Model (version 4; Marsh et al., 2013) is an integrative part of the *Community Earth System Model* suite (version 1.0.6; Hurrell et al., 2013). CESM1(WACCM) is a "high-top" CCM covering an altitude range from the surface to the lower thermosphere, i.e. up to $5 \times 10^{-6}$ hPa, equivalent to approx. 140 km. It is an extension of the *Community Atmospheric Model* (CAM4; Neale et al., 2013) with all its physical parameterisations. For this study the model has been integrated with a horizontal resolution of 1.9°latitude by 2.5°longitude and 66 levels in the vertical. CESM1(WACCM)

contains a middle atmosphere chemistry module based on the Model for Ozone and Related Chemical Tracers (MOZART3; Kinnison et al., 2007) which includes a total of 52 species with 127 gas-phase, 17 heterogeneous, and 48 photolytic reactions. A six constituent ion chemistry model is included with 13 ionization reactions and 14 ion-neutral and recombination reactions. Its photolysis scheme resolves 100 spectral bands in the UV and VIS range (121-750 nm). The SW radiation module is a combination of different parameterisations. Above approx. 70 km the spectral resolution is identical to the photolysis scheme

(plus the parameterisation of Solomon and Qian, 2005, based on the F10.7 cm solar radio flux to account for EUV irradiances). Below approx. 60 km the SW radiation of CAM4 is retained, employing 19 spectral bands between 200 and 5,000 nm (Collins, 1998). For the transition zone (60-70 km), SW heating rates are calculated as weighted averages of the two approaches. CESM1(WACCM) features relaxation of stratospheric equatorial winds to an observed or idealised (used here) QBO (Matthes et al., 2010). The ionisation in the auroral regions by energetic particles is parameterised according to Roble and Ridley (1987)

using the Kp index as input parameter. To achieve a setup for low auroral activity, for WACCM the Kp index is set to a constant value of Kp = 0.67, as this corresponds to a geomagnetic index of Ap = 3, which is used in the EMAC submodel UBCNOX.

### 3.1 CCM simulations

All EMAC and WACCM simulations have been made explicitly for this study, to ensure that the differences in the solar responses are exclusively related to the SSI data sets prescribed or the CCM applied, and not due to differences in the sce-

nario. For both CCMs (EMAC and WACCM) time slice simulations have been performed with the same basic scenario in all simulations, except only for the prescribed SSI data set. The basic scenario consists of year 2000 conditions for prescribed greenhouse gas mixing ratios (GHGs), ozone depleting substances (ODSs), and monthly climatological sea surface temperatures (SSTs) and sea ice concentrations (SICs) (average from 1995 to 2004). After dismissing five or three years of spinup for EMAC and WACCM, respectively, 45 years of data are available from each simulation for analyses. The QBO is included in

all simulations by relaxation of the zonal wind in the tropical lower stratosphere between 90 and 10 hPa with the strongest nudging applied from 50 to 15 hPa in EMAC. Whereas EMAC uses the time series of the observed zonal winds (Naujokat, 1986), an idealized 28-months varying QBO is used for WACCM. The reference simulation with perpetual solar minimum conditions, performed by each CCM, uses the ATLAS-3 based SSI reference spectrum (Thuillier et al., 2004, s. Section 2). To

our knowledge, this is the first time that this observational SSI dataset has been used to force CCM simulations. In addition, five sensitivity simulations with perpetual solar maximum conditions of the solar cycle 22 maximum in November 1989 from five different SSI data sets have been performed by each CCM. The five spectra for solar maximum conditions are constructed by adding the difference of the SSI between the solar maximum in November 1989 and the near solar minimum in November

1994 of NRLSSI1 (Lean, 2000), NRLSSI2 (Coddington et al., 2016), SATIRE-T (Krivova et al., 2010), SATIRE-S (Yeo et al., 2015), and CMIP6-SSI (Matthes et al., 2017) to the common ATLAS-3 based observational reference SSI data set which is defined for the solar minimum state. By this procedure we ensure that we use a common spectral distribution of energy that only varies in the solar maximum sensitivity simulations by the genuine difference from solar minimum to solar maximum of each SSI data set. Table 1 summarises the resulting percentage changes for five spectral regions, and also gives an overview of

the five sensitivity simulations for solar maximum, which have been performed by WACCM and EMAC.

The solar minimum reference simulation as well as the solar maximum sensitivity simulations are performed for low auroral activity. Due to these model configurations, the $NO_y$ changes are expected to be caused by the SSI changes from solar minimum to solar maximum and not by variations in auroral activity.

## 4   Annual mean solar response of heating rates, temperature, and ozone

To analyse the solar response, time series of anomalies have been calculated for each simulation performed by EMAC and WACCM using one of the five SSI data sets constructed for solar maximum conditions with respect to the time series of the reference simulations of both models using the ATLAS-3 based SSI near solar minimum. These five anomaly fields for each CCM can be interpreted as solar response of the model variables to SSI and TSI changes over the 11–year solar cycle. Figure 3 (left) shows the solar response of the ensemble mean, i.e. averaged over the simulations of both CCMs applying five different

SSI data sets each, in (a) SW heating rate (HR), (b) temperature, and (c) ozone mixing ratio. The averaged solar response is significant at the 95% level in regions that are not masked by grey hatching. A t-test is applied to the complete concatenated ensemble (10 simulations in each group without performing an ensemble mean of the simulations at solar maximum and minimum in advance). By this procedure the variability of the solar response is maintained and the regions where the significance reaches the 95% level are smaller compared to the results of a t-test for an ensemble average.

25                                             [Figure 3 about here.]

The solar SW heating rate response is significant throughout the middle atmosphere at pressures lower than 30 hPa with peaks in the tropics near the stratopause ($\sim$0.2 K day$^{-1}$) and in the upper mesosphere ($\sim$0.38 K day$^{-1}$) (Figure 3a). Whereas the solar response in the upper mesosphere is due to enhanced solar radiation in the spectral range from Lyman-$\alpha$ to the Schumann-Runge bands (absorbed by oxygen), the solar response near the stratopause is mainly due to an increase over the

solar cycle in the Hartley and Huggins bands (absorbed by ozone). The very low standard deviation of the solar response of the ensemble mean throughout the stratosphere and mesosphere (except for the upper mesospheric polar regions) provides evidence that the primary annual mean radiative response of the middle atmosphere is a robust feature and not particularly sensitive to the specified SSI data set or the CCM configuration.

A direct consequence of the stronger SW heating during solar maximum is a solar response in temperature, which also peaks in the upper mesosphere with up to 2 K and near the stratopause in the subtropics with more than 0.8 K (Figure 3b). This is weaker than the more than 1 K solar temperature response derived from combined SSU/MSU4 satellite data reported by Randel et al. (2009), but in the same order of magnitude as reported by SPARC CCMVal (2010) for CCMs and slightly larger than analysed by Mitchell et al. (2015) for an ensemble of CMIP5 high-top models. The secondary lower stratospheric maximum in the solar temperature response in the tropics, which has been identified in reanalyses (e.g., Frame and Gray, 2010), is not present in the ensemble mean. At high latitudes, the solar response in temperature shows generally an enhanced spread between the ensemble members which is due to the high internal dynamical variability of the polar winter atmosphere.

The solar response in ozone mixing ratio has a first peak in the stratosphere near 7 hPa with two regions exceeding 2%, one in the southern hemisphere (SH) extending from mid-latitudes to the subtropics and one in the northern hemisphere (NH) extending from the subtropics to polar latitudes. The solar ozone response decreases in the upper stratosphere and lower mesosphere, turns to negative values of up to -1.5% in the mesosphere and again to large positive values of more than 4.5% in the upper mesosphere (Figure 3c). Previous studies analysing the stratospheric solar ozone response in CCMs have found a comparable magnitude. E.g., Hood et al. (2015) found a significant 2–3% ozone response to the solar cycle between 1979 and 2005 at a slightly higher altitude near 3–4 hPa in three out of six CCMs within CMIP5. In the upper mesosphere, the solar ozone signal differs between the two CCMs: the negative response in EMAC appears only in a narrow layer from 0.1–0.03 hPa in WACCM, where it strongly increases above, dominating the ensemble mean ozone response (s. Figure 5c). The negative solar ozone signal in EMAC is due to a strong enhancement of $H_2O$ photolysis during solar maximum leading to an increase in $HO_x$ and enhanced catalytic ozone depletion. The divergent solar responses in ozone in the upper mesosphere between EMAC and WACCM (Figure 5c) are further discussed in Section 5.1.

## 5   Uncertainty in solar response due to SSI data sets and CCMs

[Figure 4 about here.]

The averaged solar response discussed in Section 4 is supposed to be different with respect to the SSI data set prescribed and with respect to the CCM applied in each run. From the differences in the SSI amplitudes in the broad bands ($\Delta$SSI in Table 1, Figure 1b) we expect the solar responses to be slightly different for each SSI data set, as we do for each of the CCMs. To quantify the uncertainty of the mean solar response emerging from the usage of different SSI data sets on the one hand and different CCMs on the other, we apply a two-way analysis of variance (ANOVA) approach in this study (s. Appendix A for details). The two-way ANOVA is used, as there are two treatments influencing the annual mean solar responses. As these treatments are not applied independently the interactions of the CCMs and SSI data sets have to be taken into account.

The data set to analyse consists of the annual mean solar responses from simulations using the five SSI data sets in both CCMs, i.e. a time series with 450 years from 10 simulations with 45 years each, as shown for the ensemble average in Figure 3. The overall variance of the solar response, as expressed by its annual standard deviation (Figure 3, white contours), is partitioned according to Equation A3 into a contribution arising from the applied SSI data sets ($SS_{bB}$), a contribution arising

from the applied CCMs ($SS_{bA}$), the interaction of both treatments ($SS_{bAB}$), and an error term ($SS_w$), describing the random, unexplained interannual variances. The results of the ANOVA for the solar response are shown in Figure 4. The adjusted coefficients of determination $R_{a,B}$ (left column), $R_{a,A}$ (middle column), and $R_{a,AB}$ (right column) (Equations A11, A12) are coloured and additional white contours are included when the values reach or exceed the limits of the colour coding. They

indicate the percentage of the solar response variance that can be explained by the differences between the SSI data sets, the CCMs, or the interaction of both treatments, respectively. Superimposed grey hatching masks areas where the solar responses are not significantly different when grouped according to the SSI data sets or the CCMs. Note that the contributions of the variances explained by the SSI data set, the CCM, and the interaction of both in Figure 4 do not add up to 100%, as often the random contribution to the total variance is largest.

Significant differences in the solar response can be identified when the simulations of the ensemble mean are grouped according to the SSI data sets (Figure 4, left) and the CCMs (Figure 4, middle). In the upper mesosphere, differences between the CCMs explain more than 80% of the spread in the solar ozone signal – due to the change of sign in the ozone response between WACCM and EMAC –, more than 25% of the spread of the solar temperature signal, and up to 50% of the spread of the SW heating rate signal at high latitudes. In contrast, between 60°S and 60°N up to 70% of the spread of the solar response

in SW heating rates is explained by the different SSI data sets, due to large solar amplitude variations in the FUV between the SSI data sets. As a result, the SSI data sets also induce a significant fraction of the variability of the solar response in temperature (9%) at the mesopause.

In the upper stratosphere and lower mesosphere, the SSI data sets are responsible for a relatively large part of the variance of the solar responses, while the CCMs explain much less variance as the radiation schemes of both models possess a sufficient

spectral resolution to capture the SW heating rate peak. The largest fractions of the SSI induced variance of the solar response in SW heating rates (30%) and ozone (30%) peak in the subtropics, while the SSI induced variance of the temperature solar response (10%) maximizes in the tropics.

Compared to $R_{a,A}$ and $R_{a,B}$, the fraction of explained variance by the interaction of the SSI data set and CCM ($R_{a,AB}$) is only small. Some significant differences of the solar responses in SW heating rates and ozone are explained by the interaction

in the upper mesosphere, and near the stratopause for temperature.

In the lower stratosphere and troposphere, the SSI data sets do not contribute significantly to the variability of the solar response in annual mean SW heating rates, temperature, and ozone. But some significant contributions of the applied CCMs to the variance of the solar response are found, which have similar vertical and latitudinal structures for all three variables, peaking in the tropics between 30 and 10 hPa and in the northern subtropics to mid-latitudes between 100 and 10 hPa. The

minor CCM contribution to the SW heating rate solar response in the lower stratosphere (p > 10 hPa) is consistent with the CCM induced variance contribution of ozone and seems to be related to differences in ozone transport affecting SW ozone absorption.

The simulations performed with the SATIRE-T data set show considerable deviations in the solar response compared to simulations using the other data sets, as further discussed in Section 5.1. When omitting the simulations with the SATIRE-T

data set in the ANOVA (see Figure S1 in the supplement), less variability is explained by the SSI data set, revealing that a large fraction of the variability attributed to the SSI data set is caused by the specific behaviour of SATIRE-T.

[Figure 5 about here.]

## 5.1 Differences resolved by SSI data set

The ANOVA of the ensemble mean in the previous section has shown that significant differences in the solar responses can be attributed to the SSI data sets mainly in the region of the most active ozone production in the upper stratosphere to lower mesosphere, whereas significant differences attributed to the CCMs are mainly located in the upper mesosphere and lower stratosphere, further discussed in Section 5.2. In this section, we examine the differences in the solar responses of EMAC and WACCM arising from the use of the different SSI data sets. The atmospheric response to solar irradiance variations is primarily determined by radiative and photochemical processes that are represented by a variety of parameterisations in climate models or CCMs. To separate the effects of differences in the input SSI data sets on the respective SW radiation and photolysis schemes of EMAC and WACCM, we first present profiles for simulations applying the five individual SSI datasets, averaged over both CCMs. Figure 5 shows vertical profiles of the solar cycle amplitude in SW heating rates, temperature, ozone, atomic oxygen ($O(^3P)$ and $O(^1D)$), $NO_y$, $HO_x$, and water vapour ($H_2O$), averaged from 60°S to 60°N between 100 hPa close to the tropopause and 0.01 hPa in the upper mesosphere. For each sub-figure, the 11-year solar response is at first calculated as the difference between the individual solar maximum simulations and the ATLAS3 solar minimum reference simulation for EMAC and WACCM, respectively. The differences are then grouped according to the SSI data set and averaged over EMAC and WACCM. As in Figure 3 (left), a t-test is applied to the solar response of the complete ensemble. The 95% confidence interval from this test is included as error bars in each panel of Figure 5. To better assess the solar responses of the photochemically influenced quantities shown in Figure 5, the solar responses of the photolysis rates of EMAC and WACCM (averaged over both CCMs as for Fig. 5) are shown in Figure 6 for a single simulation time step in January at 180°E, averaged from 60°S to 60°N. The shaded areas in figures1 5 and 6 indicate the range of the solar responses between the EMAC and WACCM ensemble means, enframed by the dotted and dashed black contours for the EMAC and WACCM ensemble means, respectively. By comparing the CCM averaged profiles for individual SSI datasets with the SSI averaged ensemble mean for each CCM (shading) the relative roles of the SSI datasets and the CCMs can directly be inferred for the different quantities and altitude regions.

The average over both CCMs shows the strongest stratospheric solar response in SW heating rates and temperatures when using the SATIRE-S data set, whereas SATIRE-T leads to the weakest solar response throughout the middle atmosphere (Figure 2a,b). In the upper mesosphere, these differences are a direct consequence of the magnitude of the FUV-amplitude over the solar cycle (s. Table 1) with SATIRE-S showing the largest and SATIRE-T showing the smallest solar response in FUV-heating by oxygen absorption/photolysis and subsequent temperature increase. In the stratosphere, the photochemical Chapman cycle is more effective during solar maximum, as shown with the positive solar responses of atomic oxygen ($O(^1D)$, $O(^3P)$, Figure 5d,e) and the photolysis rates of oxygen and ozone ($JO_2$, $JO_3 \rightarrow O(^1D)$, $JO_3 \rightarrow O(^3P)$, Figure 6a,c,d). This results in a positive solar response in ozone, peaking in the upper stratosphere near 7 hPa (Figure 5c). The SATIRE-T solar

response in ozone is the weakest in this comparison, as its SSI amplitude in the FUV and 201–242 nm spectral range, important for the photochemical ozone production, is considerably weaker (7.6% and 2.6%) than in the other SSI data sets (11.1–12.1% and 3.3–3.6%) (Table 1), whereas its SSI amplitude in the ozone-destroying UV-band (243–380 nm) is comparable to the other SSI data sets. The two competing, wavelength dependent effects of ozone production and ozone loss in the Chapman cycle

lead to the relative weak solar ozone response in simulations using the SATIRE-T data set. Compared to the SATIRE-T-based simulations, the simulations using the remaining SSI data sets show solar responses in SW heating rates, temperature, and ozone, that are relatively close to each other.

[Figure 6 about here.]

    Besides the oxygen chemistry of the Chapman cycle, ozone depleting catalytic cycles, e.g. the $HO_x$ and $NO_x$ cycles,

are involved in contributing to the solar response in ozone. Whereas the $HO_x$ catalytic cycle is the most important one in the upper mesosphere, the $NO_x$ catalytic cycle dominates in the middle and upper stratosphere. $NO_y$ is specified as $NO_y = N + NO + NO_2 + NO_3 + 2\,N_2O_5 + HNO_3 + HNO_4 + ClNO_2 + BrNO_3$, but in the mesosphere $NO_y$ is very close to the active nitrogen defined as $NO_x = NO + NO_2$. For all SSI-based averages, the solar response of $NO_y$ is negative in the stratosphere and lower mesosphere but positive in the uppermost mesosphere above 0.03 hPa (Figure 5f). The stratospheric and

lower mesospheric decrease is consistent with Hood and Soukharev (2006) who attributed the negative solar response of $NO_y$ in the stratosphere at low latitudes to enhanced photolysis of nitric oxide (NO) by solar FUV irradiance during solar maximum. The major source of NO is the oxidation of nitrous oxide ($N_2O + O(^1D) \rightarrow 2NO$) in the middle stratosphere at low latitudes where the abundance of $O(^1D)$ is sufficiently high due to the photolysis of ozone at wavelengths < 310 nm (e.g., Seinfeld and Pandis, 2006). The major sinks of NO are photolysis ($NO + h\nu(183nm < \lambda < 193nm) \rightarrow N + O$) and the subsequent reac-

tion of NO with atomic nitrogen ($N + NO \rightarrow N_2 + O$) in the upper stratosphere, mesosphere and lower thermosphere (e.g., Minschwaner and Siskind, 1993). During solar maximum, $O(^1D)$ increases (Figure 5e) implying enhanced NO production from $N_2O$. However, at the same time a clear increase of the NO photolysis rates by 8 to 9% is found in the stratosphere and mesosphere at solar maximum (Figure 6f). This increase is the result of the larger solar irradiance amplitude in the FUV in all applied SSI data sets (see Table 1). With enhanced photolysis at solar maximum, the $NO_y$ abundances decrease (Figure 5f).

The $NO_y$ solar response is of the same magnitude up to the lower mesosphere for all SSI data sets, except for SATIRE-T, which due to its weaker FUV solar amplitude produces a weaker increase of the NO photolysis rates and a weaker negative $NO_y$ solar response, respectively. The negative $NO_y$ solar response results in a slowdown of the catalytic $NO_x$ cycle of ozone destruction, which indirectly enhances the positive solar response in ozone (Sukhodolov et al., 2016).

    There are two possible reasons for the increase of $NO_y$ in the uppermost mesosphere above 0.03 hPa during solar maximum

compared to solar minimum. EUV photoionization of neutrals increases during solar maximum, leading to an increase of electron, ion and excited species production and subsequently, NO, above about 80 km. On the other hand, the response of $NO_y$ in the uppermost mesosphere could also reflect changes in the auroral NO production in the lower thermosphere. NO in the lower thermosphere is mainly produced by the reaction of an excited nitrogen atom with molecular oxygen ($N(^2D) + O_2 \rightarrow NO + O$) (Marsh et al., 2004). As low auroral activity is prescribed for both the solar minimum reference and the solar maximum simu-

lations, no solar response of thermospheric NO is expected from this reaction. However, NO in the lower thermosphere is also formed by the reaction of the ground-state of N with molecular oxygen ($N(^4S) + O_2 \rightarrow NO + O$). This reaction is strongly temperature dependent and thus more effective in the warmer lower thermosphere during the solar maximum (e.g., Sinnhuber and Funke, 2020).

The solar response in $HO_x$ (defined as $OH + HO_2$) (Figure 5g) is positive throughout the middle atmosphere in all CCM-averaged SSI-simulations. In the stratosphere, $HO_x$ is mainly produced by reactions of $O(^1D)$ with $H_2O, CH_4$ or $H_2$ (e.g., $H_2O + O(^1D) \rightarrow 2OH$). Increasing abundance of $O(^1D)$ (Figure 5e) during solar maximum conditions is leading to a positive solar response in $HO_x$ mixing ratios (Figure 5g). A clear dependence of the $O(^1D)$ and $HO_x$ solar responses on the UV-SSI amplitude is found with the largest solar response in the simulations using the SATIRE-S SSI data set. This SSI-amplitude

dependence of the $HO_x$ solar response continues in the upper mesosphere where it is mainly produced by photolysis of water vapour at wavelengths in the Schumann-Runge bands and Lyman-$\alpha$.

    Despite the loss of $H_2O$ through reaction with the more abundant $O(^1D)$ during solar maximum, the $H_2O$ mixing ratios increase during solar maximum in the stratosphere which is most pronounced when the SATIRE-S SSI data set is used (Figure 5h). While this signal is not statistically significant for the complete CCM ensemble, there is a significant solar response in the

stratosphere for the individual simulations of WACCM. This positive solar response of $H_2O$ in the upper stratosphere and lower mesosphere has also been identified in HALOE satellite data by Remsberg et al. (2018). It is explained by chemical reaction of $CH_4$ and $H_2$ with OH, producing finally, after a reaction chain including photolytic reactions, $H_2O$ (Remsberg et al., 1984). The positive solar response of $H_2O$ in the lower stratosphere corresponds to the positive solar response of temperature in the TTL region, which increases the saturation vapour pressure during solar maximum and thus counteracts the water vapour

limiting freeze drying mechanism. This is in contrast to the results of Schieferdecker et al. (2015) who analysed water vapour from MIPAS and HALOE satellite instruments and found an anti-correlation of a slightly time shifted 11-year solar cycle proxy with lower stratospheric water vapour. In the upper mesosphere, the solar response in $H_2O$ is negative as a direct consequence of the stronger water vapour photolysis during solar maximum (Figure 5h).

    In summary, the analysis of CCM-averaged quantities has revealed a dependence of the solar responses on the SSI data

sets, with solar responses for most quantities showing a clear relation to the SSI amplitude. Whereas the solar responses are relatively close to each other in the stratosphere and lower mesosphere when using NRLSSI1, NRLSSI2, and CMIP6, clear differences appear for SATIRE-T, which shows the smallest solar responses for all analyzed variables. SATIRE-S produces an enhanced solar response for $HO_x$ and $H_2O$ but agrees well with NRLSSI1, NRLSSI2, and CMIP6 for the other variables. The differences in the SSI amplitude are responsible for 10% of the temperature, 30% of the ozone, and up to 40% of the SW

heating rate variability of the solar response in the stratosphere and lower mesosphere. An even larger part of the variability of the solar response in the stratosphere can be attributed to the SSI data set for $O(^3P)$ (60%) and $O(^1D)$ (70%) (see Figure S2 in the supplement). In the upper mesosphere, the choice of the SSI data set has the largest influence on the solar response variability of the SW heating rates (70%) for which the solar amplitude of the SSI data set in the FUV is the main driver.

[Figure 7 about here.]

## 5.2 Differences resolved by CCM

Large differences in the variability of the solar response explained by CCM differences have been identified by the ANOVA (Figure 3, right) in the upper mesosphere. The range of the solar responses between the EMAC and WACCM ensemble means over the SSI data sets, indicated by the shaded areas in figures1 5 and 6, also identifies the largest differences in the upper mesosphere.

[Figure 8 about here.]

The changes in the chemistry and dynamics over the 11-year solar cycle are superimposed on the climatological reference states of the CCMs, and are influenced by the climatological temperatures and abundances of photochemically active species. In particular the climatological temperature has a large effect on the chemistry in the middle atmosphere and lower thermosphere either directly on the chemical gas phase reaction rates or indirectly, as the temperature in the tropical lower stratosphere determines the abundance of $H_2O$ in the middle atmosphere and thereby also affects many chemical reactions. The climatological background state of a CCM is determined by a number of factors, ranging from the horizontal and vertical resolutions and the vertical model domain to the physical and chemical processes either resolved or parameterized by the models. In this section, we examine the uncertainty in the atmospheric solar response arising from the model specifications of the EMAC and WACCM models, regarded here as representatives for typical state-of-the-art CCMs. For this purpose, two ensembles of five simulations each for EMAC and WACCM have been constructed by averaging the simulations with different SSI data sets of each model to the respective model ensemble. To achieve an overall ensemble mean representative for a mean solar state, the reference solar minimum simulation in each model ensemble is weighted by a factor of five, to balance the five simulations with the SSI data sets at solar maximum in each model ensemble.

Figure 7 shows the deviations of the climatological annual mean temperature of EMAC and WACCM from the climatological temperatures of the ERA-5 reanalysis (Hersbach et al., 2018). The ERA-5 climatology includes 37 years from 1982 to 2018 centered around the year 2000. The temperature deviations of both CCMs relative to ERA-5 are largely statistically significant on the 95% level (all regions not masked by hatching), as estimated by a Student's t-test.

EMAC has a pronounced cold bias in the tropical upper troposphere/lower stratosphere region (UTLS), where WACCM simulates slightly higher temperatures than the ERA-5 climatology. In the upper stratosphere and upper mesosphere, both models show a cold bias over large regions, which is more pronounced in the mesosphere of EMAC. In the lower mesosphere both CCMs show higher temperatures compared to ERA-5. At SH high latitudes, temperatures in WACCM are too low in the stratosphere and too high in the mesosphere compared to the ERA-5 climatology (Figure 7).

Figure 8 presents a direct comparison between the annual mean climatologies of the EMAC and WACCM ensemble means in the middle atmosphere in terms of zonal mean differences in SW heating rates, temperature, ozone, atomic oxygen ($O(^3P)$, $O(^1D)$), $NO_y$, $HO_x$, and $H_2O$. Substantially lower (by up to 1.2 K day$^{-1}$) SW heating rates are found in the upper mesosphere of EMAC. This bias is indicative of less effective FUV-heating from oxygen absorption in the Schumann-Runge bands, presumably due to differences in the $O_2$ absorption parameterisation of Strobel (1978) with 19 bands (used in EMAC) and the heating rates from the photolysis parameterisation based on Koppers and Murtagh (1996) (used in WACCM). These dif-

ferences in SW parameterisations between the CCMs also affect the solar response in the upper mesosphere which is smaller in EMAC for the SW heating rates and temperatures (Figure 5a,b). They lead to the patterns of solar response variance explained by CCMs in Figure 3a,b (right) which resemble the structure of the anomaly pattern in SW heating rates in the upper mesosphere (Figure 8a). By contrast, in the lower mesosphere and large areas of the stratosphere, EMAC shows higher SW heating rates, possibly an effect of the degraded spectral resolution of the SW radiation scheme in WACCM below 60 km or of the higher ozone mixing ratios in EMAC (Figure 8c). However, WACCM shows larger solar responses in ozone mixing ratios and $JO_3 \rightarrow O(^3P)$ photolysis rates in the lower and middle stratosphere, possibly related to the finer spectral resolution of the photolysis scheme used in WACCM.

The temperature deviations in the mesosphere between EMAC and WACCM (Figure 8b) are not congruent with those in SW heating rates and, therefore, most likely have to be attributed to differences in dynamical heating by dissipating planetary or gravity waves. The largest temperature differences between EMAC and WACCM are located in high southern latitudes. The southern polar vortex is much colder in WACCM than in EMAC and the ERA-Interim reanalyses. This shows that with the implementation of the modified gravity wave parameterisation of Garcia et al. (2017) in our WACCM simulations, the low temperature bias has been alleviated but still exists. Interaction between dynamics and chemistry in EMAC is leading to higher ozone mixing ratios in the lower stratosphere over the south pole, where a less intense and warm biased south polar vortex is avoiding more severe heterogeneous ozone depletion, compared to WACCM.

Besides the differences in the average mesospheric SW heating rates and temperatures between EMAC and WACCM, there are also large differences in the chemical composition (and the associated solar responses) in this region. The odd oxygen mixing ratios in the upper mesosphere are lower in EMAC by 50% for ozone, 80% for $O(^3P)$, and more than 50% for $O(^1D)$ (Figure 8c-e). This is partly explained by the fact that oxygen photolysis in the Schumann-Runge continuum $(O_2 + h\nu(\lambda < 175.9\text{nm}) \rightarrow O(^1D) + O(^3P))$ is neglected in EMAC, because it becomes important only in the lower thermosphere, which is above the upper lid of EMAC. The larger abundances of $O(^1D)$ and $O(^3P)$ in WACCM, which has a higher upper lid than EMAC, are the result of photochemical production in the lower thermosphere and downward transport into the mesosphere by the residual circulation during the winter seasons, where they affect the climatological averages as well as the solar responses of atomic oxygen, $O_3$, and $HO_x$. The model differences in January and July exhibit a clear enhancement of $O(^1D)$ and $O(^3P)$ in the respective winter hemisphere of WACCM indicative for strong downward transport from the lower thermosphere (see Figure S3 and S4 in the supplement). As a result of the larger $O(^1D)$ and $O(^3P)$ mixing ratios, the equilibrium of ozone producing and destroying processes leads to higher ozone mixing ratios in WACCM, apparent in the climatology (Figure 8c) as well as in the solar response (Figure 5c) in the upper mesosphere. Due to the larger solar responses in the mixing ratios of $O(^1D)$ and $O(^3P)$ (Figure 5d,e), WACCM produces a strong ozone increase from solar minimum to maximum, in contrast to EMAC, where the more intense $HO_x$ cycle at solar maximum (Figure 5g) dominates and leads to a negative solar response. Noticeable are the higher ozone mixing ratios in the upper mesospheric high latitudes of EMAC (Figure 8c) compared to WACCM which result from a less effective catalytic $HO_x$ cycle of ozone destruction during the winter seasons, due to much lower $HO_x$ mixing ratios in EMAC in these regions.

As already discussed for atomic oxygen in the previous paragraph, the climatologies and solar responses of chemcial species in the upper mesosphere are strongly affected by differences in the vertical transport between WACCM and EMAC. With an upper boundary in the lower thermosphere, WACCM is capable to simulate the downward transport of $NO_y$ by the gravity wave driven, residual circulation from the thermosphere down to the lower mesosphere at high latitudes in the respective winter
seasons (figures1 S3 and S4 in the supplement). In EMAC we use the UBCNOX parameterisation to include $NO_y$ produced in the thermosphere by auroral and medium-energy electrons. Nevertheless, the $NO_y$ mixing ratios are up to 60% lower in EMAC than in WACCM in large parts of the upper mesosphere, except for northern polar latitudes where the WACCM $NO_y$ mixing ratios are exceeded by 100% (Figure 5f). The solar response of $NO_y$ in the upper mesosphere discussed in Sec. 5.1 is much stronger in WACCM than in EMAC, as obvious from the shaded area in Figure 2f. In WACCM, the increased NO production
during solar-maximum in the uppermost mesosphere and lower thermosphere is driven by the increase in EUV photoionization and lower thermospheric temperatures as discussed in Sec. 5.1. In EMAC, the $NO_y$ mixing ratios in the uppermost four model levels are determined by the UBCNOX parameterisation, which depends on the prescribed, constantly low Ap index, but not on the solar UV/EUV radiation, and thus suppresses a solar response in $NO_y$.

    Differences between EMAC and WACCM are also found in the climatological distribution of water vapour and the related
$HO_x$ mixing ratio. In the annual mean, EMAC has less $H_2O$ than WACCM in the stratosphere and lower mesosphere – except for the upper mesosphere, where EMAC exceeds the WACCM values (Figure 8h). The lower $H_2O$ abundance of EMAC in the middle atmosphere is the consequence of a cold bias in the tropical UTLS region (Figure 8b), a feature of EMAC also discussed in Jöckel et al. (2016). As a result, also $HO_x$, which is produced by photolysis of $H_2O$, is lower in EMAC (Figure 8g). From solar minimum to solar maximum, the abundances of $H_2O$ (Figure 5h) and the photolysis rates of $H_2O$ (Figure 6e) increase
in the stratosphere and lower mesosphere, leading to a $HO_x$ increase by about 2–3% in both models (see also Section 5.1). The smaller solar response of $HO_x$ in EMAC in this height region can be attributed to differences in the model climatologies because of the lower $HO_x$ and $H_2O$ mixing ratios in EMAC. In the mesosphere, the magnitude of the solar responses of $H_2O$ (negative) and $HO_x$ (positive) grow fast with altitude. At solar maximum, strongly enhanced $H_2O$ photolysis (Figure 6e) induces a decrease in $H_2O$ (Figure 5h) and an increase in $HO_x$ by about 11% (Figure 5g). The $HO_x$ production in the
upper mesosphere is dominated by $H_2O$ photolysis at wavelengths in the Schumann-Runge bands and Lyman-$\alpha$ with the major reaction $H_2O + h\nu(\lambda < 200) \rightarrow H + OH$ (70%, $J^a_{H_2O}$) and two minor reactions, producing $O + 2H$ (12%, $J^b_{H_2O}$) and $H_2 + O(^1D)$ (10%, $J^c_{H_2O}$) (Nicolet, 1984). However, while the solar responses in $H_2O$ photolysis (positive) and $H_2O$ mixing ratio (negative) further increase towards the upper mesosphere, the positive solar response in $HO_x$ peaks near 70 km and declines above, implying the existence of a $HO_x$ depleting process that increases with solar activity in the upper mesosphere
and counteracts $HO_x$ production by $H_2O$ photolysis. In addition $H_2O$ declines in the upper mesosphere, limiting the potential for $HO_x$ production. The CCMs start to deviate more strongly in the upper mesosphere, with EMAC showing a less intense decrease of the solar response in $HO_x$ than WACCM. This cannot be attributed to the $H_2O$ photolysis as EMAC has a larger solar response in $H_2O$ photolysis. The reason might be the combination of the up to 50% larger upper mesospheric $H_2O$ mixing ratio in EMAC (Figure 8h) and the stronger solar response of WACCM in $O(^3P)$ (Figure 5d) which acts in WACCM
as a more effective sink of $HO_x$ through reaction $OH + O(^3P) \rightarrow H + O_2$. The smaller $H_2O$ mixing ratio of WACCM in the

upper mesosphere is a consequence of the larger $O(^1D)$ mixing ratios in WACCM, leading to a more effective decomposition of water vapour through $H_2O + O(^1D) \rightarrow 2\,OH$. During solar maximum this $H_2O$ decomposition is even enhanced, due to more abundant $O(^1D)$ mixing ratios in the upper mesosphere (Fig. 5e), leading to a more pronounced negative $H_2O$ solar responsein WACCM (Figure 5h).

In summary, both models show for all quantities comparable solar responses to the different SSI data sets up to the lower mesosphere near 0.1 hPa. Above, the solar responses deviate substantially, as shown for ozone, oxygen, $HO_x$, and $NO_y$. The comparison of the upper mesospheric solar responses in EMAC and WACCM, as well as the differences in the climatologies in the upper mesosphere, shows that a realistic simulation of solar cycle effects might better achieved when the residual downward transport of thermospheric photolysis reactants is taken into account.

## 10    6    Solar response in total ozone

In this section we focus on the solar response in total column ozone (TCO) in our simulations and investigate to what extent TCO is influenced by the applied SSI data set and CCM. The solar response in TCO is the vertically integrated solar response in ozone mixing ratios. In the previous sections, we have identified differences in the solar response in ozone mixing ratios depending on the applied SSI data set (Section 5.1) as well as on the applied CCM (Section 5.2), especially in the middle to
lower stratosphere which contributes most to the solar response in TCO.

Hood (1997) who analysed SBUV (Solar Backscatter Ultraviolet) Radiometer data for a relatively short period from 1979–1993, found an annual mean solar response of TCO in the tropics of 1–2%. For the same time period, Zerefos et al. (1997) detected a significant correlation of the solar activity with annual mean TCO from TOMS between 40°S and 40°N, whereas no significant correlation was found at higher latitudes, due to the large dynamically induced variability. Soukharev and Hood
(2006) suggested that the solar response of TCO is mainly caused by ozone abundances in the tropical lower stratosphere. A study of Randel and Wu (2007) identified a significant annual mean solar response in TCO between 40°S and 60°N in TOMS/SBUV data from 1979–2005.

[Figure 9 about here.]

[Figure 10 about here.]

The latitudinal distribution of the solar response in TCO for the 10 simulations performed for this study is shown in Figure 9. Clear differences occur between both CCMs, with WACCM showing in general a larger solar response in TCO at all latitudes. Between 40°S and 40°N significant solar TCO responses are simulated in both models, reaching 1.5% (3.9 DU) for WACCM and 1.1% (3.0 DU) for EMAC in the tropics (20°S–20°N). The chosen SSI data set has only a very minor impact on the solar TCO signal in both models, except for SATIRE-T. The WACCM and EMAC simulations using SATIRE-T show the smallest
solar responses in the tropics, which reflects the small solar amplitude in the UV spectral region of this SSI data set. At mid and high latitudes, differences in the solar TCO signals between the SSI data sets become larger, however remain generally smaller than the differences between the models, as particular evident for high southern latitudes.

The solar response in TCO at high latitudes is strongly influenced by stratospheric dynamic variability during the respective winter seasons in both hemispheres. As a measure of this dynamic variability, we introduce in Figure 10 the thickness of the stratospheric layer between the 100 and 10 hPa pressure levels. High values of the 100–10 hPa layer thickness correspond to a warm layer and a weak polar vortex occurring in dynamically disturbed periods, such as minor or major sudden stratospheric warmings (SSWs). The associated downward transport of air in high latitudes leads to an increase in TCO. On the other hand, cold conditions in polar regions, represented by low values of 100–10 hPa layer thickness, can lead to enhanced chemical ozone depletion and low TCO values (Farman et al., 1985; Rex et al., 2002; Manney et al., 2011). While the 100–10 hPa layer thicknesses for all 10 simulations show comparable and significantly increased values at solar maximum between 40°S and 40°N – consistent with the warming in Figure 2b –, systematic differences occur between the two models at southern high latitudes where EMAC simulates a colder and stronger polar vortex during solar maximum and WACCM shows a warmer and weaker polar vortex (Figure 10). At northern polar latitudes, both models simulate warmer and weaker polar vortices at solar maximum - except for the SATIRE-S WACCM simulation.

The high correlation between the solar responses in TCO and 100–10 hPa layer thickness at high latitudes is reflected in Table 2 which gives the correlation coefficients between the solar maximum anomalies of the annual average 100–10 hPa layer thickness in the polar region (70°–90°) and the respective TCO anomalies for both hemispheres. The annual mean correlation at high latitudes is mainly a result of the high correlation during the winter/spring seasons of both hemispheres, with the highest correlation occurring in January/February/March (JFM) for the NH and in September/October/November (SON) for the SH (for seasonal correlation coefficients see Table S1 in the supplement). More intense downwelling of ozone during episodes of stratospheric warming events, or less intense downwelling during cold conditions in combination with a more effective chemical ozone depletion, are the main driver of the TCO anomalies in high latitudes in both hemispheres. Although all anomalies shown in Figures 9 and 10 are derived for simulations during solar maximum conditions with respect to the simulation during solar minimum conditions, a consistent, significant solar response for TCO and lower stratospheric layer thickness in polar latitudes can not be found, implying that the derived changes at high latitudes are rather due to random internal dynamical variability in the models than to the external solar forcing.

In Figure 11 (top) we show the zonally averaged differences in TCO between the simulations for solar maximum and solar minimum conditions for the average of all simulations (black contour), the EMAC- (blue contour) and WACCM-simulations (red contour), the standard deviation (light grey shading), and the two-way ANOVA with SSI- (dark grey shading) and CCM-treatment (hatches). The range of the standard deviation shows the largest variability of the TCO anomaly in polar regions. As discussed above, the high latitude TCO anomaly is mainly a result of the internal, dynamic variability of the CCMs. Consequently in the north polar region, there is only a relatively small part of the TCO anomaly variability that can be explained by the SSI data set (1%) or the CCM (0.6%). Only in the southern polar region, a larger but still only small fraction of the variability can be explained by differences in the CCMs (1.4%), as there are systematic differences in the anomalies of the lower stratospheric layer thickness and TCO between WACCM and EMAC (Figures 9, 10). As in the northern polar region, the SSI data set can only explain a small fraction of the variability of the TCO anomaly in the southern polar region (0.8%).

The standard deviation of the solar TCO response is considerably smaller in the tropics, where also the fraction of the standard deviation that can be explained by either the choice of the SSI data set (22%) or the CCM (13%) is much larger, with a larger contribution of the SSI data sets. When approaching the northern mid-latitudes the influence of the CCMs on the variability of the solar response grows, whereas the influence of the SSI data set gets smaller. This is reflected by the pronounced minimum of the solar TCO response in the mid-latitudes of the EMAC simulations, whereas three out of five WACCM simulations show a relatively large positive anomaly in this region.

The annual, global mean solar response of partial column ozone integrated for different pressure regions is shown in Figure 11 (bottom) with the same type of contour lines/shading as in Figure 11 (top). Averaged over all simulations, the largest solar response in ozone is in the lower stratosphere. In this region, simulations performed by WACCM show a much larger solar response than the averaged EMAC simulations, with largest differences occurring in the lowest region with pressures higher than 32 hPa. In the WACCM simulations, this layer contributes 32.5% to the solar response in TCO, whereas it is only 16.7% in the EMAC simulations. The analysis of SBUV-SBUV/2 data by Hood (1997) shows that 85% of the solar response in TCO is from the contribution of the lower stratospheric layer (pressure > 16 hPa). The same analysis for the CCMs gives smaller contributions from the lower stratosphere with 52.8% on average, 57.8% for WACCM, and 45.3% for EMAC. This different behaviour of the CCMs in the lower stratosphere is also reflected in the relatively large fraction of the standard deviation of the solar response that can be explained by differences between the CCMs. A larger part of the standard deviation can be explained by differences between the SSI data sets in the middle stratosphere (16–4 hPa), although still the largest part of the standard deviation can not be attributed to either differences between the CCMs or the SSI data sets.

[Figure 11 about here.]

## 7  Summary and conclusions

This study aimed at investigating the uncertainty in simulations of the atmospheric solar response to 11-year solar cycle variability. In particular, the effects of two sources for uncertainty, i.e. the prescribed spectral solar irradiance (SSI) data set and potential differences in the CCM background state and configuration, were examined. For this purpose, simulations with two CCMs, each forced with five different SSI data sets, were performed. The CCMs EMAC and WACCM are representative of state-of-the-art CCMs, including the prerequisites for simulating solar cycle variability, i.e. spectrally resolved SW radiation and photolysis schemes. Both models contributed to the evaluation of the simulated solar signature in the SPARC CCMVal-2 initiative (SPARC CCMVal, 2010). The SSI data sets represent the best available estimates of 11-year solar variability in electromagnetic radiation, and either have been used in previous model studies (i.e., CMIP5 or CCMVal-2) or are recommended for current model intercomparions (i.e., CMIP6). We apply a novel approach to extract the effects of the pure SSI solar cycle amplitudes, as a common reference SSI distribution for solar minimum conditions was defined, based on ATLAS3 measurements in November 1994, and SSI data sets for solar maximum were constructed by adding the solar amplitude of the five SSI data sets to the solar minimum reference distribution. To separate the influences of the SSI data sets and the CCMs, respectively, on

the solar responses in SW heating rates, temperature and ozone, a two-way analysis of variance (ANOVA) was applied here for the first time in this context.

Our study revealed that differences in SSI data sets provide the largest fraction of solar cycle variance in the upper stratosphere/lower mesosphere, contributing 40% to the SW heating rate, 30% to the ozone and 10% to the temperature solar cycle variance. A second region with a considerable SSI induced spread in the SW heating rate solar response up to 70% is the upper mesosphere (except for polar latitudes), affecting also the ozone solar response in this height region. In the region of largest ozone mixing ratio, in the stratosphere from 50 to 10 hPa, the SSI data sets do not contribute much to the variability of the solar response when SATIRE-T is omitted. Differences between CCMs have a major effect in the upper mesosphere, where they explain more than 80% of the ozone, 25% of the temperature and 50% of the SW heating rate variability of the solar responses. CCMs add a minor contribution to the SW heating rate and ozone solar cycle variance in the lower stratosphere. However, in the upper stratosphere/lower mesosphere the largest fraction of solar cycle response variance is random and not related to differences in SSI data sets or the applied CCM.

To isolate the causes for the contributions to the solar response spread, a detailed analysis was performed of the solar response profiles of SW heating rate, temperature, ozone, different chemical compounds and photolysis rates for the five SSI-simulation ensemble means (each including 2 CCMs) on the one hand and for the two CCM ensemble means (each including 5 SSI data sets) on the other. The analysis of CCM-averaged quantities, involved in the radiative and photochemical processes, has revealed a dependence of the solar responses on the SSI data sets, with solar responses for most quantities showing a clear relation to the SSI amplitude. Whereas the solar responses are relatively close to each other in the stratosphere and lower mesosphere when using NRLSSI1, NRLSSI2, and CMIP6, distinct differences appear for SATIRE-T, which shows the smallest solar responses for all analyzed variables. Weaker solar responses in temperature can be explained by a reduced solar cycle amplitude in the 201–242 nm spectral irradiance range, mainly responsible for solar radiative heating in this height range. The ozone increase at solar maximum in the middle and upper stratosphere is the combined result of enhanced oxygen chemistry in the photochemical Chapman cycle and reduced ozone destruction in the $NO_y$ catalytic cycle. The NRLSSI1, NRLSSI2, and CMIP6 data sets produce similar solar responses in $O(^1D)$, $O(^3P)$ as well $NO_y$ at solar maximum in the stratosphere. The solar ozone signal is considerably weaker in the SATIRE-T data set, as its SSI amplitude in the FUV and 201–242 nm spectral ranges, important for photochemical ozone production, is considerably weaker than in the other SSI data sets, whereas its SSI amplitude in the ozone-destroying UV-band (243–380 nm) is comparable. Positive solar responses were also derived for $HO_x$ and $H_2O$ throughout the middle atmosphere for all SSI data sets. In the stratosphere, where $HO_x$ is mainly produced by reactions of $O(^1D)$ with $H_2O$, $CH_4$ or $H_2$, the increasing abundance of $O(^1D)$ during solar maximum conditions is leading to a positive solar response in $HO_x$ mixing ratios. A clear dependence of the $O(^1D)$ and $HO_x$ solar responses on the UV-SSI amplitude was found with the largest solar response in the simulations using the SATIRE-S SSI data set. The increase in $H_2O$ mixing ratios with the solar cycle throughout the stratosphere, emerging for all SSI data sets, can be explained by chemical production in the upper stratosphere and enhanced $H_2O$ transport from the troposphere into the lower stratosphere in a warmer UTLS at solar maximum. In the upper mesosphere, the choice of the SSI data set has the largest influence on the solar response variability of the SW heating rates (70%) for which the solar amplitude of the SSI data set in the FUV is the main driver.

In addition to the SSI induced spread in the solar response, differences between the CCMs turned out to have their strongest impact on the solar responses in SW heating rates, temperature and ozone in the upper mesosphere at altitudes above about 60 km. The two CCMs used for this study are representative for the current CCM generation which consists primarily of models with a top level in the upper mesosphere around 0.01 hPa (or 80 km altitude), as the EMAC version used in this study,
plus a few CCMs that also include the thermosphere, as WACCM with a top level at about 140 km. Resulting differences in the radiation parameterisations and the vertical transport of substances have been identified to cause the spread in the upper mesosphere solar responses. For example, the spread in the solar response of the upper mesosphere SW heating rates arises from an underestimation of oxygen absorption in the FUV by the parameterisation of Strobel (1978) in EMAC, reaching up to -1.2 K day$^{-1}$ compared to WACCM in the climatological annual mean. Moreover, the odd oxygen mixing ratios in the
upper mesosphere are substantially lower in EMAC than in WACCM, as EMAC – due to its lower lid – does not capture the photochemical production of $O(^1D)$ and $O(^3P)$ in the lower thermosphere of WACCM nor their downward transport into the mesosphere by the residual circulation during the winter seasons. As a result, solar ozone responses of opposite sign are produced in the upper stratosphere where WACCM exhibits more ozone at solar maximum due to its enhanced odd oxygen abundances, while EMAC exhibits less ozone due to its more intense $HO_x$ cycle. This shows that a realistic simulation of
solar cycle effects is better achieved when the residual downward transport of thermospheric photolysis reactants is taken into account. Some of these effects could be included in CCMs with a model top in the upper mesosphere by a thoroughly formulated upper boundary condition, as already included for $NO_y$ produced in the thermosphere by auroral and medium-energy electrons.

For annual mean total column ozone (TCO) a significant solar response could be identified from the southern midlatitudes
to the northern polar region when all simulations from both CCMs are considered. In the southern high latitudes, these TCO anomalies are the result of natural dynamical variability of the Antarctic polar vortex. Distinct differences in TCO anomalies between the CCMs are also expressed by the relatively large fraction of the anomaly variability that can be explained by differences between the CCMs. The usage of the SSI data set has the largest influence on the variability of the TCO solar response in the tropics. The largest contribution to the annual mean TCO solar response is from the lower stratospheric layers
with pressures > 16 hPa which on average contributes 53%, or when analysed separately, 58% (WACCM) and 45% (EMAC). Both CCMs underestimate the lower stratospheric contribution to the solar response in TCO, compared to the analysis of SBUV-SBUV/2 data by Hood (1997) who found a 85% contribution of the lower stratosphere to the TCO solar response.

Note that the individual contributions of the SSI data sets and CCM configurations derived in our study are constrained by the choice of the CCMs. While the possible spread of solar cycle SSI variation is very well captured in our study by considering
the five currently usable SSI data sets, a similar coverage of the CCM induced spread cannot be achieved, given the number and diversity of available CCMs. Thus, the CCM contribution to the variance of the solar response are, to some extent, determined by the specifics of the EMAC and WACCM models. For example, the CCM contribution to the solar SW heating rate signal would increase in models applying SW radiation schemes with low spectral resolution or employing TSI scaling procedures.

The usage of an alternative SSI reference data set such as (Meftah et al., 2020, 2018), which is representative of the 2008
solar minimum, may have an influence on the climatological state of the CCMs, as higher or lower SSI values in certain

spectral bands lead to higher or lower SW heating rates, thus affecting the temperature and potentially also the zonal wind of the CCMs. As the reference SSI data set serves as a common base state for the solar minimum of all other SSI data sets, we do not expect significant differences in the uncertainties of the solar responses when using a different SSI reference spectrum. The quantification of the uncertainties of the solar responses in the CCMs is based on only one exemplary solar amplitude

(descending phase of solar cycle 22, November 1989 to November 1994). However, other solar cycles show different, spectrally resolved characteristics in their amplitudes, which also differ among the individual SSI data sets. From comparison to other solar cycle amplitudes we found the magnitude of the applied solar amplitude in this study to be representative for solar cycle amplitudes in the satellite era.

Finally, as all simulations of this study were carried out under conditions of low auroral activity, only effects of 11-year

variations in solar electromagnetic radiation have been considered in this study. The impact of variations of energetic particle precipitation for different levels of auroral activity will be subject of further studies.

## 8  Data availability

The data of the EMAC and CESM1(WACCM) simulations, which have been performed for this study, are available for download on request.

**Appendix A:  Analysis of variance (ANOVA)**

A two-way analysis of variance (ANOVA) (Fisher, 1925) is applied to the time series of anomaly data $x_{ijk}$, i.e. a data set consisting of the time series of the differences between the simulations performed by both CCMs ($i = 1, 2$) for solar maximum conditions of five SSI data sets ($j = 1, \ldots, 5$) and the respective simulation of each CCM for solar minimum conditions. It is created by using the five simulations of each CCM with solar maximum SSI data sets minus the simulations of each CCM with

the ATLAS-3 SSI data set at solar minimum. Thus, the complete time series consists of 450 annual mean anomalies ($n_t$) from 10 simulations, each with a length of 45 years ($n$). The anomalies $x_{ijk}$ are the individual solar responses, and their fluctuations around the averaged solar response $\overline{x}$ can be described as

$$x_{ijk} = \overline{x} + (\overline{x_i} - \overline{x}) + (\overline{x_j} - \overline{x}) + (\overline{x_{ij}} - \overline{x_i} - \overline{x_j} + \overline{x}) + \epsilon_{ijk}$$
$$= \overline{x} + \alpha_i + \beta_j + \gamma_{ij} + \epsilon_{ijk} \tag{A1}$$

with $\alpha_i$ the deviations of the averages of the two CCMs from the overall average $\overline{x}$; $\beta_j$ the deviations of the averages of simulations applying the five SSI data sets from $\overline{x}$; $\gamma_{ij}$ the deviations of the averages of the individual simulations $\overline{x_{ij}}$, $\overline{x_i}$, and $\overline{x_j}$ from $\overline{x}$; and $\epsilon_{ijk}$ the random fluctuations that cannot be explained by $\alpha_i$, $\beta_j$, or $\gamma_{ij}$. The null hypotheses ($H_0$) are that the averaged solar response does not depend on the CCM ($H_{0,1} : \alpha_i = 0$), the applied SSI data set ($H_{0,2} : \beta_j = 0$), or the interaction of CCM with applied SSI data set ($H_{0,3} : \gamma_{ij} = 0$). We apply the two-way ANOVA to test the validity of these hypotheses. The

total sum of squares ($SS_t$) of the complete time series is calculated as

$$SS_t = \sum_{l=1}^{n_t}(x_l - \bar{x})^2 = \sum_{i=1}^{N_A}\sum_{j=1}^{N_B}\sum_{k=1}^{n}(x_{ijk} - \bar{x})^2 \tag{A2}$$

with the individual annual mean solar response $x_{ijk}$ and the overall mean $\bar{x}$. The $SS_t$ of the complete time series is further split by applying two treatments as

$$SS_t = SS_{bA} + SS_{bB} + SS_{bAB} + SS_w. \tag{A3}$$

One treatment (A) takes into consideration the applied CCM, building two groups ($N_A = 2$), the WACCM and the EMAC solar responses with $n_A = 225$ elements each. The second treatment (B) takes into consideration the applied SSI data set, building five groups ($N_B = 5$) with $n_B = 90$ elements each. For each treatment ($K = A, B$) the sum of squares between the groups ($SS_{bK}$) are calculated as

$$SS_{bA} = n_A \sum_{i=1}^{N_A}(\bar{x_i} - \bar{x})^2, \tag{A4}$$

$$SS_{bB} = n_B \sum_{j=1}^{N_B}(\bar{x_j} - \bar{x})^2, \tag{A5}$$

with $\bar{x_i}, \bar{x_j}$ the mean solar response of each group, $N_A$, $N_B$ the number of groups within each treatment, and $n_A$, $n_B$ the number of elements within each group. The sum of squares emerging by the interaction of the treatments ($SS_{bAB}$) is calculated as

$$SS_{bAB} = n \sum_{i=1}^{N_A}\sum_{j=1}^{N_B}(\bar{x_{ij}} - \bar{x_i} - \bar{x_j} + \bar{x})^2, \tag{A6}$$

with $\bar{x_{ij}}$ the mean of the individual simulations and $\bar{x_i}, \bar{x_j}$ the mean solar responses of each group within the treatments. The sum of squares within ($SS_w$), which accounts for the random, unexplained or error part of the variability, is calculated as

$$SS_w = \sum_{i=1}^{N_A}\sum_{j=1}^{N_B}\sum_{k=1}^{n}(x_{ijk} - \bar{x_{ij}})^2, \tag{A7}$$

with $x_{ijk}$ the individual solar responses. The degrees of freedom of the model are calculated as

   $$df_d = n_t - N_A N_B \tag{A8}$$

$$df_{nK} = N_K - 1 \tag{A9}$$

$$df_{nAB} = (N_A - 1)(N_B - 1). \tag{A10}$$

These are the degrees of freedom within the groups ($df_d$) which is the degrees of freedom of the denominators when calculating the F-statistics (Equation A15), $df_{nK}$ the degrees of freedom between the groups, and $df_{nAB}$ the degrees of freedom of the interaction between the treatments which are the degrees of freedom of the nominators in Equation A15. Note that the CMIP6

SSI data set depends on the SATIRE-S and NRLSSI2 data sets and therefore cannot be counted as an independent SSI data set. For the calculations of the degrees of freedom we use $N_B = 4$. By the ratios

$$R_{a,K}^2 = \frac{SS_{bK} - \frac{df_{nK}}{df_d}SS_w}{SS_t}, \tag{A11}$$

$$R_{a,AB}^2 = \frac{SS_{bAB} - \frac{df_{nAB}}{df_d}SS_w}{SS_t} \tag{A12}$$

the adapted coefficients of determination (von Storch and Zwiers, 1999) are calculated, which are a measure of the variance explained by the treatment (K = A, B) or by the interaction between both treatments. $R_{a,A}^2$, $R_{a,B}^2$, and $R_{a,AB}^2$ are shown in Figure 4. The mean sum of squares within the groups $\text{MSS}_w$ is calculated as

$$MSS_w = \frac{SS_w}{df_d}, \tag{A13}$$

and the mean sum of squares between the groups $\text{MSS}_{bK}$ and the MSS interacting between the groups $\text{MSS}_{bAB}$ are calculated as

$$MSS_{bK} = \frac{SS_{bK}}{df_{nK}}, \quad MSS_{bAB} = \frac{SS_{bAB}}{df_{nAB}}. \tag{A14}$$

As $\text{MSS}_w$, $\text{MSS}_{bK}$, and $\text{MSS}_{bAB}$ are assumed to be unbiased estimators of the variance $\sigma^2$ we can use these estimators to calculate the F-statistics as

$$F = \frac{MSS_{bK}}{MSS_w}, \quad F = \frac{MSS_{bAB}}{MSS_w}. \tag{A15}$$

*Author contributions.* MK wrote the manuscript, performed the EMAC simulations, and the data analyses; TK performed the WACCM simulations and contributed to the manuscript; UL initiated the study, acquired the BMBF funding, and participated in writing the manuscript; KM, and MS initiated the study, acquired the BMBF funding, and contributed to the manuscript; TR contributed to the manuscript and the EMAC model developement.

*Competing interests.* The authors declare that they have no competing interests.

*Acknowledgements.* This study is supported by the German Ministry of Research (BMBF) within the nationally funded project ROMIC-SOLIC (grant number 01LG1219). Markus Kunze acknowledges support by the Deutsche Forschungsgemeinschaft (DFG) through grant KU 3632/2-1. The EMAC simulations have been performed on the massive parallel supercomputing system of the North-German Supercomputing Alliance (HLRN). All WACCM simulations have been performed on the high performance computing facilities of the Kiel University (Christian-Albrechts-Universität zu Kiel).

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

## List of Figures

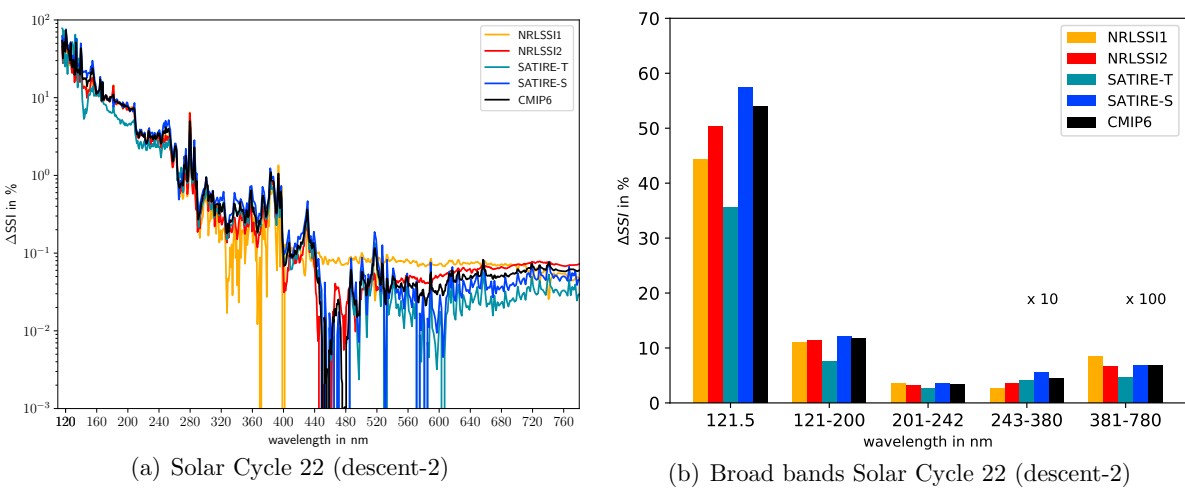

(a) Solar Cycle 22 (descent-2)

(b) Broad bands Solar Cycle 22 (descent-2)

**Figure 1.** (a) Solar cycle SSI variations from Nov. 1989 to Nov. 1994 relative to Nov. 1994 ($\Delta SSI$) in % for wavelengths ranging from FUV to the VIS bands; (b) as in (a) for the Lyman-$\alpha$ (121.5 nm), Far-UV (121–200 nm), Herzberg continuum/Hartley bands (201–242 nm), Hartley-/Huggings-bands (243–380 nm) (multiplied by a factor of 10) and visible (381–780 nm) (multiplied by a factor of 100).

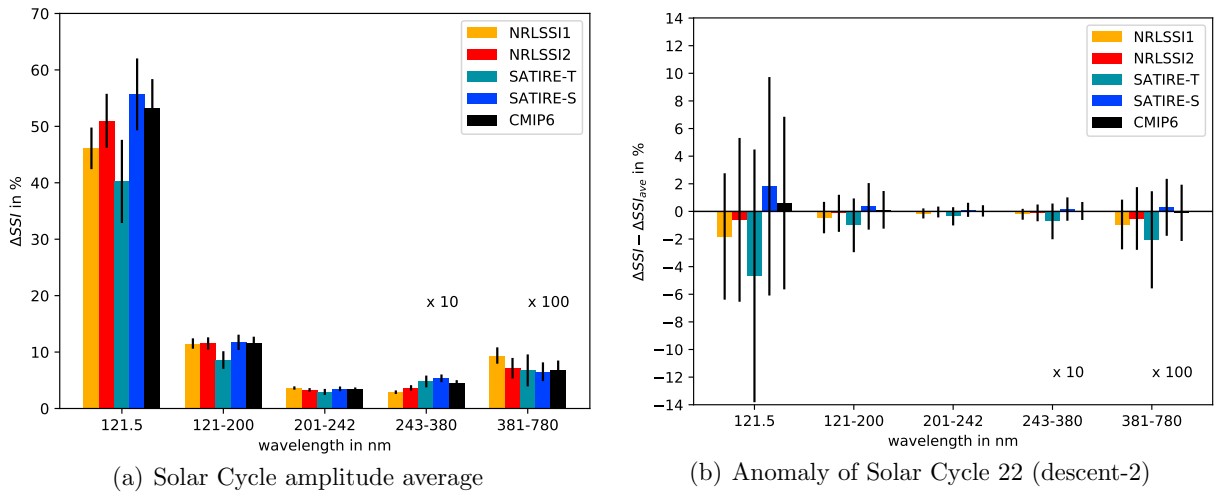

(a) Solar Cycle amplitude average

(b) Anomaly of Solar Cycle 22 (descent-2)

**Figure 2.** (a) Averaged SSI variations for solar cycle amplitudes relative to ATLAS3 ($\Delta SSI$) in % for the Lyman-$\alpha$ (121.5 nm), Far-UV (121–200 nm), Herzberg continuum/Hartley bands (201–242 nm), Hartley-/Huggings-bands (243–380 nm) (multiplied by a factor of 10) and visible (381–780 nm) (multiplied by a factor of 100) spectral ranges; with the 95% confidence interval given as error bar. (b) Anomaly of SSI variations for solar cycle 22 (descent-2) with respect to the averaged solar cycle shown in (a).

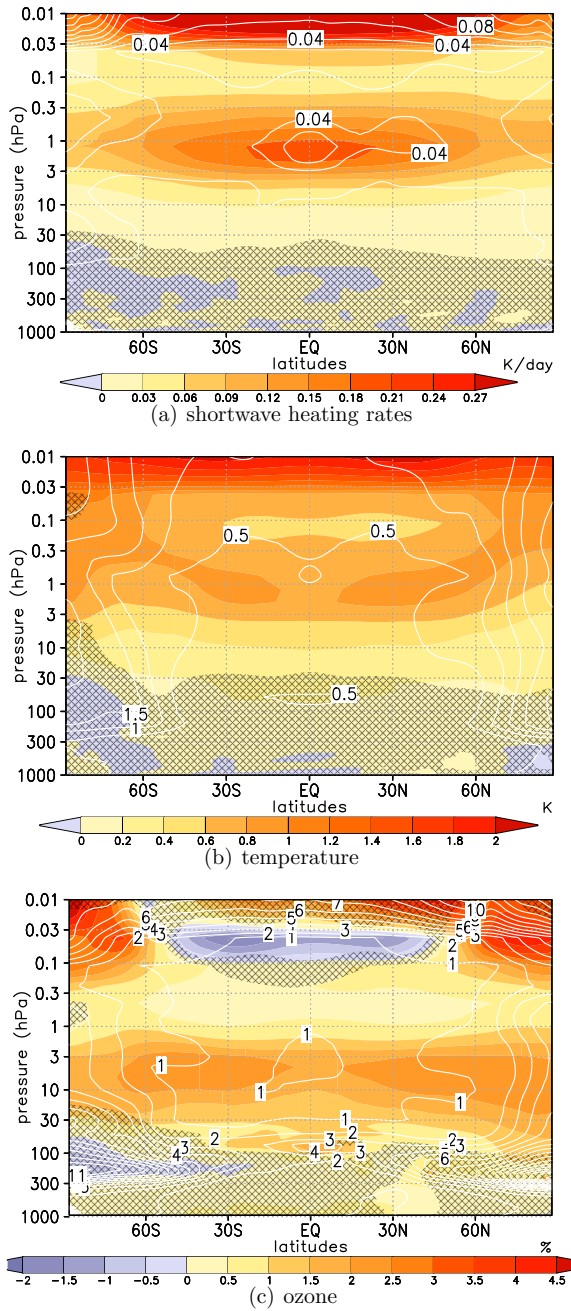

**Figure 3.** Annual mean 11–year solar cycle response (shaded) and signal variance (white contours) in terms of the solar response annual standard deviation for SW heating-rates (a), temperature (b), and ozone mixing ratios (c). Solar response derived as ensemble mean over both models and all SSI data sets; solar minimum SSI based on ATLAS3 reference state. The grey hatching masks areas where signal or ratio of explained variance does not pass a test for statistical significance (p > 5 %).

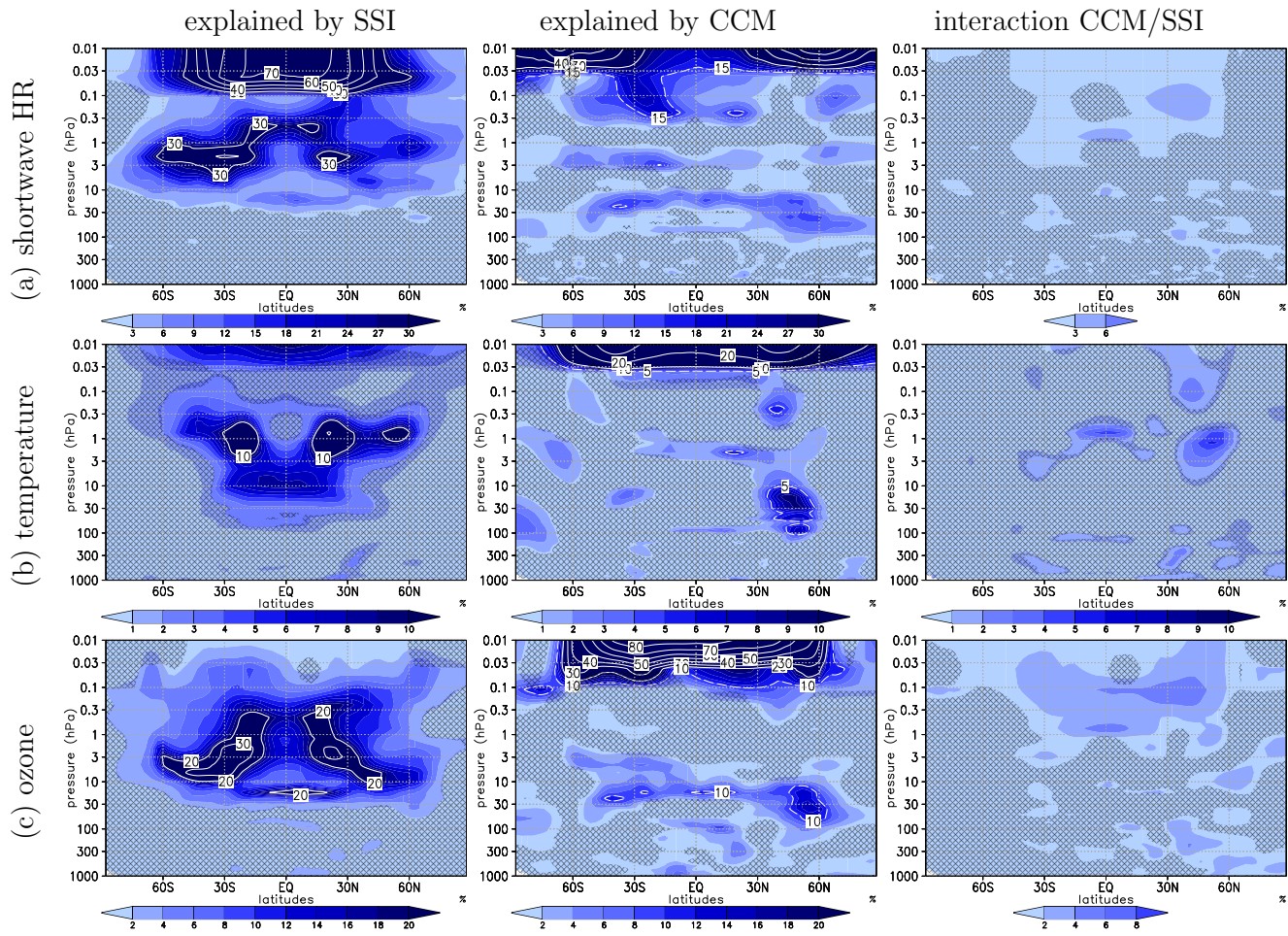

**Figure 4.** Left column: Percentage of signal variance (square of white contours of left figures) explained by systematic differences between forcing SSI data sets ($R_{a,B}$ Equation A11, blue shading). The white contours indicate levels of explained variance larger than the range of shading. Middle column: as left column but for systematic differences between CCMs ($R_{a,A}$ Equation A11). Right column: as left column but for signal variance explained by the interaction of the CCM and the SSI data set treatments ($R_{a,AB}$ Equation A12). The grey hatching masks areas where the ratio of explained variance does not pass a test for statistical significance ($p > 5$ %).

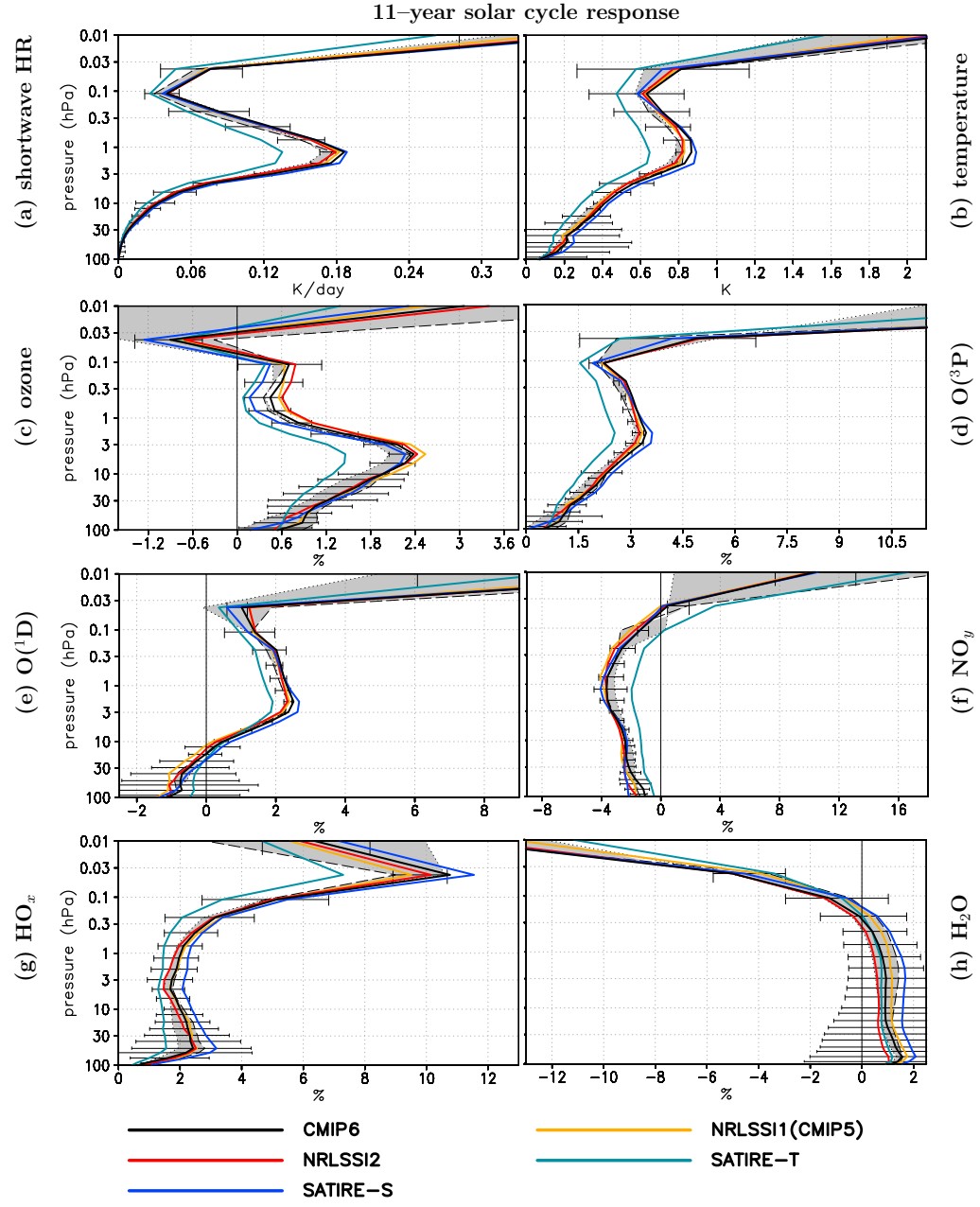

**Figure 5.** Annual mean 11–year solar cycle response (60°S–60°N) in (a) SW heating rates, (b) temperature, (c) ozone concentrations, (d) atomic oxygen (O($^3$P)), (e) atomic oxygen (O($^1$D)), (f) NO$_y$, (g) HO$_x$, and (h) H$_2$O. Solar responses are derived for an average of WACCM and EMAC simulations using five SSI data sets at solar maximum NRLSSI1(CMIP5) (yellow), NRLSSI2 (red), SATIRE-S (blue), SATIRE-T (dark blue), and CMIP6 (black) relative to the average of the WACCM and EMAC reference solar minimum simulations. The shaded area indicates the range of the WACCM (black long dash contour) and EMAC (black dotted contour) ensemble means. The 95% uncertainty error bar is given for the averaged solar response over the complete ensemble, calculated with a Student's t test.

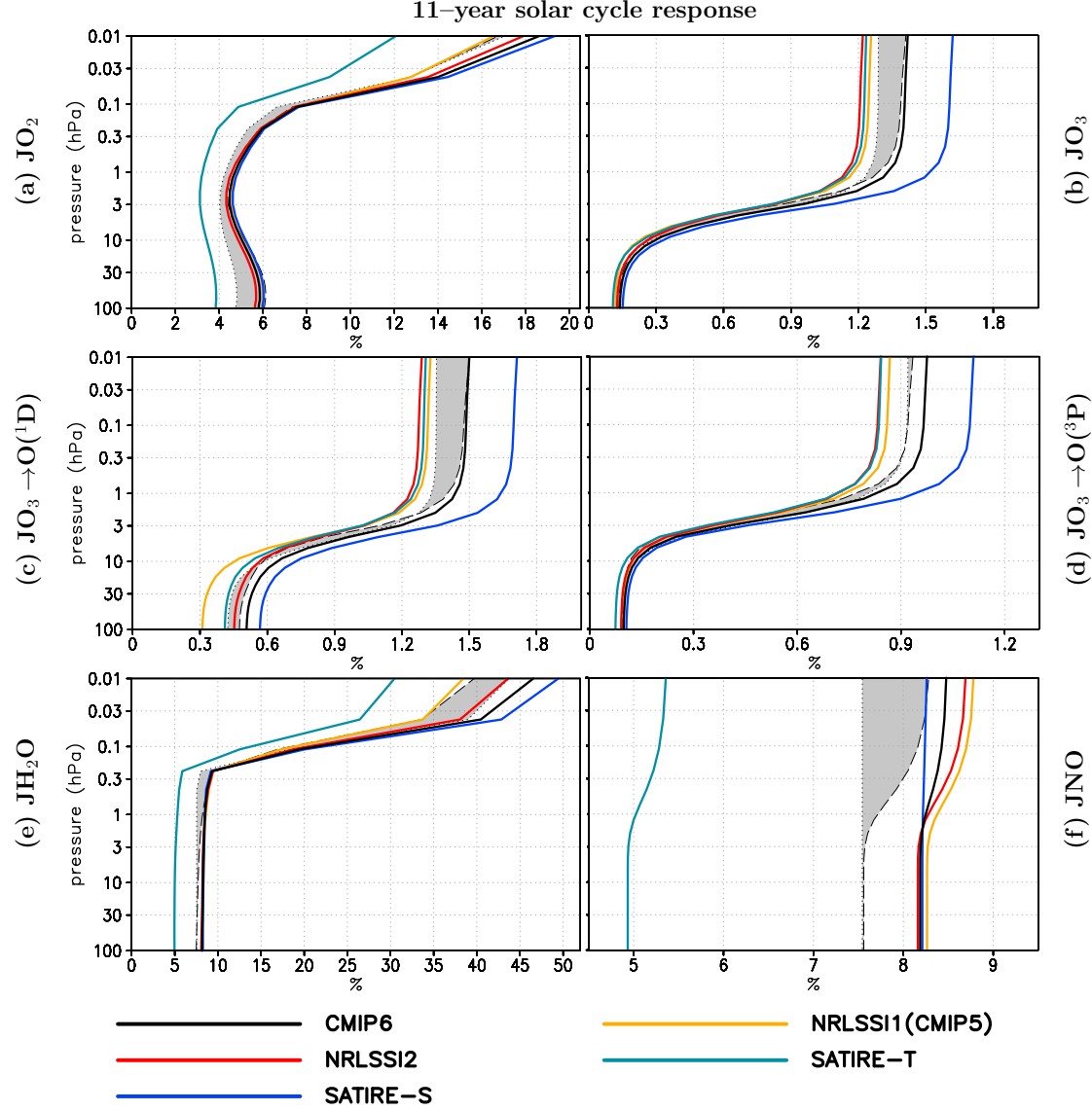

**Figure 6.** Percentage change of the photolysis rates from solar minimum to maximum of (a) oxygen ($JO_2$), (b) ozone ($JO_3$=$JO_3 \rightarrow O(^1D) + JO_3 \rightarrow O(^3P)$), (c) $O_3$ producing $O(^1D)$ ($JO_3 \rightarrow O(^1D)$), (d) $O_3$ producing $O(^3P)$ ($JO_3 \rightarrow O(^3P)$), (e) water vapour ($JH_2O$), and (f) nitric oxide ($JNO$) for a single time step at $180°$E averaged from $60°$S to $60°$N. Changes are derived for an average of WACCM and EMAC simulations using five SSI data sets at solar maximum NRLSSI1(CMIP5) (yellow), NRLSSI2 (red), SATIRE-S (blue), SATIRE-T (dark blue), and CMIP6 (black) relative to the average of the WACCM and EMAC reference solar minimum simulations. The shaded area indicates the range of the WACCM (black long dash contour) and EMAC (black dotted contour) ensemble means.

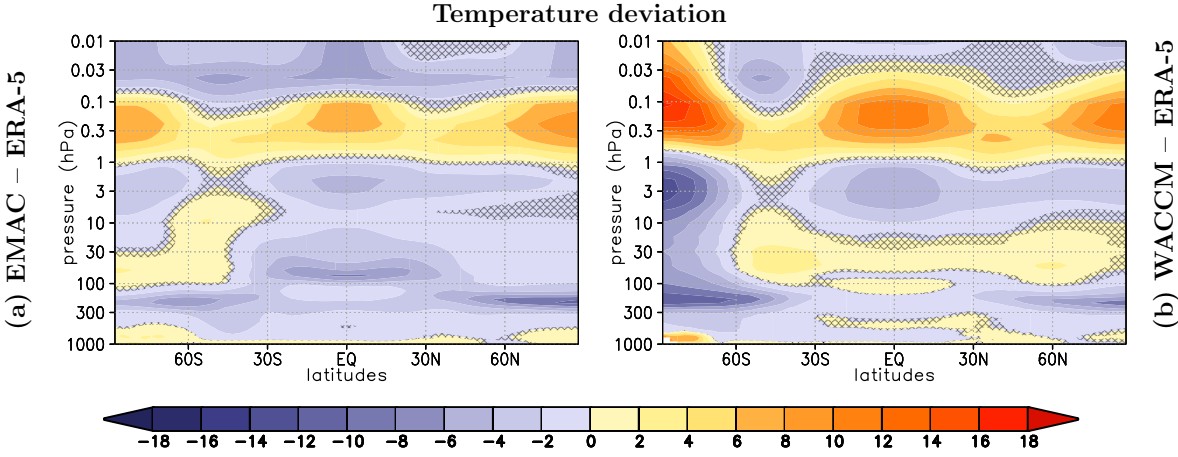

**Figure 7.** Annual mean temperature deviation of EMAC and WACCM to ERA-5 climatology. (a) EMAC – ERA-5 and (b) WACCM – ERA-5 climatology. The ensemble mean for each CCM consists of the solar minimum reference simulation (included 5 times in the ensemble mean) and the 5 simulations of the solar maximum. The ERA-5 data consists of annual mean data from 1982 to 2018. Grey hatching masks areas where differences do not pass a test for statistical significance ($p > 5\%$), for differences relative to ERA-5.

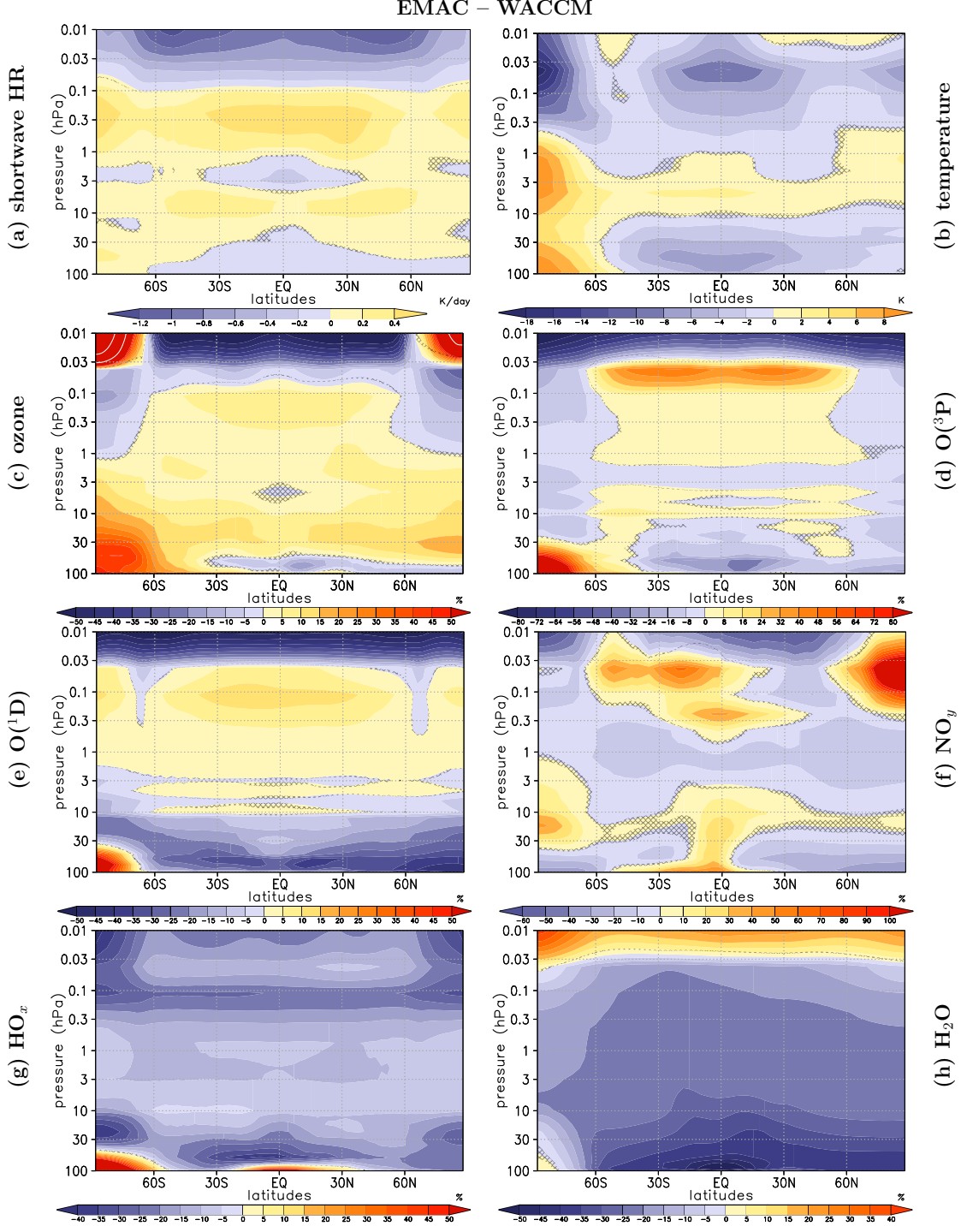

**Figure 8.** Annual mean differences for EMAC (ensemble mean) minus WACCM (ensemble mean) (shaded) of (a) SW heating rates, (b) temperature, (c) ozone mixing ratios, (d) atomic oxygen (O($^3$P)), (e) HO$_x$, and (f) NO$_y$. The ensemble mean for both CCMs consists of the solar minimum reference simulation (included 5 times in the ensemble mean) and the 5 simulations for the solar maximum. Grey hatching masks areas where differences do not pass a test for statistical significance (p > 5 %).

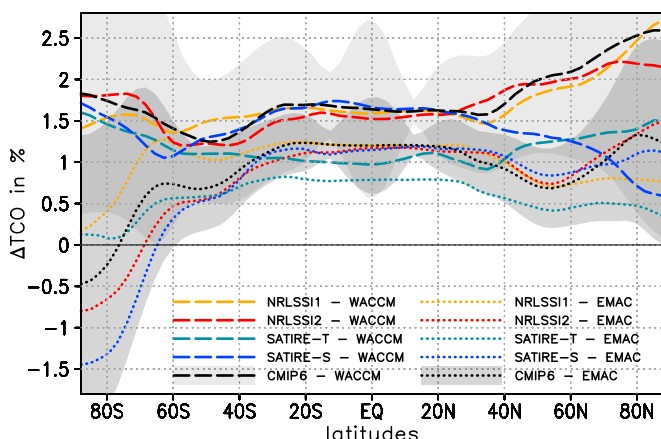

**Figure 9.** Annual mean zonally averaged 11–year solar cycle response in total column ozone (TCO) in % for WACCM (long dashed) and EMAC (dotted) for prescribed SSI data sets NRLSSI1(CMIP5) (yellow), NRLSSI2 (red), SATIRE-S (blue), SATIRE-T (light blue), and CMIP6 (black). The 95% uncertainty range is given for simulations with the CMIP6 data set for WACCM (light grey shaded) and EMAC (dark grey shaded).

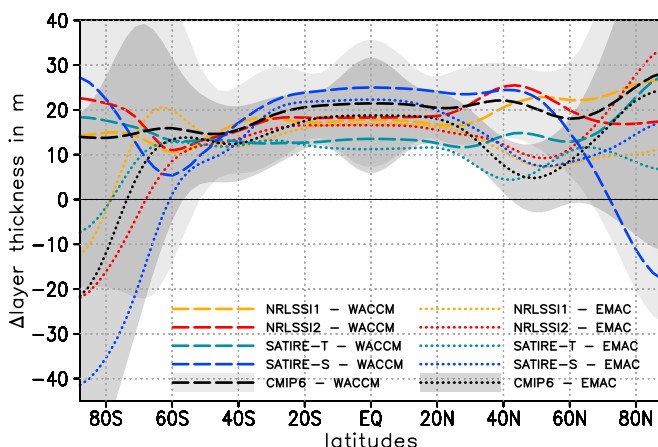

**Figure 10.** Annual mean zonally averaged 11–year solar cycle response in layer thickness from 100 to 10 hPa in m for WACCM (long dashed) and EMAC (dotted) with prescribed SSI data sets NRLSSI1(CMIP5) (yellow), NRLSSI2 (red), SATIRE-S (blue), SATIRE-T (light blue), and CMIP6 (black). The 95% uncertainty range is given for simulations with the CMIP6 data set for WACCM (light grey shaded) and EMAC (dark grey shaded).

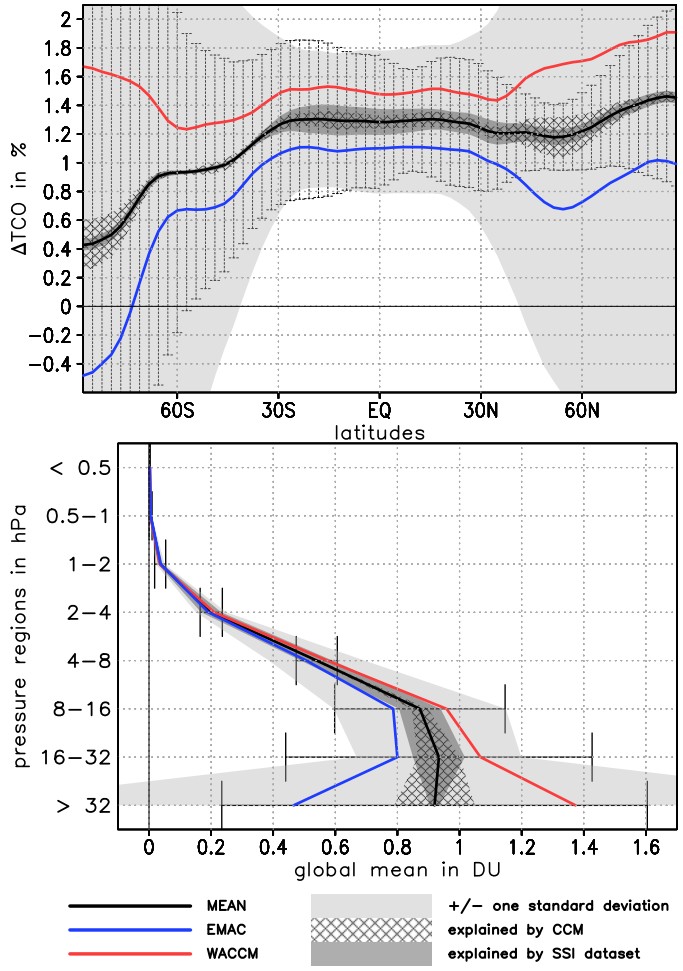

**Figure 11.** Top: Ensemble average, annual and zonal mean 11–year solar cycle response in partial column ozone in % (solid black) and the 95% confidence interval as error bars; WACCM simulations only (solid red); EMAC simulations only (solid blue); light grey shading denotes the standard deviation of the ensemble mean solar response; hatched region denotes the part of the standard deviation explained by the models; dark shading denotes the part of the standard deviation explained by the SSI data sets. Bottom: As top for annual, global mean 11–year solar cycle response in TCO in DU (solid black) for pressure regions as indicated on the y-axis.

**List of Tables**

**Table 1.** Solar cycle spectral solar irradiances changes from Nov. 1989 to Nov. 1994 relative to Nov. 1994 ($\Delta SSI$) in % and relative contribution of SSI changes to the TSI change ($\frac{\Delta SSI}{\Delta TSI}$) in % for the Lyman-$\alpha$ (121.5 nm), Far-UV (121–200 nm), Herzberg continuum/Hartley bands (201–242 nm), Hartley-/Huggings-bands (243–380 nm) and visible (381–780 nm) spectral ranges.

| SSI dataset | Lyman-$\alpha$ 121.5 nm | | Far-UV 121–200 nm | | Herzberg cont. Hartley bands 201–242 nm | | Hartley-Huggings-bands 243–380 nm | | visible 381–780 nm | |
|---|---|---|---|---|---|---|---|---|---|---|
| | $\Delta SSI$ | $\frac{\Delta SSI}{\Delta TSI}$ | $\Delta SSI$ | $\frac{\Delta SSI}{\Delta TSI}$ | $\Delta SSI$ | $\frac{\Delta SSI}{\Delta TSI}$ | $\Delta SSI$ | $\frac{\Delta SSI}{\Delta TSI}$ | $\Delta SSI$ | $\frac{\Delta SSI}{\Delta TSI}$ |
| NRLSSI1 | 44.29 | 0.27 | 11.07 | 1.14 | 3.48 | 5.32 | 0.27 | 22.91 | 0.08 | 54.90 |
| NRLSSI2 | 50.38 | 0.29 | 11.39 | 1.13 | 3.26 | 4.79 | 0.35 | 29.04 | 0.07 | 41.41 |
| SATIRE-T | 35.57 | 0.30 | 7.58 | 1.08 | 2.58 | 5.49 | 0.41 | 48.55 | 0.05 | 42.79 |
| SATIRE-S | 57.48 | 0.33 | 12.09 | 1.19 | 3.60 | 5.30 | 0.55 | 45.52 | 0.07 | 42.96 |
| CMIP6 | 53.94 | 0.31 | 11.74 | 1.15 | 3.43 | 5.00 | 0.45 | 36.86 | 0.07 | 41.86 |

**Table 2.** Correlations of annual average polar region (70°N–90°N) anomalies (solar maximum – solar minimum) of total column ozone (TCO) and the layer thickness from 100 to 10 hPa. TCO change in DU per 100 m geopotential height change and the 95% confidence interval.

| Hemisphere | EMAC | | WACCM | |
|---|---|---|---|---|
| | Correlation | $\Delta$TCO/100 m | Correlation | $\Delta$TCO/100 m |
| | CMIP6 | | | |
| NH | 0.82 | 6.08±0.37 | 0.68 | 6.11±0.57 |
| SH | 0.81 | 7.49±0.47 | 0.77 | 6.06±0.43 |
| | SATIRE-T | | | |
| NH | 0.81 | 5.91±0.37 | 0.69 | 6.03±0.55 |
| SH | 0.82 | 8.02±0.48 | 0.76 | 5.38±0.40 |
| | SATIRE-S | | | |
| NH | 0.83 | 6.20±0.36 | 0.71 | 5.91±0.51 |
| SH | 0.81 | 7.72±0.48 | 0.69 | 5.09±0.46 |
| | NRLSSI1 | | | |
| NH | 0.84 | 5.95±0.34 | 0.70 | 6.25±0.55 |
| SH | 0.82 | 7.26±0.44 | 0.76 | 5.94±0.44 |
| | NRLSSI2 | | | |
| NH | 0.82 | 6.39±0.38 | 0.66 | 5.84±0.57 |
| SH | 0.83 | 7.81±0.45 | 0.76 | 5.54±0.41 |