# Peer review of "Quantifying uncertainties of climate signals in Chemistry Climate Models related to the 11-year solar cycle. Part I: Annual mean response in heating rates, temperature and ozone"

_Atmospheric Chemistry and Physics, 2019_

## Short Comment (SC1) · 1 Feb 2020

For a climate signal such as QBO, the observed 28-month cycle is directly related to the interaction between the semiannual solar nodal crossing with the 27.21 day lunar draconic (or nodical) cycle. Above the altitude of the QBO, the semi-annual SAO occurs, suggesting a transition from tidal forcing to a primarily solar semi-annual radiative forcing cycle. This set of forcing factors is certainly more important than the rather weak 11-year cycle in sunspot activity, and the asymptotic agreement with a tidal forcing pattern only gets more apparent as more data is accumulated over the years. This

agreement is shown in Fig. 1 shown below. The only question is what causes the fluctuation over the years and perhaps this is in some way related to disturbances such as SSW, ENSO, or 2nd-order solar variations such as sunspot levels. (p.s. thank you for maintaining the QBO data at fu-berlin.de)

**Average QBO period - 30 mBar data**

○— cumulative mean   —— draconic asymptote   + individual period length

*average cycle length (years)*

http://www.geo.fu-berlin.de/met/ag/strat/produkte/qbo/qbo.dat

Year

**Fig. 1.** Asymptotic trend of QBO approaches that predicted by the aliased draconic cycle of ~2.37 years

---

## Referee Comment (RC1) · Anonymous Referee #1 · 25 Feb 2020

This interesting and well written study addresses the impact of the choice of solar forcing model versus choice of CCM model through an ANOVA analysis of annual mean response rates. Although the study itself is performed under controlled conditions (no forcing from particles, no solar cycle, yearly means, ...)  it sheds new light on the relative impact of the model choice and solar spectrum choice on heating rates, ozone, etc. In particular, the study highlights the influence of the CCM choice on the upper mesosphere and the impact of the prescribed solar forcing in the FUV on the response of the upper stratosphere and lower mesosphere. This excellent work is definitely worth

publishing in ACP.

General comments:

p3 line 10: the amplitudes are added... If the reference ATLAS3 spectrum underestimates the SSI in some spectral band, then this means that the departure from the true spectrum will affect all reconstructions, thus impacting the climatological state of the atmosphere. This effect may be significant in the visible and near-IR where Delta-SSI is relatively small as compared to the uncertainty on the reference spectrum. Although you briefly mention this in the conclusions I would recommend to address this issue (if it is one) here already because what follows heavily relies on the ATLAS3 reference spectrum.

p3 line 10: The SORCE dataset has received considerable attention (e.g. Haigh et al., 2010, https://doi.org/10.1038/nature09426) because of its anomalous solar cycle variability. Alas, it is implictly excluded from your analysis because of the considered time interval. Yet, I would still mention it here because of the continuing debate.

p3 line 15: here it is important to give a physical flavour of why your ANOVA analysis can be useful, e.g. by mentioning that it is closely connected to regression analysis. Just saying that you're the first to use it does not help much in understanding what it is about.

p8 line 16: please replace "solar signal" by solar signature or similar because you are not really considering a signal, rather perpetual conditions. In this whole section the question that immediately arises is to what degree the modulation of that solar forcing by the 11-year cycle can affect your conclusions, e.g. through coupling with the NAO or, more generally, with the lagged ocean response. Please explain if and how these effects may impact your conclusions.

p9 line 24: Here a brief rationale of why the ANOVA approach is pertinent is a must. Most readers are familiar with multilinear regression analysis, so that this

analogy can be easily exploited. Please also give some adequate references (e.g. H. von Storch and F. W. Zwiers, Statistical analysis in climate research, Cambridge Univeristy Press, 2002) and above all, explain in more physical terms what you are trying to quantify with your ANOVA analysis. I also recommend to cite some climate studies that illustrate the use of ANOVA analysis in climate studies, such as the early https://doi.org/10.1357/0022240943076911 or the more recent https://doi.org/10.1002/joc.3991

One additional request: why a two-way analysis? Again, for those who are unfamiliar with ANOVA analysis, I recommend to motivate these choices here and then defer to the appendix for technical details.

p9 line 25: Why consider annual means only and not separate seasons (e.g. DJF) for which we know that the sensitivity may be higher ? Yearly averages tends to smear these seasonal differences.

p17 I would suggest to mention as well the comparisons between the different spectral irradiance models and ozone observations (e.g. Ball et al., 2016, https://doi.org/10.1038/ngeo2640) which, broadly speaking, support your conclusions or at least do not contradict them.

p21 line 20: The investigation of more recent periods, instead of the 1989-1994 comparison would allow to better constrain the SSI variability (with SOLAR-ISS as you mention, but also other observational datasets such as AURA-OMI) and better overcome the main source of uncertainty in the solar forcing, which comes from the FUV range.

p23 line 15: Most ANOVA studies focus on the F ratio although one could also consider the coefficient of multiple determination R (e.g. von Storch and Zwiers, p. 176) which, arguably, gives a better physical picture. Did you consider it?

Technical corrections

[Figure]

Title: since this is part 1, what should we expect to find in part 2? The manuscript does not really tell this.

p4 line 27: the appropriate reference for the Bremen MgII is Snow et al., 2014 (https://doi.org/10.1051/swsc/2014001)

p6 line 10: observations

p35 Legend of Fig 4: does not pass –> do not pass

––––––––––––––––––––––––––––––

---

## Referee Comment (RC2) · Anonymous Referee #2 · 26 Feb 2020

Review of "Quantifying uncertainties of climate signals related to the 11 year solar cycle. Part I: Annual mean response in Heating Rates, Temperature and Ozone" by Kunze et al.

This manuscript presents results from a series of CCM simulations using two different models (WACCM and EMAC) and five different SSI data sets (NRLSSI1, NRLSSI2, SATIRE-T, SATIRE-S and CMIP6). These simulations are used to investigate how the annual mean response of solar heating rates (due to UV absorption by ozone and O2), temperature, and ozone in the stratosphere and mesosphere in each model depends

on the input variations in SSI from these 5 data sets relative to common solar minimum SSI specified by the ATLAS-3 reference spectrum. The relative responses of the modeled variables due to differences in SSI (i.e, external forcing) versus differences between the two models (i.e., internal variability) are calculated using two-way analysis of variance (ANOVA). According to the abstract (page 1 of manuscript, lines 9-12), the authors report that differences among the SSI data sets have strongest influence on the modeled shortwave heating rates, ozone, and temperature in the upper stratosphere and mesosphere, while the largest differences attributed to the treatment of photochemistry, etc. in the CCMS are identified in the upper mesosphere (page 1 lines 13-14). The authors also indicate an apparent model-dependent solar cycle response in the lower stratosphere.

The subject of this paper, i.e., identifying and quantifying sources of uncertainty in modeled atmospheric response to 11-year variations in SSI, is compelling and the basic tools (2 sets of CCM simulations, SSI data sets, application of well-established ANOVA statistical methods) are appropriate. The use of a common reference SSI data set for solar minimum and application of relative changes in SSI informed by the different data sets makes sense. There are several areas (identified below) where the present manuscript does not provide enough information to allow the reader to understand this work and its implications. These areas can be summarized as follows: (1) Context – why did the authors undertake this study, and what solid, quantitative conclusions does this study offer that will be of use to other researchers; (2) The advantages and limitations of the ANOVA approach – the limitations in particular need to be clarified; (3) Presentation –some figures and some of the discussion were difficult to understand, some reorganization and revision is warranted to improve the overall readability of the manuscript. These areas should be addressed in a revised manuscript before I can recommend publication.

Major Comments:

1. The title of the manuscript is not very descriptive. It should probably be stated

somewhere in the title that this is a modeling study. One could read this title and think this is an observational study of variations in climate (e.g., surface temperatures, precipitation patterns, etc.), rather than a very specific examination of how sensitive middle atmospheric processes in CCMs are to imposed variations in SSI related to the 11-year solar cycle. The title also says this is Part I, but I do not understand why this is so. What will part II be about, and why does this need to be a two-part study?

2. The authors have undertaken a very ambitious task requiring a lot of detailed statistical analysis. After reading the introduction, it is still unclear to me why this study is being performed. The authors state (page 2 line 8) that we have a good understanding of the chemical and dynamical processes, but discrepancies between the observed and modeled responses remain? What is not stated is how big these discrepancies are, and why they are important within the larger context of climate modeling. Some more quantitative discussion of these discrepancies in the introduction are needed. For example, page 2 line 25 states there is "large model spread" – how large is this and why is it important? Is this spread larger than uncertainties in observational-based estimates of 11-year variations related to solar forcing?

3. The discussion of the "top down" vs. "bottom up" mechanisms (page 2 lines 13-25) should be condensed and revised to clearly state that this study is focusing entirely on the "top down" effect. The "bottom up" effect relies not only on changes in TSI but also on a very complex interaction between ocean and atmosphere, and it should be stated that the present study cannot address this mechanism with the model simulations presented here.

4. The ANOVA method finds the largest uncertainties in the upper mesosphere, but I don't think any of the observational studies cited in the Introduction deal specifically with the upper mesosphere. From what's presented in the manuscript, it's unclear how quantifying these upper mesospheric uncertainties are directly relevant to improving our understanding of climate signals. Looking at Figure 2, it appears that in the stratosphere (where most of the ozone resides and where this "top-down" mechanism

dominates), the details of the SSI input don't really matter – you get essentially the same modeled response (excluding SATIRE-T), i.e., any differences are smaller than the internal model variability. If this is the case, this should be clearly stated in the abstract and in Section 7.

5. Appendix A describes the ANOVA method. I am not an expert in this field, so my comments here are for clarification rather than criticism.

a. My understanding is that part of the ANOVA approach is to construct a model describing the sources of variance, making certain assumptions about what these sources are, and then testing this model to see how much of the variance is explained. The model should be described and listed in equation form – is it a two way model with interaction? Is there a specific term for variance from random error?

b. With regard to figure 1, center and right columns, it's clear that not all the variance is being explained by the SSbB (center) and SSbA (right) terms. This is alluded to on page 10 line 1-2, where the authors state that the random contribution is largest.

c. For a complete description of the problem, would it be better to limit figure 1 to the annual mean responses, and construct a new figure 2 listing all terms of the ANOVA model so we can see all relevant terms (e.g. the treatment A term treatment B term and the interaction term)? It would be most helpful if the description of current Figure 1 middle and right columns referred directly to the terms and equations in the Appendix so we know for sure what is being plotted, i.e. middle column is SSbB term equation A3, etc. Also the hatched areas in the middle column for SSI/temperature and SSI/ozone are extremely hard to see, making it difficult to understand what is and isn't significant.

d. Please explain in more detail how degrees of freedom were determined. The CMIP6 data set is an average of NRLSSI2 and SATIRE data sets, so it's not an independent member of the K=B group. Shouldn't this affect the degrees of freedom that ultimately impact the significance tests with the F statistic?

e. Outside of the upper mesosphere, it seems like the SSI changes and CCM differences together don't explain the majority of the variance in the total sum of squares. What does this mean? Is this analysis meaningful? Should we conclude that differences in SSI reconstructions or differences in details of model photochemistry or spectral resolution in the SW heating aren't that important relative to the random model variability?

6. Section 7 needs revision as noted below:

a. First page 19 lines 12-25 repeat what has already been said in the introduction and could be removed or condensed significantly.

b. Page 19 line 17: it is stated that SSI data sets provide largest fraction of solar cycle variance in the upper stratosphere/lower mesosphere (30% for heating rate, 30% for ozone, 10 % for temperature) but that is not strictly true. The majority of the variance is unexplained, wrapped up in an interaction term or some other manifestation of random model variability. I think it would suffice to say that the SSI differences explain up to 30% in variance in heating and ozone, and only 10% in temperature.

c. Page 21 lines 5-13: The discussion of the total ozone effects is confusing, and does not seem to produce any specific conclusions. The sentence "Distinct differences in TCO anomalies between the CCMs are also reflected by the relatively large fraction of the anomaly variability that can be explained by differences between the CCMs" seems circular and it's not clear to me what the authors are trying to say. The finding that WACCM and EMAC models have lower percentage of TCO response from p > 16 hPa compared to one observational study (Hood 1997) does not seem to be directly relevant to the state purpose of this study, especially since by design these CCM simulations do not have realistic decadal variations in lower atmosphere forcing (i.e., fixed repeating monthly mean SST's, etc). It's clear from Figure 8 that the ANOVA method cannot disentangle variance related to SSI and model transport in the lower stratosphere. Based on what's presented in this paper, it would make sense to keep Figure

6, omit Figures 7 and 8 and related discussion, and summarize your findings (i.e., that statistically significant attribution of TOC variance related to SSI or CCM differences as you've defined them is not possible due to large internal model variability).

d. Page 21 lines 14-19: Based on the discussion here, it's not clear why a two-way ANOVA approach is warranted compared to a 1-way (SSI changes) approach. Basically, you are saying you don't think you have fully sampled the "CCM spread" as it is referred to here. So why is 2-way justified? This might be a good place to note the importance of experimental design when using ANOVA that could help guide future investigations.

Additional comments, revisions, suggestions:

1. Abstract: It should be noted somewhere in abstract that you are using time slice integrations based on 1989-1994 differences in SSI.

2. Page 1 line 16: can you define middle atmosphere?

3. Page 2 line 3: the authors cite one reference here (McCormack and Hood), but there are a lot of subsequent studies on observed solar cycle variations that should also be referenced. As mentioned above, observational studies for the mesosphere in particular would be good, since this is where you end up seeing the biggest impact of SSI differences. For example, Beig at al JGR 2012 (https://doi.org/10.1029/2011JD015697).

4. Section 2: It would be most helpful to have a plot comparing the different SSI data sets somehow. Is it possible to plot the SSI differences relative to the ATLAS solar min values over a range of UV wavelengths. This would illustrate for the reader how differences among the different data sets compare to the overall 11-year max-min differences. Since the change in SSI from solar max – min is strongly wavelength dependent, this might be more informative than Table 1 that averages over very large intervals.

5. Section 3.1: Why are the QBO treatments different? What observed winds are used

for the relaxation in EMAC and for what period of time? In doing ANOVA, experimental design is very important. Were these CCM simulations designed and performed especially for this study, or is this study using simulations that were generated previously. This could be helpful to note in the paper. If these were simulations already generated, this paper is more of a proof of concept on how to apply ANOVA and how perhaps future multi-model CCM experiments should be designed in order to best use the ANOVA method.

6. Figure 4 and related discussion in the text could be removed. In its present form, it doesn't add much information, especially since I'm not sure how much data ERA5 uses in the upper stratosphere/mesosphere, meaning the ERA5 fields could be very model-dependent themselves, and not the best standard to compare with. It might be more illuminating to directly compare differences in zonal mean T, zonal wind, and ozone between WACCM and EMAC.

7. Page 9 line 33 – I really can't see the grey hatching in Figure 1 very well. Is it possible to plot it another way? Maybe only plot significant values?

8. Page 11, line 9: it is stated that the t test is and resulting error bars come from the complete ensemble but Figure 2 caption states the error bars are for the WAC-CMX/EMAC CMIP6 simulations. Which is it? 9. Page 12: I'm not sure it's worth reviewing Chapman cycle photochemistry here. If the authors wish to describe specific reactions in detail, I would suggest using equation form rather than in the text, and perhaps put some of the more complex reaction in an appendix?

---

## Referee Comment (RC3) · Anonymous Referee #3 · 28 Feb 2020

Review of Quantifying uncertainties of climate signals related to the 11-year solar cycle. Part I: Annual mean response in heating rates, temperature, and ozone by M. Kunze et al. Atmos. Chem. Phys. Discuss., https://doi.org/10.5194/acp-2019-1010-RC1, 2020

This study is focused on quantifying the uncertainties in two different CCM model output (heating rates, temperature, ozone, HOx, etc.) due to the 11-year solar cycle and, in addition, to quantify what portion of the uncertainty is due to differences in SSI forcing dataset and what portion is due to differences in the radiation and photolysis schemes/parameterizations of the different CCMs. The study used a two-way ANOVA

for results. I am not an expert in ANOVA so I greatly relied on Appendix A to interpret the manuscript. My understanding is that the two-way ANOVA consisted of binning the simulations into two groups and evaluating for how each group, and the interaction of the groups, contributed to the total variability in the simulations. One "treatment" consisted of 45 years of simulations for each of 5 unique SSI data sets (45*5 = 225 elements) of each CCM (WACCM and EMAC); this first treatment then was more of a comparison between the CCM outputs. The second "treatment" consisted of 45 years of simulations for each of the 2 unique CCM's of each SSI dataset; this second treatment was then more of a comparison of the SSI datasets.

Differences in CCM's were found to contribute more to the uncertainty in the upper mesosphere whereas differences in SSI datasets were found to contribute more to the uncertainty in the upper stratosphere and lower mesosphere. However, the majority of the variability in the output was due to "internal variability" in the models.

I find the study of high interest and worthy of publication. I do provide some comments below for consideration. In brief, these focus firstly on the solar irradiance dataset aspect of the study. I find the adoption of a common spectrum from which to baseline differences in the solar cycle 'amplitudes' of the various datasets novel. Secondly, the messaging of the study could be improved to provide the background of why the study was undertaken.

SSI Comments a. The general reader may be unclear why it was necessary to use 5 different SSI datasets, or why you chose the ones you did, or even that there isn't agreement across SSI datasets on longer time scales (observed or modeled) . I would suggest adding a paragraph or two to improve the messaging behind your study, probably in the Intro or in Section 2. b. I agree that TSI observations are relatively short (since 1978) and that SSI observation record is even shorter, nor full spectral coverage, and has time gaps. However, your study does select a relative short period of time to investigate the impacts of SSI over (1989-1994). Therefore, it begs the questions of why that particular time range and not another when full spectrum observations existed

(i.e. during the SORCE era) or even partial spectrum observations (265-500 nm) by the AURA OMI instrument. In essence, I'm asking you to more directly draw the line between your "focused" study and the SSI dataset needs of the model intercomparisons studies like CMIP6 which require full spectrum and very long time coverage. This leads to necessary use of modeled SSI datasets, which have differences between them and with observations. It would be helpful to bring the discussion of the Coddington et al and Yeo et al. results (Page 6, line 7 through end of paragraph) in earlier in the section for this reason. c. I do like that you've chosen a single spectrum to adopt as a common baseline for solar minimum conditions. I feel that's quite novel. I am concerned, though, that the manuscript doesn't adequately address how this approach might impact results. You do say that a reference baseline would lead to a certain climatology state (end of page 13 to page 14) and that differences from that baseline, as would occur from using SOLAR-ISS as the reference, would result in a different climatology. However, is it necessarily true that the solar response variations are truly linear from an adopted baseline? Maybe more clear way to ask is whether gas phase reaction rates or water vapor abundances that you mention on page 14 might "bottom out" or "max out" if the baseline climatology/temperature was too high or too low? I would also suggest bringing this discussion up earlier, in addition to where it is in the conclusions. d. In conclusions you also discuss how choosing SC 22 (selected, I understand, because of time range of ATLAS 3 observations) should be reflective of other solar cycles in the 21st century. You examined the irradiances in the Lyman alpha through UV for the various SSI datasets with other solar cycles and found a linear relationship. Was that relationship with TSI magnitude, sunspot number, or something else? In the Coddington et al., 2019 paper you reference, their Tables 3 and 5 show a larger change in integrated SSI (in the 100-200 nm bin) from solar cycle to solar cycle than occurs in differences across some of the datasets you use in your study. Similar to the above comment, you might want to bring this up earlier in Section 2 as well. e. It's possible this is jargon in the CCM community, but is it typical to use phrases of 'solar cycle response' for simulations where the transition from perpetual solar minimum to perpetual

solar maximum is quite abrupt?

General comments Page 2, lines 24 – 29: The end of the one paragraph is focusing on the CCM model "spread" caused by differences in spectral resolutions of the shortwave radiation parameterizations or photolysis in the models. The next paragraph begins with different spectral distribution of the SSI data set also impacting CCM models. In the 2nd case, you are referring to the magnitude of the SSI within a spectral bin and not differences in spectral resolution of the SSI observations, but this could easily be confused during the transition of one paragraph to the next.

Page 3, line 3-4: You end with "the effects of the 11-year solar cycle differences in spectral distribution and amplitude...". However, by adopting the common reference baseline spectrum, you have removed the effects of spectral distribution from the study. It's clear from your earlier text what you mean and that it's just an error here.

Page 4, line 12: What type of scaling did you apply to make ATLAS 3 integrate to SORCE TIM TSI? Wavelength independent? "The extended ATLAS-3 spectrum was then scaled to obtain..."

Page 4, line 20: Needs some clarification. The (facular brightening and sunspot darkening) indices themselves do not describe the relationship between sunspots and faculae on the Sun's disk and irradiance. The indices are derived from observations of proxies of faculae and sunspots. It's rather the scaling factors computed from the multiple linear regression of these indices with SSI observations that are used to scale the change in faculae and sunspots into a net, wavelength-dependent, irradiance change.

Page 4, line 23: "The TSI changes are added..." should be "The SSI changes are added.."

Page 4, line 27: While Viereck et al., 2001 is a perfectly appropriate reference for a general discussion of the Mg II index, the correct citation for the University of Bremen Mg II index reference is Snow, M., Weber, M., Machol, J., Viereck, R., & Richard,

[Figure]

E. (2014). Comparison of magnesium II core‐to‐wing ratio observations during solar minimum 23/24. Journal of Space Weather and Space Climate, 4, A04. https://doi.org/10.1051/swsc/2014001

Page 5, line 23-24: I am aware that CMIP6 SSI and TSI data are the average of output from NRLSSI2 and SATIRE-S. However, it's unclear to me the relation of this is to your choice of using data from November 1989 and November 1994 in the study?

Page 6, EMAC section: I'm not an expert on CCMs but I find the description of EMAC difficult to read. It doesn't flow as easily as the following section on WACCM, and the acronyms aren't defined. I would suggest some word-smithing to bring it up to the same high quality as the rest of the paper.

Page 7, between WACCM and section and the start of Section 3.1: Again because I'm not an expert on CCM's, it would be nice to have a summarizing sentence or two here as a take home message for the non-expert. Are these suitable models to compare, and are there obvious reasons why their unique setup and execution would lead you to expect differences in their outputs?

Page 7, lines 27-28: One too many of each of the words, "both" and "simulations".

Page 9, Section 7: A general comment in this section is to make it is more clear that majority of the uncertainty, or spread, in the CCM output comes from internal variability in the CCM's. Only a fraction of the model spread can be attributed to differences in SSI datasets or differences in the CCM's themselves. (If I understood correctly).

Page 13, line 20: "...10-40% of the variability of the solar signal [ insert of what component, heating rate, temperature, etc.] in the stratosphere and ..."

Page 20, line 8-10: Is there a transition in thought from the sentence ending on line 8 about the distinct differences that appear for SATIRE-T to the next sentence discussing how reduced solar cycle amplitude explain the weaker solar signals in temperature? Does that 2nd sentence also refer to SATIRE-T? If so, Table 1 shows that SATIRE-T

has a stronger solar cycle amplitude in the 201-242 nm range, not weaker.

Page 21, line 3-4: You provide support that downward transport of thermospheric photolysis reactants is needed to realistically simulate solar cycle effects. Is this a new finding for the CCM community? It seems to me that you might emphasize the importance of this in guiding CCM model development and directing CCM advances.

Figures: I was finding that the significance hatching in the figures was very difficult to see, particularly in the middle and right hand columns of Figure 1. However, when I look today, it's much clearer on-screen. Perhaps it is just a problem with my printer.

---

## Author Comment (AC1) · 7 May 2020

We thank Paul Pukite for his comment, that we try to answer below.

*For a climate signal such as QBO, the observed 28-month cycle is directly related to the interaction between the semiannual solar nodal crossing with the 27.21 day lunar draconic (or nodical) cycle. Above the altitude of the QBO, the semi-annual SAO occurs, suggesting a transition from tidal forcing to a primarily solar semi-annual*

[Figure]

*radiative forcing cycle. This set of forcing factors is certainly more important than the rather weak 11-year cycle in sunspot activity, and the asymptotic agreement with a tidal forcing pattern only gets more apparent as more data is accumulated over the years. This agreement is shown in Fig. 1 shown below. The only question is what causes the fluctuation over the years and perhaps this is in some way related to disturbances such as SSW, ENSO, or 2nd-order solar variations such as sunspot levels. (p.s. thank you for maintaining the QBO data at fu-berlin.de)*

Your question concerning the influence of the 11-year solar cycle on the period of the QBO cannot be answered with the simulations performed for our study. The simulations are performed with a setup where the QBO is included by the relaxation of the zonal mean zonal wind to an observed (EMAC) or idealized (WACCM) QBO. Due to this setup there are no differences in the QBO phases between the time slice simulations for solar maximum and solar minimum. To simulate the influence of the 11-year solar cycle on the length of the QBO period, a setup with a higher vertical resolution of EMAC and WACCM has to be used.

---

## Author Comment (AC2) · 8 May 2020

**Answer to anonymous Referee #1**

We thank the Referee for her/his comments and suggestions. We answer point by point in the following with the Referee's comments added in *red/italics*. Text added to the revised version of the manuscript is included here in *blue/italics*.

*This interesting and well written study addresses the impact of the choice of solar forcing model versus choice of CCM model through an ANOVA analysis of annual mean response rates. Although the study itself is performed under controlled conditions (no forcing from particles, no solar cycle, yearly means, ...) it sheds new light on the relative impact of the model choice and solar spectrum choice on heating rates, ozone, etc. In particular, the study highlights the influence of the CCM choice on the*

10 *upper mesosphere and the impact of the prescribed solar forcing in the FUV on the response of the upper stratosphere and lower mesosphere. This excellent work is definitely worth publishing in ACP.*

**General comments**

*p3 line 10: the amplitudes are added... If the reference ATLAS3 spectrum underestimates the SSI in some spectral band, then this means that the departure from the true spectrum will affect all reconstructions, thus impacting the climatological state of*

15 *the atmosphere. This effect may be significant in the visible and near-IR where Delta-SSI is relatively small as compared to the uncertainty on the reference spectrum. Although you briefly mention this in the conclusions I would recommend to address this issue (if it is one) here already because what follows heavily relies on the ATLAS3 reference spectrum.*

In the introduction (page 2, lines 27–35; page 3 lines 1–4) the effect of differences in the spectral distribution between SSI data

20 sets on the simulated temperatures are briefly discussed when the atmospheric state during solar mininimum is modelled:

*Besides their solar cycle response on the thermal structure and dynamics of the middle atmosphere, the different spectral distribution of a SSI data set can also have an impact on the averaged middle atmospheric temperature, as was found in studies comparing different SSI data sets. It was shown that differences in the energy distribution during the solar minimum phases of*

25 *individual SSI data sets may cause significant differences in the simulated temperatures in the middle atmosphere (e.g., Zhong et al., 2008; Oberländer et al., 2012). Even when scaled to the same TSI, the variable spectral distribution of energy within the SSI data sets can cause significant changes of the simulated climatological temperatures in the middle atmosphere. As shown in Matthes et al. (2017), climatological annual mean middle atmospheric temperatures in the tropics can be up to 1.6 K lower when using the CMIP6 recommended SSI data set instead of NRLSSI1.*

30

This can be translated to the common solar minimum references state that is used as a base line for the maximum state of the five SSI data sets in this study.

*p3 line 10: The SORCE dataset has received considerable attention (e.g. Haigh et al., 2010, https://doi.org/10.1038/nature09426)*

35 *because of its anomalous solar cycle variability. Alas, it is implictly excluded from your analysis because of the considered time interval. Yet, I would still mention it here because of the continuing debate.*

The SORCE SSI data are not included in our study, as it is not suited for modelling studies which usually span several decades. We have mentioned some studies using the SORCE SSI data in the introduction of the revision.

40

*The deviant solar cycle behaviour of the SORCE measurements has motivated a number of CCM studies (e.g. Haigh et al., 2010; Merkel et al., 2011; Ball et al., 2011, 2016; Swartz et al., 2012) comparing simulations prescribing SORCE (Solar Radiation and Climate Experiment) SSI data and reconstructed SSI of the Naval Research Laboratory SSI (NRLSSI) or the*

*Spectral And Total Irradiance REconstructions (SATIRE) model.*

*p3 line 15: here it is important to give a physical flavour of why your ANOVA analysis can be useful, e.g. by mentioning that it is closely connected to regression analysis. Just saying that you're the first to use it does not help much in understanding what it is about.*

The method of multiple linear regression and the ANOVA method are based on the same linear statistical model, assuming constant variances and a Gaussian distribution of the residuals. However, the purposes of the ANOVA as applied in this study and regression differ. Here, the ANOVA method is used to analyse whether there is a statistical significant difference between the mean values of a data set when grouped according to some kind of treatments. In our case two treatments are applied, and the data set (the annual mean solar responses) are grouped by SSI data set and CCM. The purpose of a regression analysis is to find relationships between a response (dependent) data set and independent data sets. Both methods have in common that they allow to estimate the portion of variance that can be explained either by the treatments or the independent data sets.

We have changed the sentence introducing the ANOVA in the Introduction, added some references, and have added a sentence that explains the main purpose of the ANOVA in our context.

*To separate the influence of the SSI data sets and the CCMs on the solar responses in SW heating rates, temperature and ozone, a two-way analysis of variance (ANOVA) method (e.g Fisher, 1925; von Storch and Zwiers, 1999) has been applied. While the ANOVA is a well established method in many scientific fields, it is used rarely in the field of climate research (e.g Geinitz et al., 2015; Evin et al., 2019). Here we use ANOVA for the first time to quantify the uncertainty of the atmospheric response to decadal solar variability. The ANOVA-approach enables us to analyse if the usage of different CCMs or different SSI forcing datasets yields significantly different solar responses and to quantify which share of the total variance of the ensemble's solar response is related to either of the two factors (called treatment in the ANOVA-context).*

*p8 line 16: please replace "solar signal" by solar signature or similar because you are not really considering a signal, rather perpetual conditions.*

To reflect the time slice character of our simulations, the term "solar signal" is replaced by "solar response" throughout the manuscript.

*In this whole section the question that immediately arises is to what degree the modulation of that solar forcing by the 11-year cycle can affect your conclusions, e.g. through coupling with the NAO or, more generally, with the lagged ocean response. Please explain if and how these effects may impact your conclusions.*

The 11-year solar cycle responses from transient simulations, which are usually extracted by multiple linear regression or composite analysis, as these simulations contain the full range of natural variability. However, the extracted solar signals are often quite similar compared to the solar responses from time slice simulations for solar maximum and minimum. We do not expect large differences of the 11-year solar cycle responses for transient simulations in the upper stratosphere to upper mesosphere.

*p9 line 24: Here a brief rationale of why the ANOVA approach is pertinent is a must. Most readers are familiar with multilinear regression analysis, so that this analogy can be easily exploited. Please also give some adequate references (e.g. H. von Storch and F. W. Zwiers, Statistical analysis in climate research, Cambridge Univeristy Press, 2002) and above all, explain in more physical terms what you are trying to quantify with your https://www.overleaf.com/project/5e636cd609eeae000174ea02ANOVA analysis. I also recommend to cite some climate studies that illustrate the use of ANOVA analysis in climate studies, such as*

*the early https://doi.org/10.1357/0022240943076911 or the more recent https://doi.org/10.1002/joc.3991*

We have added some additional references, related to previous application of ANOVA in climate studies, in the introduction. To better motivate the usage of the ANOVA, we have added this short paragraph at the beginning of Section 5.

*The averaged solar response discussed in Section 4 is supposed to be different with respect to the SSI data set prescribed and with respect to the CCM applied in each run. From the differences in the SSI amplitudes in the broad bands shown with Table 1 we expect the solar responses to be slightly different for each SSI data set, as we do for each of the CCMs. As the method of regression analysis can be used to calculate the fraction of the variance explained by a regressor, the ANOVA can quantify the fraction of explained variance by a certain factor.*

*One additional request: why a two-way analysis? Again, for those who are unfamiliar with ANOVA analysis, I recommend to motivate these choices here and then defer to the appendix for technical details.*

The two-way ANOVA is a natural choice in light of our experimental setup where two CCMs apply five SSI data sets in the same way. This allows us to group the complete data set of annual mean solar responses by CCM and SSI data set. We have added an explanation to the manuscript.

*The two-way ANOVA is used, as there are two treatments influencing the annual mean solar responses. As these treatments are not applied independently the interaction of CCMs and SSI data sets has to be taken into account.*

*p9 line 25: Why consider annual means only and not separate seasons (e.g. DJF) for which we know that the sensitivity may be higher ? Yearly averages tends to smear these seasonal differences.*

The analyses of the seasonal differences is intended in Part II of the study which will focus on solar impacts on dynamical parameters of the atmosphere. This is now mentioned in the main text at the end of the introduction.

*In this Part I of our study we concentrate on the annual mean solar response in heating rates, temperature and ozone, while Part II (in preparation) focuses on the dynamical solar and auroral responses in northern winter.*

*p17 I would suggest to mention as well the comparisons between the different spectral irradiance models and ozone observations (e.g. Ball et al., 2016, https://doi.org/10.1038/ngeo2640) which, broadly speaking, support your conclusions or at least do not contradict them.*

We have cited the study of Ball et al. (2016) in a different Section, as it is not directly related to the solar response in total ozone.

*p21 line 20: The investigation of more recent periods, instead of the 1989-1994 comparison would allow to better constrain the SSI variability (with SOLAR-ISS as you mention, but also other observational datasets such as AURA-OMI) and better overcome the main source of uncertainty in the solar forcing, which comes from the FUV range.*

The motivation for selecting the solar amplitude 1989–1994 is now better motivated in Section 2. We have added a comparison of the selected solar amplitude with an averaged solar cycle amplitude over the satellite era, to show that it can serve as representative for recent solar cycles.

*Compared to other solar cycle amplitudes in the satellite era (see Table S1 in the supplement) the one used in the paper is neither especially weak nor especially strong. The averaged $\Delta SSI$ is shown in Figure 2a with the error bars indicating the 95% confidence interval of the $\Delta SSI$ within each spectral region. The main characteristics of the solar amplitude chosen for the paper are also present in the averaged solar cycle amplitude. These are the small solar amplitude of SATIRE-T in the FUV and most of the ranking of the SSI data sets within the spectral regions. The deviations of the solar cycle amplitude from the averaged solar cycle amplitude is shown in Figure 2b. All deviations are within the range of the 95% confidence intervals. Therefore the selected solar cycle amplitude can be regarded as representative for most of the solar cycle amplitudes of the satellite era.*

*p23 line 15: Most ANOVA studies focus on the F ratio although one could also consider the coefficient of multiple determination R (e.g. von Storch and Zwiers, p. 176) which, arguably, gives a better physical picture. Did you consider it?*

The F-statistics are used to estimate the probability (p value) that is plotted in Figure 1. The hatching denotes insignificant (i.e. p values larger than 0.05) differences between the solar responses when grouped according to CCM or SSI data set.

**Technical corrections**

*Title: since this is part 1, what should we expect to find in part 2? The manuscript does not really tell this.*

The study is entitled "Part I" and deals with annual mean quantities, as the "Part II" of the study is analysing the impacts on the dynamics especially in the winter seasons and also includes the impact of auroral forcing. Currently Part II, with the working title "Quantifying uncertainties of climate signals related to the 11 year solar cycle - Part II: Dynamical impacts of irradiance and auroral forcing", is in preparation. We have added the following sentence to Section 1:

*In this Part I of our study we concentrate on the annual mean solar response in heating rates, temperature and ozone, while Part II (in preparation) focuses on the dynamical solar and auroral responses in northern winter.*

*p4 line 27: the appropriate reference for the Bremen MgII is Snow et al., 2014 (https://doi.org/10.1051/swsc/2014001)*

The reference "Viereck et al. (2001)" is now replaced by "Snow et al. (2014)".

*p6 line 10: observations*

Done.

*p35 Legend of Fig 4: does not pass → do not pass*

Done.

**References**

Ball, W. T., Unruh, Y. C., Krivova, N., Solanki, S. K., and Harder, J. W.: Solar irradiance variability : a six-year comparison between SORCE observations and the SATIRE model, Astronomy & Astrophysics, 530, A71, doi:10.1051/0004-6361/201016189, 2011.

5    Ball, W. T., Haigh, J. D., Rozanov, E. V., Kuchar, A., Sukhodolov, T., Tummon, F., Shapiro, A. V., and Schmutz, W.: High solar cycle spectral variations inconsistent with stratospheric ozone observations, Nature Geoscience, 9, 206–209, doi:10.1038/ngeo2640, 2016.

Evin, G., Hingray, B., Blanchet, J., Eckert, N., Morin, S., and Verfaillie, D.: Partitioning Uncertainty Components of an Incomplete Ensemble of Climate Projections Using Data Augmentation, Journal of Climate, 32, 2423–2440, doi:10.1175/JCLI-D-18-0606.1, 2019.

Fisher, R.: Statistical methods for research workers, Edinburgh Oliver & Boyd, 1925.

10    Geinitz, S., Furrer, R., and Sain, S. R.: Bayesian multilevel analysis of variance for relative comparison across sources of global climate model variability, International Journal of Climatology, 35, 433–443, doi:10.1002/joc.3991, 2015.

Haigh, J. D., Winning, A. R., Toumi, R., and Harder, J. W.: An influence of solar spectral variations on radiative forcing of climate, Nature, 467, 696–699, doi:10.1038/nature09426, 2010.

Matthes, K., Funke, B., Andersson, M. M. E., Barnard, L., Beer, J., Charbonneau, P., Clilverd, M. A. M., Dudok De Wit, T., Haberreiter, M.,

15    Hendry, A., Jackman, C. H. C., Kretzschmar, M., Kruschke, T., Kunze, M., Langematz, U., Marsh, D. D. R., Maycock, A. A. C., Misios, S., Rodger, C. C. J., Scaife, A. A. A., Seppälä, A., Shangguan, M., Sinnhuber, M., Tourpali, K., Usoskin, I., Van De Kamp, M., Verronen, P. P. T., Versick, S., Sepp?l?, A., Shangguan, M., Sinnhuber, M., Tourpali, K., Usoskin, I., Van De Kamp, M., Verronen, P. P. T., Versick, S., Seppälä, A., Shangguan, M., Sinnhuber, M., Tourpali, K., Usoskin, I., Van De Kamp, M., Verronen, P. P. T., and Versick, S.: Solar forcing for CMIP6 (v3.2), Geosci. Model Dev., 10, 2247–2302, doi:10.5194/gmd-10-2247-2017, 2017.

20    Merkel, A., Harder, J. W., Marsh, D. R., Smith, A. K., Fontenla, J. M., and Woods, T. N.: The impact of solar spectral irradiance variability on middle atmospheric ozone, Geophysical Research Letters, 38, 1–6, doi:10.1029/2011GL047561, 2011.

Oberländer, S., Langematz, U., Matthes, K., Kunze, M., Kubin, A., Harder, J., Krivova, N. A., Solanki, S. K., Pagaran, J., and Weber, M.: The influence of spectral solar irradiance data on stratospheric heating rates during the 11 year solar cycle, Geophys. Res. Lett., 39, doi:10.1029/2011GL049539, http://dx.doi.org/10.1029/2011GL049539, L01801, 2012.

25    Swartz, W. H., Stolarski, R. S., Oman, L. D., Fleming, E. L., and Jackman, C. H.: Middle atmosphere response to different descriptions of the 11-yr solar cycle in spectral irradiance in a chemistry-climate model, Atmos. Chem. Phys., 12, 5937–5948, doi:10.5194/acp-12-5937-2012, 2012.

von Storch, H. and Zwiers, F. W.: Statistical Analysis in Climate Research, Cambridge University Press, doi:10.1017/CBO9780511612336, 1999.

30    Zhong, W., Osprey, S. M., Gray, L. J., and Haigh, J. D.: Influence of the prescribed solar spectrum on calculations of atmospheric temperature, Geophys. Res. Lett., 35, L22 813, doi:10.1029/2008GL035993, 2008.

---

## Author Comment (AC3) · 8 May 2020

**Answer to anonymous Referee #2**

We thank the Referee for her/his comments and suggestions. We answer point by point in the following with the Referee's comments added in *red/italics*. Text added to the revised version of the manuscript is included here in *blue/italics*.

**General comments**

*The subject of this paper, i.e., identifying and quantifying sources of uncertainty in modeled atmospheric response to 11-year variations in SSI, is compelling and the basic tools (2 sets of CCM simulations, SSI data sets, application of well-established ANOVA statistical methods) are appropriate. The use of a common reference SSI data set for solar minimum and application of relative changes in SSI informed by the different data sets makes sense. There are several areas (identified below) where the present manuscript does not provide enough information to allow the reader to understand this work and its implications. These areas can be summarized as follows: (1) Context – why did the authors undertake this study, and what solid, quantitative conclusions does this study offer that will be of use to other researchers; (2) The advantages and limitations of the ANOVA approach – the limitations in particular need to be clarified; (3) Presentation – some figures and some of the discussion were difficult to understand, some reorganization and revision is warranted to improve the overall readability of the manuscript. These areas should be addressed in a revised manuscript before I can recommend publication.*

**Major Comments**

*1. The title of the manuscript is not very descriptive. It should probably be stated somewhere in the title that this is a modeling study. One could read this title and think this is an observational study of variations in climate (e.g., surface temperatures, precipitation patterns, etc.), rather than a very specific examination of how sensitive middle atmospheric processes in CCMs are to imposed variations in SSI related to the 11-year solar cycle. The title also says this is Part I, but I do not understand why this is so. What will part II be about, and why does this need to be a two-part study?*

We changed the title. It now explicitly includes a hint to CCMs, to avoid misunderstanding.

*Quantifying uncertainties of climate signals in Chemistry Climate Models related to the 11–year solar cycle. Part I: Annual mean response in heating rates, temperature and ozone*

The study is entitled "Part I" and deals with annual mean quantities, while "Part II" of the study is analysing the impacts on the dynamics especially in the winter seasons and also includes the impact of auroral forcing. Currently Part II, with the working title "Quantifying uncertainties of climate signals related to the 11 year solar cycle - Part II: Dynamical impacts of irradiance and auroral forcing", is in preparation. We have added the following sentence to Section 1:

*In this Part I of our study we concentrate on the annual mean solar response in heating rates, temperature and ozone, while Part II (in preparation) focuses on the dynamical solar and auroral responses in northern winter.*

*2. The authors have undertaken a very ambitious task requiring a lot of detailed statistical analysis. After reading the introduction, it is still unclear to me why this study is being performed.*

There are numerous modelling studies dealing with the 11-year solar cycle response, as mentioned in the introduction. All these studies rely on one specific SSI data set (not necessarily the same one) and employ different CCMs. The motivation for our

study is (i) to provide a robust estimate of 11-year solar cycle responses as derived as the ensemble mean after employing two different CCMs and five different SSI datasets and (ii) to provide quantitative estimates of the relevance of these two factors (regarding the uncertainty) in this context. The following paragraph is added to the Introduction.

*The SSI data prescribed in the models are the second source of uncertainty when modelling the solar response. Shapiro et al. (2011) investigated the influence of the 27-day variations of four different SSI observations on the chemistry of the upper mesosphere in a 1D radiative-convective chemistry model. The deviant solar cycle behaviour of the SORCE (Solar Radiation and Climate Experiment) measurements has motivated a number of CCM studies (e.g. Haigh et al., 2010; Merkel et al., 2011; Ball et al., 2011, 2016; Swartz et al., 2012) comparing simulations using prescribed SORCE SSI data with reconstructed SSI of the Naval Research Laboratory (NRLSSI) or the Spectral And Total Irradiance REconstructions (SATIRE) model.*

*The authors state (page 2 line 8) that we have a good understanding of the chemical and dynamical processes, but discrepancies between the observed and modeled responses remain? What is not stated is how big these discrepancies are, and why they are important within the larger context of climate modeling.*

We have added the following sentence to the Introduction.

*The variability induced by the 11-year solar cycle SSI and TSI variations is part of the natural variability of the climate system. Besides the ability of GCMs and CCMs to model the right climatological state of the atmosphere and their chemical species, it is also an important aspect of climate models to realistically reproduce this natural variability.*

*Some more quantitative discussion of these discrepancies in the introduction are needed. For example, page 2 line 25 states there is "large model spread" – how large is this and why is it important? Is this spread larger than uncertainties in observational-based estimates of 11-year variations related to solar forcing?*

A quantitative statement about the range in simulated solar responses is now included.

*The simulated solar response in annual mean tropical ($25°S$–$25°N$) temperature (1960–2004) near the stratopause ranges from 0.45 to 1.4 K, whereas the SSU satellite data (1979–2005) show 0.85 K for a comparable height region, and ERA-40 reanalyses (1979–2001) show 1.4 K. The annual mean solar response in ozone mixing ratio for the same region and time frame shows less model spread with an ozone increase of 2% in the upper stratosphere, which is in good agreement with observations. Towards lower altitudes the model spread increases and discrepancies to the observations get larger (SPARC CCMVal, 2010).*

*3. The discussion of the "top down" vs. "bottom up" mechanisms (page 2 lines 13-25) should be condensed and revised to clearly state that this study is focusing entirely on the "top down" effect. The "bottom up" effect relies not only on changes in TSI but also on a very complex interaction between ocean and atmosphere, and it should be stated that the present study cannot address this mechanism with the model simulations presented here.*

The nature of an introduction section should be to introduce the reader into the broad scope of the topic (in this case the response of CCMs to the 11-year solar variability and its uncertainty). To include other pathways of the solar influence is therefore justified. The "bottom up" mechanism is only briefly introduced to distinguish from the more important "top down" effect for completeness, and no link is made here to the goals of our study.

*4. The ANOVA method finds the largest uncertainties in the upper mesosphere, but I don't think any of the observational studies cited in the Introduction deal specifically with the upper mesosphere. From what's presented in the manuscript, it's unclear*

*how quantifying these upper mesospheric uncertainties are directly relevant to improving our understanding of climate signals.*

Also taking into account your comment "3." in the section "Additional comments, revisions, suggestions" we have added a paragraph to the introduction, dealing with studies of the 11-year solar response in the mesosphere.

*Model intercomparions as CCMVal-2, CCMI, and CMIP5 focus on the solar response in the troposphere and stratosphere. Higher up in the mesosphere where shorter wavelength are not absorbed yet the irradiance variations over the 11-year solar cycle are even larger and have a large effect on atmospheric trace gases like $H_2O$ and $CO_2$, producing large solar responses in $HO_x$, CO and also effecting $O_3$ by subsequent catalytic cycles. These effects in the mesosphere are analysed by a number of modelling studies (e.g. Marsh et al., 2007; Merkel et al., 2011; Beig et al., 2012).*

*Looking at Figure 2, it appears that in the stratosphere (where most of the ozone resides and where this "top-down" mechanism dominates), the details of the SSI input don't really matter – you get essentially the same modeled response (excluding SATIRE-T), i.e., any differences are smaller than the internal model variability. If this is the case, this should be clearly stated in the abstract and in Section 7.*

This is correct and it is now explicitly stated in the abstract and the summary/conclusions section.

Added to abstract:

*However, in the region of the largest ozone mixing ratio, in the stratosphere from 50 to 10 hPa, the SSI data sets do not contribute much to the variability of the solar response when the Spectral And Total Irradiance REconstructions-T (SATIRE-T) SSI data set is omitted.*

Added to summary/conclusions section:

*In the region of largest ozone mixing ratio, in the stratosphere from 50 to 10 hPa, the SSI data sets do not contribute much to the variability of the solar response when SATIRE-T is omitted.*

*5. Appendix A describes the ANOVA method. I am not an expert in this field, so my comments here are for clarification rather than criticism.*
*a. My understanding is that part of the ANOVA approach is to construct a model describing the sources of variance, making certain assumptions about what these sources are, and then testing this model to see how much of the variance is explained. The model should be described and listed in equation form – is it a two way model with interaction?*

The statistic model describing the solar responses is now given in equation form in the appendix (new equations A1 and A2). The interaction term, describing the variances that are emerging by the interaction of the two treatments (CCM and SSI data set), is included in equation A3.

*The anomalies $x_{ijk}$ are the individual solar responses, and their fluctuations around the averaged solar response $\overline{x}$ can be described as*

$$x_{ijk} = \overline{x} + (\overline{x_i} - \overline{x}) + (\overline{x_j} - \overline{x}) + (\overline{x_{ij}} - \overline{x_i} - \overline{x_j} + \overline{x}) + \epsilon_{ijk}$$
$$= \overline{x} + \alpha_i + \beta_j + \gamma_{ij} + \epsilon_{ijk}$$

*with $\alpha_i$ the deviations of the averages of the two CCMs from the overall average $\overline{x}$; $\beta_j$ the deviations of the averages of simulations applying the five SSI data sets from $\overline{x}$; $\gamma_{ij}$ the deviations of the averages of the individual simulations $\overline{x_{ij}}$, $\overline{x_i}$, and $\overline{x_j}$ from $\overline{x}$; and $\epsilon_{ijk}$ the random fluctuations that cannot be explained by $\alpha_i$, $\beta_j$, or $\gamma_{ij}$. The null hypotheses ($H_0$) are that the averaged*

*solar response does not depend on the CCM ($H_{0,1} : \alpha_i = 0$), the applied SSI data set ($H_{0,2} : \beta_j = 0$), or the interaction of CCM with applied SSI data set ($H_{0,3} : \gamma_{ij} = 0$). We apply the two-way ANOVA to test the validity of these hypotheses.*

*Is there a specific term for variance from random error?*

The term describing the variance from random error is also included in equation A4 as $SS_w$.

$$SS_t = SS_{bA} + SS_{bB} + SS_{bAB} + SS_w.$$

*b. With regard to figure 1, center and right columns, it's clear that not all the variance is being explained by the SSbB (center) and SSbA (right) terms. This is alluded to on page 10 line 1-2, where the authors state that the random contribution is largest.*

*c. For a complete description of the problem, would it be better to limit figure 1 to the annual mean responses, and construct a new figure 2 listing all terms of the ANOVA model so we can see all relevant terms (e.g. the treatment A term treatment B term and the interaction term)?*

15 As recommended by the Referee, we have divided Figure 1 into a part showing only the averaged solar responses (new Figure 3) and a part showing the results of the ANOVA, including also the part of the variance that can be explained by the interaction of the two treatments (now Figure 4).

*Compared to $R_{a,A}$ and $R_{a,B}$ the fraction of explained variance by the interaction of SSI data set and CCM ($R_{a,AB}$) is only*
20 *small. Some significant differences of the solar responses in SW heating rates and ozone are explained by the interaction in the upper mesosphere, and near the stratopause for temperature.*

*It would be most helpful if the description of current Figure 1 middle and right columns referred directly to the terms and equations in the Appendix so we know for sure what is being plotted, i.e. middle column is SSbB term equation A3, etc. Also*
25 *the hatched areas in the middle column for SSI/temperature and SSI/ozone are extremely hard to see, making it difficult to understand what is and isn't significant.*

We revised the discussion of the ANOVA results. The terminology of the Appendix is now explicitly used.

30 *d. Please explain in more detail how degrees of freedom were determined. The CMIP6 data set is an average of NRLSSI2 and SATIRE data sets, so it's not an independent member of the K=B group. Shouldn't this affect the degrees of freedom that ultimately impact the significance tests with the F statistic?*

The degrees of freedom are calculated as required for a two-way ANOVA. This is now described in the appendix in more
35 detail. We agree with the Referee on the problem that the CMIP6 SSI data set is not an independent SSI data set, as it is build by averaging the NRLSSI2 and SATIRE-S SSI data sets. When decreasing the degrees of freedom for the group K=B by 1 ($df_{nB} = 3, df_{nAB} = 3$) the statistical significance is affected. We now have set $df_{nB} = 3, df_{nAB} = 3$ for the revised version, but the differences to the previous calculations are only marginal.

40 *The degrees of freedom of the model are calculated as*

$$df_d = n_t - N_A N_B$$
$$df_{nK} = N_K - 1$$
$$df_{nAB} = (N_A - 1)(N_B - 1).$$

*These are the degrees of freedom within the groups ($df_d$) which is the degrees of freedom of the denominators when calculating the F-statistics (Equation A15), $df_{nK}$ the degrees of freedom between the groups, and $df_{nAB}$ the degrees of freedom of the interaction between the treatments which are the degrees of freedom of the nominators in Equation A15. Note that the CMIP6 SSI data set depends on the SATIRE-S and NRLSSI2 data sets and therefore cannot be counted as an independent SSI data set.*

5    *For the calculations of the degrees of freedom we use $N_B = 4$.*

*e. Outside of the upper mesosphere, it seems like the SSI changes and CCM differences together don't explain the majority of the variance in the total sum of squares. What does this mean? Is this analysis meaningful? Should we conclude that differences in SSI reconstructions or differences in details of model photochemistry or spectral resolution in the SW heating aren't that*

10    *important relative to the random model variability?*

We are convinced that it is still meaningful to do the two-way ANOVA even if the majority of the variance is caused by randomness. There are large regions outside the upper mesosphere where a significant part of the variances can be explained by the usage of different SSI data sets of the different CCMs, even if the majority of the variance is due to randomness.

15

*6. Section 7 needs revision as noted below:*

*a. First page 19 lines 12-25 repeat what has already been said in the introduction and could be removed or condensed significantly.*

20

Due to the nature of a summary section, some recurrences can not be avoided.

*b. Page 19 line 17: it is stated that SSI data sets provide largest fraction of solar cycle variance in the upper stratosphere/lower mesosphere (30% for heating rate, 30% for ozone, 10 % for temperature) but that is not strictly true. The majority of the*

25    *variance is unexplained, wrapped up in an interaction term or some other manifestation of random model variability. I think it would suffice to say that the SSI differences explain up to 30% in variance in heating and ozone, and only 10% in temperature.*

We agree with the Referee and have added the following paragraph in Section 7.

30    *However, in the upper stratosphere/lower mesosphere the largest fraction of solar cycle response variance is random and not related to differences in SSI data sets or the applied CCM.*

*c. Page 21 lines 5-13: The discussion of the total ozone effects is confusing, and does not seem to produce any specific conclusions. The sentence "Distinct differences in TCO anomalies between the CCMs are also reflected by the relatively large*

35    *fraction of the anomaly variability that can be explained by differences between the CCMs" seems circular and it's not clear to me what the authors are trying to say.*

The ANOVA of TCO shows the largest contributions of the CCM treatment to the explained variability of the anomaly in regions with systematic differences between the CCM's TCO anomalies (i.e. in high southern latitudes and in northern mid-

40    latitudes).
The word "reflected" is now replaced by "expressed".

*The finding that WACCM and EMAC models have lower percentage of TCO response from $p > 16$ hPa compared to one observational study (Hood 1997) does not seem to be directly relevant to the state purpose of this study, especially since by design*

45    *these CCM simulations do not have realistic decadal variations in lower atmosphere forcing (i.e., fixed repeating monthly*

*mean SST's, etc).*

We agree with the reviewer that we miss a part of the variability without transient forcings. But even though the simulations do not capture the full scope of natural variability, we are convinced to receive ANOVA results of the TCO and partial column ozone that are relevant within the scope of this study. Our main purpose, to quantify the part of the variability of the solar response that can be explained by SSI data set or CCM, is also achievable with the time slice setup of our study.

*It's clear from Figure 8 that the ANOVA method cannot disentangle variance related to SSI and model transport in the lower stratosphere.*

That the SSI data sets have nearly no effect on the lower stratospheric ozone column is also a result worth showing.

*Based on what's presented in this paper, it would make sense to keep Figure 6, omit Figures 7 and 8 and related discussion, and summarize your findings (i.e., that statistically significant attribution of TOC variance related to SSI or CCM differences as you've defined them is not possible due to large internal model variability).*

As justified in our responses to the two preceding remarks, we want to keep the related material in the manuscript.

*d. Page 21 lines 14-19: Based on the discussion here, it's not clear why a two-way ANOVA approach is warranted compared to a 1-way (SSI changes) approach. Basically, you are saying you don't think you have fully sampled the "CCM spread" as it is referred to here. So why is 2-way justified? This might be a good place to note the importance of experimental design when using ANOVA that could help guide future investigations.*

We believe that it is useful to apply a two-way ANOVA with only two CCMs, as we do not demand to draw a general conclusion with respect to the influence of CCMs on the solar response. The null hypotheses of the ANOVA are that there are no significant influences by the usage of five different SSI data sets in the CCM simulations (i.e. by treatment $K = A$) or by the two CCMs (i.e. by treatment $K = B$) or the interaction of both treatments. This study was initiated within a national research project where two project partners could apply either EMAC (FUB) or WACCM (GEOMAR), thus we were forced to limit our analysis to two CCMs. Although, ideally it would be desirable to apply a wide range of CCMs, this option is quite unrealistic within a national research project.

**Additional comments, revisions, suggestions**

*1. Abstract: It should be noted somewhere in abstract that you are using time slice integrations based on 1989-1994 differences in SSI.*

The experimental set up is now described in more detail in the abstract:

*The solar response is derived from climatological differences of time slice simulations prescribing SSI for the solar maximum in 1989 and near the solar minimum in 1994. The SSI values for the solar maximum of each SSI data set are created by adding the SSI differences between November 1994 and November 1989 to a common SSI reference spectrum for near solar minimum conditions based on ATLAS-3 (Atmospheric Laboratory of Applications and Science-3).*

*2. Page 1 line 16: can you define middle atmosphere?*

We have rephrased this sentence slightly and skipped the term 'middle atmosphere'.

5  *Solar ultraviolet (UV) radiation is largely absorbed in the stratosphere and mesosphere, thereby heating these regions and forming the ozone layer, filtering the most harmful part out of the solar spectrum and protecting life on Earth.*

*3. Page 2 line 3: the authors cite one reference here (McCormack and Hood), but there are a lot of subsequent studies on observed solar cycle variations that should also be referenced.*

10  Additional references are now included: Soukharev and Hood, 2006; Randel and Wu, 2007; Maycock et al., 2016; Ball et al., 2019.

*As mentioned above, observational studies for the mesosphere in particular would be good, since this is where you end up*
15  *seeing the biggest impact of SSI differences. For example, Beig at al JGR 2012 (https://doi.org/10.1029/2011JD015697).*

The study of Beig et al. (2012) is now cited in the new paragraph included in the introduction, discussing modelling studies of the solar responses in the mesosphere. We have added:

20  *Model intercomparions as CCMVal-2, CCMI, and CMIP5 focus on the solar response in the troposphere and stratosphere. Higher up in the mesosphere where shorter wavelength are not absorbed yet the irradiance variations over the 11-year solar cycle are even larger and have a large effect on atmospheric trace gases like $H_2O$ and $CO_2$, producing large solar responses in $HO_x$, CO and also effecting $O_3$ by subsequent catalytic cycles (e.g. Marsh et al., 2007; Merkel et al., 2011; Beig et al., 2012).*

[Figure]

(a) Solar Cycle 22 (descent-2)          (b) Broad bands Solar Cycle 22 (descent-2)

**Figure 1.** (a) Solar cycle SSI variations from Nov. 1989 to Nov. 1994 relative to Nov. 1994 ($\Delta SSI$) in % for wavelengths ranging from FUV to the VIS bands; (b) as in (a) for the Lyman-$\alpha$ (121.5 nm), Far-UV (121–200 nm), Herzberg continuum/Hartley bands (201–242 nm), Hartley-/Huggings-bands (243–380 nm) (multiplied by a factor of 10) and visible (381–780 nm) (multiplied by a factor of 100).

*4. Section 2: It would be most helpful to have a plot comparing the different SSI data sets somehow. Is it possible to plot the SSI differences relative to the ATLAS solar min values over a range of UV wavelengths. This would illustrate for the reader how differences among the different data sets compare to the overall 11-year max-min differences. Since the change in SSI from solar max – min is strongly wavelength dependent, this might be more informative than Table 1 that averages over very large intervals.*

Figure 1 is now included in the manuscript, showing the SSI changes from November 1989 to November 1994 relative to Nov. 1994 for wavelengths ranging from 120–780 nm (new Figure 1a) and Figure 1b showing $\Delta SSI$ in % as in Table 1. There is not much new information added by this Figure, as the $\Delta SSI$ of the integrals over broad bands in Table 1 summarize the main message, but we agree with the Referee that a visualisation can highlight the differences among the SSI data sets.

*5. Section 3.1: Why are the QBO treatments different? What observed winds are used for the relaxation in EMAC and for what period of time?*

We have to admit that the different treatment of the QBO in the two models is the result of a lack in coordination. However, there is a (weak) reason why the QBO-treatment is different for the two models for the specific type of experiments as analysed in our study. As a reminder, we performed timeslice-experiments for year 2000 forcings. For EMAC this was done by starting the simulations in model year 1995 and running them towards model year 2045. For the relaxation of the zonal wind in the equatorial, lower stratosphere in EMAC the QBO data set of the FU-Berlin is used that is based on radiosonde observations at stations near the Equator (Canton Island, Gan, Singapore). The observed QBO time series is used from January 1995 to November 2011 of the 50 years EMAC time slices. The time period from December 2011 to December 2044 is covered by subsequently adding compatible segments of the observed QBO. For WACCM, the standard setup for this type of timeslice experiment starts simulations in model year 1. This was done for our study, too. We integrated the model over model years 1-48. Regarding the QBO-nudging it would have been possible to use a representation of the observed QBO. WACCM offers the possibility to do so for any simulation period via spectral coefficients, essentially repeating the observed QBO-timeseries forward and backward in time. Given the generally idealized nature of our timeslice simulations, the modeling group responsible for the WACCM-experiments refrained from using this approach (basically extrapolating a QBO-timeseries almost 2000 years back in time) and made use of the WACCM default procedure for timeslice simulations that is using an idealized QBO with a fixed period of 28 months as described in the cited study of Matthes et al., 2010. We agree that it would have been better to properly coordinate this element of our experiments, avoiding this difference between EMAC- and WACCM-simulations. However, most importantly, the identical QBO-nudging has been used for the respective solarmin- and solarmax-experiments of each of the two models. Hence, the differences between solarmin and solarmax which are further analysed in our study are purely a result of differing solar activity.

*In doing ANOVA, experimental design is very important. Were these CCM simulations designed and performed especially for this study, or is this study using simulations that were generated previously. This could be helpful to note in the paper. If these were simulations already generated, this paper is more of a proof of concept on how to apply ANOVA and how perhaps future multi-model CCM experiments should be designed in order to best use the ANOVA method.*

All EMAC and WACCM simulations in this paper have been made explicitly for this study. The intention was to apply a setup as close as possible for both the EMAC and WACCM simulations. As explained above, with respect to the QBO-nudging the setups are slightly different, as explained above. We have added a short statement to the introduction to Section 3.1.

*All EMAC and WACCM simulations have been made explicitly for this study, to ensure that the differences in the solar responses are exclusively related to the SSI data set prescribed or the CCM applied, and not due to differences in the scenario.*

*6. Figure 4 and related discussion in the text could be removed. In its present form, it doesn't add much information, especially since I'm not sure how much data ERA5 uses in the upper stratosphere/mesosphere, meaning the ERA5 fields could be very model-dependent themselves, and not the best standard to compare with.*

We disagree with the referee on this point, as a comparison of modelling results with a common base can always be helpful to identify potential problems. Comparing to ERA5, the cold bias of EMAC in the tropical UTLS that affects water vapour in the stratosphere and mesosphere can be identified.

*It might be more illuminating to directly compare differences in zonal mean T, zonal wind, and ozone between WACCM and EMAC.*

For a direct comparison of zonal mean T and ozone please see Figure 5b and 5c. The comparison of the zonal mean zonal wind is omitted, as this quantity is not in the scope of this paper.

*7. Page 9 line 33 – I really can't see the grey hatching in Figure 1 very well. Is it possible to plot it another way? Maybe only plot significant values?*

The grey hatching is now changed to cross hatches to strengthen the presentation of the non-significant regions.

*8. Page 11, line 9: it is stated that the t test is and resulting error bars come from the complete ensemble but Figure 2 caption states the error bars are for the WACCMX/EMAC CMIP6 simulations. Which is it?*

We thank the referee to point to this inconsistency. The information in the Figure caption was outdated. The description in the main text is correct. The Figure caption is now updated.

*The 95% uncertainty error bar is given for the averaged solar response over the complete ensemble, calculated with a Student's t test.*

*9. Page 12: I'm not sure it's worth reviewing Chapman cycle photochemistry here. If the authors wish to describe specific reactions in detail, I would suggest using equation form rather than in the text, and perhaps put some of the more complex reaction in an appendix?*

Chapman cycle photochemistry is discussed here, as the differences in the SSI amplitudes of the adequate UV spectral regions are directly related to the photochemical reactions involved.

**References**

Ball, W. T., Unruh, Y. C., Krivova, N., Solanki, S. K., and Harder, J. W.: Solar irradiance variability : a six-year comparison between SORCE observations and the SATIRE model, Astronomy & Astrophysics, 530, A71, doi:10.1051/0004-6361/201016189, 2011.

Ball, W. T., Haigh, J. D., Rozanov, E. V., Kuchar, A., Sukhodolov, T., Tummon, F., Shapiro, A. V., and Schmutz, W.: High solar cycle spectral variations inconsistent with stratospheric ozone observations, Nature Geoscience, 9, 206–209, doi:10.1038/ngeo2640, 2016.

Beig, G., Fadnavis, S., Schmidt, H., and Brasseur, G. P.: Inter-comparison of 11-year solar cycle response in mesospheric ozone and temperature obtained by HALOE satellite data and HAMMONIA model, J. Geophys. Res.: Atmos., 117, doi:10.1029/2011JD015697, 2012.

Haigh, J. D., Winning, A. R., Toumi, R., and Harder, J. W.: An influence of solar spectral variations on radiative forcing of climate, Nature, 467, 696–699, doi:10.1038/nature09426, 2010.

Marsh, D. R., Garcia, R. R., Kinnison, D. E., Boville, B. A., Sassi, F., Solomon, S. C., and Matthes, K.: Modeling the whole atmosphere response to solar cycle changes in radiative and geomagnetic forcing, Journal of Geophysical Research, 112, D23 306, doi:10.1029/2006JD008306, 2007.

Merkel, A., Harder, J. W., Marsh, D. R., Smith, A. K., Fontenla, J. M., and Woods, T. N.: The impact of solar spectral irradiance variability on middle atmospheric ozone, Geophysical Research Letters, 38, 1–6, doi:10.1029/2011GL047561, 2011.

Shapiro, A. V., Rozanov, E., Egorova, T., Shapiro, A. I., Peter, T., and Schmutz, W.: Sensitivity of the Earth's middle atmosphere to short-term solar variability and its dependence on the choice of solar irradiance data set, Journal of Atmospheric and Solar-Terrestrial Physics, 73, 348–355, doi:10.1016/j.jastp.2010.02.011, 2011.

Swartz, W. H., Stolarski, R. S., Oman, L. D., Fleming, E. L., and Jackman, C. H.: Middle atmosphere response to different descriptions of the 11-yr solar cycle in spectral irradiance in a chemistry-climate model, Atmos. Chem. Phys., 12, 5937–5948, doi:10.5194/acp-12-5937-2012, 2012.

---

## Author Comment (AC4) · 8 May 2020

**Answer to anonymous Referee #3**

We thank the anonymous Referee for her/his comments and suggestions. We answer point by point in the following with the Referee's comments added in *red/italics*. Text added to the revised version of the manuscript is included here in *blue/italics*.

*I find the study of high interest and worthy of publication. I do provide some comments below for consideration. In brief, these focus firstly on the solar irradiance dataset aspect of the study. I find the adoption of a common spectrum from which to baseline differences in the solar cycle 'amplitudes' of the various datasets novel. Secondly, the messaging of the study could be improved to provide the background of why the study was undertaken.*

**SSI comments**

*a. The general reader may be unclear why it was necessary to use 5 different SSI datasets, or why you chose the ones you did, or even that there isn't agreement across SSI datasets on longer time scales (observed or modeled). I would suggest adding a paragraph or two to improve the messaging behind your study, probably in the Intro or in Section 2.*

We have chosen these five SSI data sets, as they are available for long time periods. Therefore they are of special interest for climate modellers in simulations that cover these long time periods. We have added this sentence to the introduction of Section 2.

*The SSI/TSI reconstructions of SATIRE, NRLSSI/TSI, and the combination of both in the CMIP6 SSI/TSI data set, are the most common SSI/TSI data sets used in GCMs and CCMs and, therefore, subject to our investigation.*

*b. I agree that TSI observations are relatively short (since 1978) and that SSI observation record is even shorter, nor full spectral coverage, and has time gaps. However, your study does select a relative short period of time to investigate the impacts of SSI over (1989-1994). Therefore, it begs the questions of why that particular time range and not another when full spectrum observations existed (i.e. during the SORCE era) or even partial spectrum observations (265-500 nm) by the AURA OMI instrument. In essence, I'm asking you to more directly draw the line between your "focused" study and the SSI dataset needs of the model intercomparisons studies like CMIP6 which require full spectrum and very long time coverage. This leads to necessary use of modeled SSI datasets, which have differences between them and with observations. It would be helpful to bring the discussion of the Coddington et al and Yeo et al. results (Page 6, line 7 through end of paragraph) in earlier in the section for this reason.*

We have moved the paragraph to the introduction of Section 2.

*However, there is an ongoing debate about the reliability of TSI and SSI reconstructions. Coddington et al. (2019) compare solar amplitudes of 11–year solar cycles in the satellite period produced with the NRLSSI2 and SATIRE-S for a number of broad wavelengths bands to SSI amplitudes derived from the SOLID composite. In the FUV spectral region they report the highest SSI amplitude for the SATIRE-S data set and a negative secular interminima trend over the satellite period in the SATIRE-S TSI and SSI from the FUV to NIR spectral regions which is not present in any observational record or other TSI/SSI reconstructions. Yeo et al. (2015) compare the SSI variability of NRLSSI1 and SATIRE-S with SSI observations over the satellite period and report the low UV variability of NRLSSI compared to the SATIRE-S data set, whereas the latter is in better agreement to the satellite SSI observations.*

*c. I do like that you've chosen a single spectrum to adopt as a common baseline for solar minimum conditions. I feel that's quite novel. I am concerned, though, that the manuscript doesn't adequately address how this approach might impact results. You do say that a reference baseline would lead to a certain climatology state (end of page 13 to page 14) and that differences from that baseline, as would occur from using SOLAR-ISS as the reference, would result in a different climatology. However, is it necessarily true that the solar response variations are truly linear from an adopted baseline? Maybe more clear way to ask is whether gas phase reaction rates or water vapor abundances that you mention on page 14 might "bottom out" or "max out" if the baseline climatology/temperature was too high or too low? I would also suggest bringing this discussion up earlier, in addition to where it is in the conclusions.*

It is true that the adaptation of atmospheric processes and gas phase chemical reactions to a different reference spectrum or a different solar cycle amplitude is not necessarily linear. However, the differences in the reference spectrum or the solar cycle amplitude are relatively small and the expected changes in temperature should not be that large to reach "bottom out" or "max out" effects in the chemistry.

We have moved the discussion about alternative reference spectra and different solar cycles to Section 2. The respective part in Section 7 is shortened.

*d. In conclusions you also discuss how choosing SC 22 (selected, I understand, because of time range of ATLAS 3 observations) should be reflective of other solar cycles in the 21st century. You examined the irradiances in the Lyman alpha through UV for the various SSI datasets with other solar cycles and found a linear relationship. Was that relationship with TSI magnitude, sunspot number, or something else?*

The nearly linear relationship of the SSI solar cycle amplitude applied in the paper with other solar cycle amplitudes refers to the wavelength dependency of the scaling factor which can be used to convert one solar cycle amplitude to another. Within one SSI data set there exist a nearly linear scaling factor for wavelengths from 121 to 280 nm among different solar cycle amplitudes. We agree that this sentence is confusing and therefore have removed it.

To better classify the solar amplitude used in the paper, we compare it with other possible solar amplitudes. The Table 1 (Table S1 in the supplement) includes the $\Delta SSI$ and $\frac{\Delta SSI}{\Delta TSI}$ of other solar cycle amplitudes. The average and the standard deviation over these solar cycles is given in Table 2 (Table S2 in the supplement). We have added the following paragraph to Section 2. The Figure 1 is now included in the manuscript as the new Figure 2.

*Compared to other solar cycle amplitudes in the satellite era (see Table S1 in the supplement), the one used in this study is neither especially weak nor especially strong. The averaged $\Delta SSI$ is shown in Figure 2a, with the error bars indicating the 95% confidence interval of the $\Delta SSI$ within each spectral region. The main characteristics of the solar amplitude chosen here are also present in the averaged solar cycle amplitude, such as the small solar amplitude of SATIRE-T in the FUV and most of the ranking of the SSI data sets within the spectral regions. All deviations of the chosen solar cycle amplitude from the averaged solar cycle amplitude are within the range of the 95% confidence intervals (Figure 2b). Therefore, the selected solar cycle amplitude can be regarded as representative for most of the solar cycle amplitudes of the satellite era.*

**Table 1.** Solar cycle spectral solar irradiance variations for Solar Cycles indicated in the first row relative to ATLAS3 ($\Delta SSI$) in % and relative contribution of SSI changes to the TSI change ($\frac{\Delta SSI}{\Delta TSI}$) in % for the Lyman-$\alpha$ (121.5 nm), Far-UV (121–200 nm), Herzberg continuum/Hartley bands (201–242 nm), Hartley-/Huggings-bands (243–380 nm) and visible (381–780 nm) spectral ranges.

| Time period | SSI dataset | 121.5 nm $\Delta SSI$ | $\frac{\Delta SSI}{\Delta TSI}$ | 121–200 nm $\Delta SSI$ | $\frac{\Delta SSI}{\Delta TSI}$ | 201–242 nm $\Delta SSI$ | $\frac{\Delta SSI}{\Delta TSI}$ | 243–380 nm $\Delta SSI$ | $\frac{\Delta SSI}{\Delta TSI}$ | 381–780 nm $\Delta SSI$ | $\frac{\Delta SSI}{\Delta TSI}$ |
|---|---|---|---|---|---|---|---|---|---|---|---|
| Cycle 21 descent Max:Dec.1979 Min:Sep.1986 | NRLSSI1 | 45.602 | 0.214 | 11.373 | 0.913 | 3.599 | 4.300 | 0.311 | 20.748 | 0.109 | 55.521 |
| | NRLSSI2 | 51.609 | 0.228 | 11.668 | 0.882 | 3.360 | 3.778 | 0.402 | 25.251 | 0.095 | 45.247 |
| | SATIRE-T | 47.283 | 0.199 | 10.081 | 0.724 | 3.445 | 3.684 | 0.592 | 35.363 | 0.106 | 48.028 |
| | SATIRE-S | 47.178 | 0.281 | 9.907 | 1.008 | 2.987 | 4.523 | 0.477 | 40.368 | 0.069 | 44.130 |
| | CMIP6 | 49.405 | 0.250 | 10.788 | 0.932 | 3.175 | 4.080 | 0.440 | 31.585 | 0.082 | 44.785 |
| Cycle 22 ascent Max:Nov.1989 Min:Sep.1986 | NRLSSI1 | 50.740 | 0.250 | 12.673 | 1.067 | 3.994 | 5.006 | 0.319 | 22.315 | 0.104 | 55.470 |
| | NRLSSI2 | 56.657 | 0.273 | 12.808 | 1.053 | 3.668 | 4.488 | 0.407 | 27.845 | 0.081 | 42.196 |
| | SATIRE-T | 44.322 | 0.247 | 9.454 | 0.899 | 3.235 | 4.580 | 0.529 | 41.801 | 0.075 | 45.107 |
| | SATIRE-S | 59.989 | 0.331 | 12.634 | 1.191 | 3.756 | 5.269 | 0.576 | 45.149 | 0.070 | 41.540 |
| | CMIP6 | 58.338 | 0.300 | 12.722 | 1.117 | 3.715 | 4.855 | 0.492 | 35.938 | 0.076 | 42.139 |
| Cycle 22 descent-2 Max:Nov.1989 Min:Nov.1994 | NRLSSI1 | 44.286 | 0.266 | 11.067 | 1.137 | 3.482 | 5.324 | 0.268 | 22.913 | 0.084 | 54.902 |
| | NRLSSI2 | 50.377 | 0.291 | 11.388 | 1.125 | 3.257 | 4.788 | 0.354 | 29.039 | 0.066 | 41.415 |
| | SATIRE-T | 35.572 | 0.297 | 7.576 | 1.081 | 2.583 | 5.486 | 0.407 | 48.288 | 0.047 | 42.340 |
| | SATIRE-S | 57.481 | 0.329 | 12.090 | 1.183 | 3.601 | 5.244 | 0.552 | 44.874 | 0.068 | 42.130 |
| | CMIP6 | 53.943 | 0.309 | 11.741 | 1.149 | 3.431 | 4.997 | 0.453 | 36.855 | 0.068 | 41.856 |
| Cycle 22 descent Max:Nov.1989 Min:Jun.1996 | NRLSSI1 | 49.420 | 0.250 | 12.343 | 1.066 | 3.890 | 4.999 | 0.311 | 22.306 | 0.102 | 55.503 |
| | NRLSSI2 | 53.857 | 0.279 | 12.175 | 1.078 | 3.485 | 4.594 | 0.384 | 28.299 | 0.075 | 42.076 |
| | SATIRE-T | 46.024 | 0.233 | 9.820 | 0.848 | 3.363 | 4.321 | 0.557 | 39.972 | 0.085 | 46.134 |
| | SATIRE-S | 64.121 | 0.304 | 13.546 | 1.096 | 4.044 | 4.869 | 0.629 | 42.293 | 0.085 | 43.436 |
| | CMIP6 | 59.005 | 0.291 | 12.862 | 1.083 | 3.767 | 4.722 | 0.507 | 35.494 | 0.080 | 42.850 |
| Cycle 23 ascent Max:Mar.2000 Min:Jun.1996 | NRLSSI1 | 40.457 | 0.286 | 10.116 | 1.223 | 3.177 | 5.716 | 0.237 | 23.772 | 0.070 | 53.692 |
| | NRLSSI2 | 42.445 | 0.360 | 9.594 | 1.388 | 2.730 | 5.879 | 0.274 | 32.971 | 0.040 | 36.438 |
| | SATIRE-T | 27.988 | 0.383 | 5.989 | 1.399 | 2.066 | 7.180 | 0.312 | 60.660 | 0.025 | 37.320 |
| | SATIRE-S | 49.546 | 0.473 | 10.446 | 1.704 | 3.041 | 7.380 | 0.439 | 59.580 | 0.034 | 35.442 |
| | CMIP6 | 46.007 | 0.407 | 10.021 | 1.514 | 2.887 | 6.492 | 0.357 | 44.849 | 0.037 | 35.713 |

**Table 2.** Solar cycle SSI variations for an average of five Solar Cycle amplitudes relative to ATLAS3 ($\Delta SSI$) in % and relative contribution of SSI changes to the TSI change ($\frac{\Delta SSI}{\Delta TSI}$) in % for the Lyman-$\alpha$ (121.5 nm), Far-UV (121–200 nm), Herzberg continuum/Hartley bands (201–242 nm), Hartley-/Huggings-bands (243–380 nm) and visible (381–780 nm) spectral ranges. $\pm$ 95% CI indicates the confidence interval.

| Time period | SSI dataset | 121.5 nm $\Delta SSI$ | $\frac{\Delta SSI}{\Delta TSI}$ | 121–200 nm $\Delta SSI$ | $\frac{\Delta SSI}{\Delta TSI}$ | 201–242 nm $\Delta SSI$ | $\frac{\Delta SSI}{\Delta TSI}$ | 243–380 nm $\Delta SSI$ | $\frac{\Delta SSI}{\Delta TSI}$ | 381–780 nm $\Delta SSI$ | $\frac{\Delta SSI}{\Delta TSI}$ |
|---|---|---|---|---|---|---|---|---|---|---|---|
| Average Cycle | NRLSSI1 | 46.101 | 0.253 | 11.514 | 1.081 | 3.628 | 5.069 | 0.289 | 22.411 | 0.094 | 55.018 |
| | $\pm$ 95% CI | 4.576 | 0.029 | 1.138 | 0.127 | 0.363 | 0.578 | 0.039 | 1.228 | 0.018 | 0.871 |
| | NRLSSI2 | 50.989 | 0.286 | 11.527 | 1.105 | 3.300 | 4.705 | 0.364 | 28.681 | 0.071 | 41.474 |
| | $\pm$ 95% CI | 5.931 | 0.053 | 1.341 | 0.203 | 0.393 | 0.843 | 0.061 | 3.098 | 0.023 | 3.533 |
| | SATIRE-T | 40.238 | 0.272 | 8.584 | 0.990 | 2.938 | 5.050 | 0.479 | 45.217 | 0.067 | 43.786 |
| | $\pm$ 95% CI | 9.150 | 0.079 | 1.945 | 0.291 | 0.660 | 1.504 | 0.129 | 10.883 | 0.035 | 4.617 |
| | SATIRE-S | 55.663 | 0.344 | 11.725 | 1.236 | 3.486 | 5.457 | 0.535 | 46.453 | 0.065 | 41.335 |
| | $\pm$ 95% CI | 7.910 | 0.084 | 1.685 | 0.302 | 0.510 | 1.241 | 0.085 | 8.435 | 0.021 | 3.832 |
| | CMIP6 | 53.340 | 0.311 | 11.627 | 1.159 | 3.395 | 5.029 | 0.450 | 36.944 | 0.069 | 41.468 |
| | $\pm$ 95% CI | 6.249 | 0.065 | 1.362 | 0.239 | 0.411 | 0.988 | 0.065 | 5.392 | 0.020 | 3.792 |

[Figure]

(a) Solar Cycle amplitude average        (b) Anomaly of Solar Cycle 22 (descent-2)

**Figure 1.** (a) Averaged solar cycle SSI variations for solar cycle amplitudes relative to ATLAS3 ($\Delta SSI$) in % for the Lyman-$\alpha$ (121.5 nm), Far-UV (121–200 nm), Herzberg continuum/Hartley bands (201–242 nm), Hartley-/Huggings-bands (243–380 nm) (multiplied by a factor of 10) and visible (381–780 nm) (multiplied by a factor of 100) spectral ranges; with the standard deviation indicated as error bar. (b) Anomaly of SSI variations for solar cycle 22 (descent-2) with respect to the averaged solar cycle shown in (a).

*In the Coddington et al., 2019 paper you reference, their Tables 3 and 5 show a larger change in integrated SSI (in the 100-200 nm bin) from solar cycle to solar cycle than occurs in differences across some of the datasets you use in your study. Similar to the above comment, you might want to bring this up earlier in Section 2 as well.*

5    The values in the Tables 3 and 5 of Coddington et al. (2019) are given in W m$^{-2}$ whereas the values of Table 1 in our study are given as percentage changes. When the values of Coddington et al. (2019) for the SSI range from 100 to 200 nm are converted to percentage changes, these value are not in contradiction to our values.

|  | $\frac{\Delta SSI}{\Delta TSI}$ NRLSSI2 | $\frac{\Delta SSI}{\Delta TSI}$ SATIRE-S |
|---|---|---|
| Coddington et al. (2019) (100–200 nm) | 1.17% | 1.20% |
| Table 1 this study (120–200 nm) | 1.13% | 1.19% |

The values of Coddington et al. (2019) are slightly larger, as they are calculated for the full descending phase of solar cycle 22
10    which ended in September 1996, whereas in our study we use November 1994 as solar minimum.

*e. It's possible this is jargon in the CCM community, but is it typical to use phrases of 'solar cycle response' for simulations where the transition from perpetual solar minimum to perpetual solar maximum is quite abrupt?*

15    In this study, where time slice simulations for solar minimum and maximum conditions are analysed, it is justified to interpret the significant differences between the time slice at solar maximum and solar minimum as the "solar response". The only parameters that changed are the prescribed SSI and TSI values.

**General comments**

*Page 2, lines 24 – 29: The end of the one paragraph is focusing on the CCM model "spread" caused by differences in spectral resolutions of the shortwave radiation parameterizations or photolysis in the models. The next paragraph begins with different spectral distribution of the SSI data set also impacting CCM models. In the 2nd case, you are referring to the magnitude of the SSI within a spectral bin and not differences in spectral resolution of the SSI observations, but this could easily be confused during the transition of one paragraph to the next.*

As requested by Reviewer 2, we have extended the paragraph about the model specific influences on the solar response. To better separate the SSI data set influences on the climatological state from the model influence on the solar response, we have added an introductory sentence.

*The SSI data prescribed in the models are the second source of uncertainty when modelling the solar response. Shapiro et al. (2011) investigated the influence of the 27-day variations of four different SSI observation on the chemistry of the upper mesosphere in a 1D radiative convective chemistry model. The deviant solar cycle behaviour of the SORCE measurements has motivated a number of CCM studies (e.g. Haigh et al., 2010; Merkel et al., 2011; Ball et al., 2011, 2016; Swartz et al., 2012) comparing simulations prescribing SORCE (Solar Radiation and Climate Experiment) SSI data and reconstructed SSI of the Naval Research Laboratory SSI (NRLSSI) or the Spectral And Total Irradiance REconstructions (SATIRE) model.*

*Page 3, line 3-4: You end with "the effects of the 11-year solar cycle differences in spectral distribution and amplitude . . . ". However, by adopting the common reference baseline spectrum, you have removed the effects of spectral distribution from the study. It's clear from your earlier text what you mean and that it's just an error here.*

The spectral distribution of the SSI amplitudes differs between the five SSI data sets. These charcteristics are preserved even when a common reference SSI is used for the solar minimum.

*Page 4, line 12: What type of scaling did you apply to make ATLAS 3 integrate to SORCE TIM TSI? Wavelength independent? "The extended ATLAS-3 spectrum was then scaled to obtain . . . "*

The applied scaling is wavelength independent. The integrated original ATLAS-3 SSI data (0.1 to 2395 nm) amounts to 1330.2 W m$^{-1}$. Adding the missing longer wavelengths requires (i) to scale the NRLSSI1 data set to the same SSI value at 2,395 nm as ATLAS-3, and (ii) to scale the SATIRE-S data set to the same SSI value at 99,975 nm as the combined ATLAS-3/NRLSSI1 sata set before the three SSI data sets are combined. The combined ATLAS-3/NRLSSI1/SATIRE-S SSI data set results in a TSI of 1382.9 W m$^{-1}$ and therefor the extended ATLAS3-3 SSI data are scaled with a constant factor of 0.9842, to achieve a TSI of 1361.05 W m$^{-1}$. We have changed the text to:

*To assure smooth transitions at 2,395 and 99,975 nm, the NRLSSI1 and SATIRE-S data sets are scaled accordingly. The extended ATLAS-3 spectrum was then scaled with a constant factor to obtain the integrated TSI of 1361.05 W m$^{-2}$ . . .*

*Page 4, line 20: Needs some clarification. The (facular brightening and sunspot darkening) indices themselves do not describe the relationship between sunspots and faculae on the Sun's disk and irradiance. The indices are derived from observations of proxies of faculae and sunspots. It's rather the scaling factors computed from the multiple linear regression of these indices with SSI observations that are used to scale the change in faculae and sunspots into a net, wavelength-dependent, irradiance change.*

We have reformulated this description.

*. . . based on the empirical, wavelength-dependent relationship between sunspot darkening and facular brightening on the solar disk with SSI changes. Indices which are derived from observations and proxies for sunspot darkening and faculae are used in regression models to determine the coefficients required to estimate the time-varying SSI changes.*

*Page 4, line 23: "The TSI changes are added . . . " should be "The SSI changes are added.."*

We are not sure about that. Maybe it is confusing that NRLSSI1 and NRLSSI2 are given in brackets? The intention is to emphasize the different TSI references in NRLSSI/TSI1 and NRLSSI/TSI2.

*Page 4, line 27: While Viereck et al., 2001 is a perfectly appropriate reference for a general discussion of the Mg II index, the correct citation for the University of Bremen Mg II index reference is Snow, M., Weber, M., Machol, J., Viereck, R., & Richard, E. (2014). Comparison of magnesium II core–to–wing ratio observations during solar minimum 23/24. Journal of Space Weather and Space Climate, 4, A04. https://doi.org/10.1051/swsc/2014001*

The reference "Viereck et al. (2001)" is now replaced by "Snow et al. (2014)".

*Page 5, line 23-24: I am aware that CMIP6 SSI and TSI data are the average of output from NRLSSI2 and SATIRE-S. However, it's unclear to me the relation of this is to your choice of using data from November 1989 and November 1994 in the study?*

The main reason for choosing November 1994 as the base state for solar minimum condition is the timing of the ATLAS3 reference SSI data. With this choice of the solar minimum the natural choice was to take November 1989 for the solar maximum condition, as the November 1994 is located in the descent to the ending solar minimum of cycle 22 in August 1996. So the choice of this cycle is not related to CMIP6.

*Page 6, EMAC section: I'm not an expert on CCMs but I find the description of EMAC difficult to read. It doesn't flow as easily as the following section on WACCM, and the acronyms aren't defined. I would suggest some word–smithing to bring it up to the same high quality as the rest of the paper.*

The EMAC section is now slightly revised.

*Page 7, between WACCM and section and the start of Section 3.1: Again because I'm not an expert on CCM's, it would be nice to have a summarizing sentence or two here as a take home message for the non-expert. Are these suitable models to compare, and are there obvious reasons why their unique setup and execution would lead you to expect differences in their outputs?*

We have included the following paragraph to the introduction of Section 3.

*Both CCMs have a good spectral resolution of their SW radiation and photolysis parametrization and therefore are well suited for this study. The main difference between EMAC and WACCM, as applied here, is WACCM's model top in the lower thermosphere which allows for a better representation of the chemical processes in the upper mesosphere in WACCM.*

*Page 7, lines 27-28: One too many of each of the words, "both" and "simulations".*

One of these "both" is now deleted.

*Page 9, Section 7: A general comment in this section is to make it is more clear that majority of the uncertainty, or spread, in the CCM output comes from internal variability in the CCM's. Only a fraction of the model spread can be attributed to differences in SSI datasets or differences in the CCM's themselves. (If I understood correctly).*

This is correct. It is stated in Section 5:
*Note that the contributions of the variances explained by the SSI data set, the CCM, and the interaction of both in Figure 4 do not add up to 100%, as often the random contribution to the total variance is largest.*

In Section 7 we have added:
*However, in the upper stratosphere/lower mesosphere the largest fraction of solar cycle response variance is random and not related to differences in SSI data sets or the applied CCM.*

*Page 13, line 20: "...10-40% of the variability of the solar signal [ insert of what component, heating rate, temperature, etc.] in the stratosphere and ..."*

We have added the additional information.

*The differences in the SSI amplitude are responsible for 10% of the temperature, 30% of the ozone, and up to 40% of the SW heating rate variability of the solar response in the stratosphere and lower mesosphere.*

*Page 20, line 8-10: Is there a transition in thought from the sentence ending on line 8 about the distinct differences that appear for SATIRE-T to the next sentence discussing how reduced solar cycle amplitude explain the weaker solar signals in temperature? Does that 2nd sentence also refer to SATIRE-T? If so, Table 1 shows that SATIRE-T has a stronger solar cycle amplitude in the 201-242 nm range, not weaker.*

The 2nd sentence also refers to SATIRE-T. The $\Delta SSI$ and the $\frac{\Delta SSI}{\Delta TSI}$ in Table 1 show opposing characteristics. Whereas for SATIRE-T $\frac{\Delta SSI}{\Delta TSI}$ is largest it is lowest for $\Delta SSI$, and it is $\Delta SSI$ which effects the solar responses in SW heating rate and temperature.

*Page 21, line 3-4: You provide support that downward transport of thermospheric photolysis reactants is needed to realistically simulate solar cycle effects. Is this a new finding for the CCM community? It seems to me that you might emphasize the importance of this in guiding CCM model development and directing CCM advances.*

We have added a remark on a possible 'upper boundary condition' for thermospheric photolysis reactants for CCMs with a model top in the upper mesosphere.

*Some of these effects could be included in CCMs with a model top in the upper mesosphere by a thoroughly formulated upper boundary condition, as already included for $NO_y$ produced in the thermosphere by auroral and medium-energy electrons*

*Figures: I was finding that the significance hatching in the figures was very difficult to see, particularly in the middle and right hand columns of Figure 1. However, when I look today, it's much clearer on-screen. Perhaps it is just a problem with my printer.*

We have changed the single hatching to double hatching.

**References**

Ball, W. T., Unruh, Y. C., Krivova, N., Solanki, S. K., and Harder, J. W.: Solar irradiance variability : a six-year comparison between SORCE observations and the SATIRE model, Astronomy & Astrophysics, 530, A71, doi:10.1051/0004-6361/201016189, 2011.

Ball, W. T., Haigh, J. D., Rozanov, E. V., Kuchar, A., Sukhodolov, T., Tummon, F., Shapiro, A. V., and Schmutz, W.: High solar cycle spectral variations inconsistent with stratospheric ozone observations, Nature Geoscience, 9, 206–209, doi:10.1038/ngeo2640, 2016.

Coddington, O., Lean, J., Pilewskie, P., Snow, M., Richard, E., Kopp, G., Lindholm, C., DeLand, M., Marchenko, S., Haberreiter, M., and Baranyi, T.: Solar Irradiance Variability: Comparisons of Models and Measurements, Earth and Space Science, 34, 2019EA000 693, doi:10.1029/2019EA000693, 2019.

Haigh, J. D., Winning, A. R., Toumi, R., and Harder, J. W.: An influence of solar spectral variations on radiative forcing of climate, Nature, 467, 696–699, doi:10.1038/nature09426, 2010.

Merkel, A., Harder, J. W., Marsh, D. R., Smith, A. K., Fontenla, J. M., and Woods, T. N.: The impact of solar spectral irradiance variability on middle atmospheric ozone, Geophysical Research Letters, 38, 1–6, doi:10.1029/2011GL047561, 2011.

Shapiro, A. V., Rozanov, E., Egorova, T., Shapiro, A. I., Peter, T., and Schmutz, W.: Sensitivity of the Earth's middle atmosphere to short-term solar variability and its dependence on the choice of solar irradiance data set, Journal of Atmospheric and Solar-Terrestrial Physics, 73, 348–355, doi:10.1016/j.jastp.2010.02.011, 2011.

Swartz, W. H., Stolarski, R. S., Oman, L. D., Fleming, E. L., and Jackman, C. H.: Middle atmosphere response to different descriptions of the 11-yr solar cycle in spectral irradiance in a chemistry-climate model, Atmos. Chem. Phys., 12, 5937–5948, doi:10.5194/acp-12-5937-2012, 2012.

Yeo, K. L., Ball, W. T., Krivova, N. A., Solanki, S. K., Unruh, Y. C., and Morrill, J.: UV solar irradiance in observations and the NRLSSI and SATIRE-S models, J. Geophys. Res.: Space Physics, doi:10.1002/2015JA021277, 2015JA021277, 2015.